# LEARNING IMBALANCED DATA WITH BENEFICIAL LABEL NOISE

## ABSTRACT

Data imbalance and label noise are common factors hindering classifier's performance. Data-level approaches to addressing imbalanced learning usually involve resampling by adding or removing samples, which often results in information loss or generative errors. Building upon theoretical studies of the impact of imbalance ratio on decision boundaries across various evaluation metrics in binary classification, it is uncovered that introducing appropriate label noise can alter the biased decision boundaries and thus enhance the performance of classifiers in imbalanced learning. In this paper, we introduce the Label-Noise-based Rebalancing (LNR) approach to solve both binary and multi-class imbalanced classifications by employing a novel design of asymmetric label noise model. In contrast to other data-level methods, our approach is easy to implement and alleviates the issues of informative loss and generative errors. We validated the superiority of this method on synthetic and real-world datasets. More importantly, our LNR approach can integrate seamlessly with any classifiers and other algorithm-level methods for imbalanced learning. Overall, our work opens up a new avenue for addressing imbalanced learning, highlighting the potential advantages of balancing data through beneficial label noise.

## 1 INTRODUCTION

Imbalanced data is pervasive in many real-world problems, including fraud transactions (Phua et al., 2004), disease diagnoses (Fotouhi et al., 2019), and behavioral analysis (Azaria et al., 2014). These datasets are often characterized by a significant disparity in the number of samples between classes, with some classes possessing a large number of samples while others containing very few samples. This imbalance poses a challenge for standard classification algorithms, which typically require either a similar sample size across classes or a sufficient number of minority class samples to achieve a desired level of efficacy (He & Garcia, 2009; Krawczyk, 2016).

In practice, the rarity of samples (often corresponding to abnormal cases) makes the minority class the primary focus, rendering accuracy an inadequate metric for classification, especially in the binary case. Commonly used metrics for imbalanced learning, such as the F1 score, G-mean, and Area Under the Curve (AUC) for binary classification (He & Garcia, 2009; Krawczyk, 2016), or Recall, Precision and Many/Medium/Few-shots for evaluating step-wise or long-tailed multi-class data, better reflect performance in classifying minority classes samples (Cao et al., 2019; Wang et al., 2021a). In binary classification, this paper theoretically demonstrates that the decision boundary optimized for accuracy deviates from one that maximizes other metrics, such as the F1 score, impairing classifier performance. When optimizing for accuracy, classifiers tend to classify all samples into the majority class as the class imbalance becomes extreme, thus limiting the F1 score. Extensive research has attempted imbalanced learning through **resampling** techniques, which aim to create a more balanced dataset by either removing or adding samples (Kaur et al., 2019; Napierala & Stefanowski, 2016; Wang et al., 2020; 2021a). These data-level methods often come with drawbacks such as potential information loss, over-fitting, and the introduction of generative errors. Another line of research focuses on algorithm-level methods (Chawla et al., 2004; Cao et al., 2019; Wang et al., 2021b; Li et al., 2022a), which adjust the learning algorithm itself. However, these methods are typically tailored to specific classifiers or problem setups, limiting their general applicability.

Presentation or introduction of **label noises**, which solely flips the label of instances to another class, significantly impacts the decision boundary of classifiers. This paper theoretically investigates how the imbalance ratio affects the decision boundaries under different metrics like accuracy and F1 score in binary classification. Inspired by recent studies (Ahfock & McLachlan, 2021; Cannings et al., 2020) on **harmless** label noises that do not degrade classifier performance under certain conditions, we propose a novel Label-Noise-Re-balancing (LNR) approach that introduces artificial label noise to reallocate imbalanced data labels and thus modify the decision boundary to improve classifier performance. In addition, we extend the LNR approach to address multi-class classification problems where class imbalance is present. Our proposed LNR method operates at the data level, making it flexible and adaptable for use with any classifier or algorithm-level method. This seamless integration allows for enhanced performance in both binary and multi-class settings, without being restricted to specific classifiers or requiring significant algorithmic modifications, thereby offering a robust solution to imbalanced learning challenges.

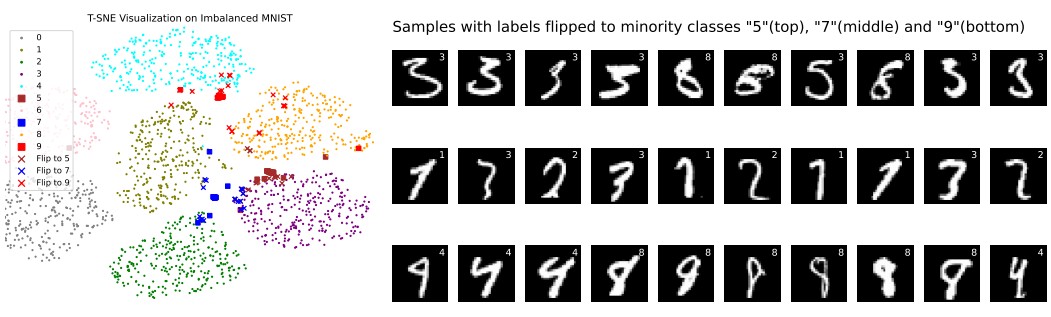

(a) T-SNE visualizations.      (b) Top-10 flipped labels into minority classes 5,7,9 by **LNR**.

Figure 1: Adding label noises to the imbalanced MNIST data. (a) shows the hidden vectors of the minority class (squares) and the majority class (dots), where crosses indicate the top ten majority class samples with flipped probabilities for each minority class, which are plotted in (b).

Figure 1 provides an example of adding beneficial label noises to imbalanced MNIST data with minority classes "5", "7", and "9". The flipped labels correspond to the samples with the top 10 probabilities of being relabeled as each of the minority classes. In Figure 1(a), we observe that these flipped samples are located at the boundaries between different classes, and they exhibit similar features to the minority class samples as shown in Figure 1(b). By applying our LNR approach, the classifier's accuracy on the extremely imbalanced MNIST dataset, with only 30 samples per minority class at an imbalance ratio of 200, improved from 89.93% to 94.75%.

## 2 RELATED WORKS AND CONTRIBUTIONS

**Imbalance learning.** In imbalanced binary classification, resampling is a one-time data-level approach that balances sample sizes through under-sampling or over-sampling. Over the past decades, extensive approaches have been proposed to address the issues of information loss from under-sampling and over-fitting from over-sampling. Solutions include methods for selectively removing samples based on distribution information, such as Tomek-links (Tomek, 1976), ENN (Wilson & Martinez, 2000), NCL (Li et al., 2022a), OSS (Kubat & Matwin, 2000), and ClusterCentroids (Lin et al., 2017), as well as techniques for synthesizing new minority class samples based on minority class distribution information, such as SMOTE (Chawla et al., 2002), Borderline-SMOTE (Han et al., 2005), and ADASYN (He et al., 2008). These methods are often combined with conventional machine learning algorithms.

In multi-class classification, imbalance scenarios such as long-tailed or step-wise sample size distributions become more complex due to multiple minority classes and varying imbalance ratios. Algorithm-level methods can be divided into two main types: multi-expert and multi-stage approaches. Multi-expert methods, such as RIDE (Wang et al., 2021b) and NCL (Li et al., 2022a), reduce uncertainty in minority (tail) classes by employing multiple experts, enhancing representation through knowledge aggregation. Multi-stage methods, like LDAM-DWR (Cao et al., 2019)

and GCL (Li et al., 2022b), improve tail class accuracy by splitting training into feature learning and classifier learning phases, using techniques like logit margin adjustments and reweighting. Data-level methods enrich minority classes by generating new samples using generative models or auto-encoders (Wang et al., 2020; Schwartz et al., 2018; Wang et al., 2018; 2021a). These generative solutions typically rely on high-quality pre-trained models or encoders to minimize generative errors, which can increase model complexity and uncertainty. In recent state-of-the-art (SOTA) approaches, a combination of data-level and algorithm-level techniques is common, as they complement each other to enhance performance in imbalanced learning.

**Harmless Label Noise.** Instance-dependent noise (Frénay & Verleysen, 2013; Manwani & Sastry, 2013), where the label flip-rate (or mislabel-rate) depends on its true class and instance features, is the most general case of label noise. It is commonly assumed that samples with similar features are more likely mislabeled (Cannings et al., 2020; Goldberger & Ben-Reuven, 2022; Song et al., 2022), particularly in the class-overlapping region. While label noise is typically harmful and most research (Goldberger & Ben-Reuven, 2022; Liu & Tao, 2015; Patrini et al., 2017; Xiao et al., 2015; Zhu et al., 2003) has mainly focused on mitigating the impact of label noise on the decision boundary, recent studies (Ahfock & McLachlan, 2021; Cannings et al., 2020) have demonstrated that under certain conditions, label noise can be harmless or even beneficial. In Ahfock & McLachlan (2021), the label noise model, proportional to classification difficulty, exhibits symmetry around the decision boundary, preserving the decision boundary of logistic regression unchanged in a balanced dataset. Contrarily, in Cannings et al. (2020), the noise model is proportional to the posterior probability of the opposite class, making the regret ratio less or equal to 1, thereby slightly improving the classifier performance. However, both works primarily focus on balanced data and do not explore how observed phenomena can be leveraged to develop methods for improving classifier performance in imbalanced real-world data by introducing harmless label noise.

**Motivation and Contributions.** Previous works (Ahfock & McLachlan, 2021; Cannings et al., 2020) on harmless label noise inspire the development of a methodology to reshape decision boundaries through label noise under general models, extending from harmless to beneficial categories when addressing real-world imbalanced learning problems. It is worth mentioning that the label noise employed in this paper is asymmetric such that only the samples from the majority class have a certain probability of being flipped to the minority class, aligning with imbalanced learning. This contrasts with the symmetric label noise considered in Ahfock & McLachlan (2021); Cannings et al. (2020), where the minority samples may also be flipped to the majority class. This paper revisits fundamental questions in imbalanced learning: **A)** How does data imbalance bias the classifier's decision boundary? **B)** How can biased decision boundaries be corrected through label flipping to improve classification performance? **C)** How to introduce artificial label noises to improve classifier performance in both binary and multi-class settings?

Addressing these questions, the main contributions of this paper are: **a)** Theoretical justification of how imbalance ratio biases the decision boundary and quantification of its deviation under different evaluation metrics in binary classification. **b)** Definition of a beneficial label noise model that corrects the biased classifier's decision boundary by harmlessly reallocating imbalanced data labels. **c)** Development of a novel methodology (LNR) to introduce carefully designed beneficial label noise to imbalanced data, improving the performance of classifiers on predicting minority class samples in both binary and multi-class classification. Experiment results on synthetic and real-world data validated the superiority of our proposed method and its versatility in integrating with existing algorithm-level methods.

## 3 IMPACT OF IMBALANCE RATIO ON DECISION BOUNDARY

In this section, we analyze the impact of the imbalance ratio on decision boundaries across various evaluation metrics in binary classification, which motivates the development of our Label-Noise-based Re-balancing (LNR) approach. Assume that two imbalanced classes have prior probabilities $\pi_1 = \Pr(Y = 1)$ and $\pi_0 = \Pr(Y = 0) = 1 - \pi_1$, respectively, where $Y = 1$ refers to the minority (positive) class while $Y = 0$ represents the majority (negative) class. In the imbalanced scenario, the sample class priors satisfy $\pi_1 \ll \pi_0$ and the imbalance ratio is $\pi_0/\pi_1$. For class $c \in \{0, 1\}$ and the sample space $\mathcal{X} \subset \mathbb{R}^d$, we denote the class-conditional distribution as $X \mid Y = c \sim P_c$ and the marginal distribution of $X$ as $P_X(x) = \pi_1 P_1(x) + \pi_0 P_0(x)$.

### 3.1 IMPACT OF IMBALANCE RATIO ON OPTIMAL BAYESIAN DECISION BOUNDARY

Define $\eta(x) = \Pr(Y = 1 \mid X = x)$. The Bayesian decision boundary of the optimal classifier that maximizes the accuracy in the sample space $\mathcal{X}$ is defined as $S = \{x \in \mathcal{X} : \eta(x) = 0.5\}$.

**Lemma 1.** *The optimal Bayesian decision boundary is $S = \{x \in \mathcal{X} : P_1(x)/P_0(x) = \pi_0/\pi_1\}$.*

If the data is balanced with $\pi_0 = \pi_1$, Lemma 1 implies that the Bayesian decision boundary corresponds to the intersection points of $P_0(x)$ and $P_1(x)$. However, when the data is imbalanced with $\pi_1 \ll \pi_0$, the decision boundary intrudes deeply into the minority class region, increasing the risk of erroneously classifying the minority class and leading to a diminished true positive rate approaching 0. In pursuit of optimizing overall classification accuracy, the predictive ability of minority class samples, which is crucial in practical applications, is compromised. In extremely imbalanced cases, the accuracy remains high even if all samples are classified into the majority class.

**Example 1.** *If $P_0 \sim N(\mu_0, \Sigma)$ and $P_1 \sim N(\mu_1, \Sigma)$, the optimal Bayesian decision boundary is $S = \left\{ x \in \mathcal{X} : \omega^T x + \beta = \ln(\pi_0/\pi_1) \right\}$ with $\omega = \Sigma^{-1}(\mu_1 - \mu_0)$ and $\beta = \frac{1}{2}\mu_0^T \Sigma^{-1}\mu_0 - \frac{1}{2}\mu_1^T \Sigma^{-1}\mu_1$. For 1-dimensional normal distributions with common variance $\sigma^2$, the decision boundary is $S = \left\{ x \in \mathcal{X} : x = \frac{1}{2}(\mu_1 + \mu_0) + \frac{\sigma^2}{\mu_1 - \mu_0} \ln(\frac{\pi_0}{\pi_1}) \right\}$. If $\pi_0 = \pi_1$, the Bayesian decision boundary lies on the midpoint of $\mu_1$ and $\mu_0$. However, for imbalanced data with $\pi_1 < \pi_0$, the decision boundary shifts towards the minority class sample mean $\mu_1$ as $\ln(\pi_0/\pi_1) > 0$. The term $\sigma^2/(\mu_1 - \mu_0)$ determines the relevance of the imbalance ratio $\pi_0/\pi_1$ to the deviation of the decision boundary from the midpoint, which is inversely proportional to $\mu_1 - \mu_0$. Thus, the decision boundary shift depends on the imbalance ratio and class mean difference, indicating the degree of class overlap.*

Lemma 1 is on the Bayesian decision boundary that maximizes the population-level accuracy. Similar phenomena exist in many classical classification algorithms, such as the $k$-nearest neighbors (KNN). We refer to Section B in the Appendix for more detailed discussions and demonstrations.

### 3.2 IMPACT OF IMBALANCE RATIO ON F1 SCORE

In many scientific applications, such as fraud detection, the predictive performance of minority class samples is often prioritized over overall accuracy. Precision and recall are two metrics that prioritize the positive samples. Their definitions are Precision $= \mathrm{TP}/(\mathrm{TP} + \mathrm{FP})$ and Recall $= \mathrm{TP}/(\mathrm{TP} + \mathrm{FN})$, where TP, FP, and FN represent true positives, false positives, and false negatives, respectively. Precision and recall are inversely proportional and constrained by each other. F1 score considers the performance of precision and recall, and it is widely used in imbalanced learning:

$$\mathrm{F1} = \frac{2 \times \text{Precision} \times \text{Recall}}{\text{Precision} + \text{Recall}} = \frac{2\mathrm{TP}}{2\mathrm{TP} + \mathrm{FP} + \mathrm{FN}}.$$

For a given classifier, TP, FP, and FN are discrete statistics based on its decision boundary. Let $N$ be the total sample size of two classes. By the law of large numbers, as $N \to \infty$, $\mathrm{TP}/N \to \Pr(\hat{Y} = 1, Y = 1)$, $\mathrm{FP}/N \to \Pr(\hat{Y} = 1, Y = 0)$, and $\mathrm{FN}/N \to \Pr(\hat{Y} = 0, Y = 1)$ in probability , where $\hat{Y}$ is the predicted label of $X$ with true label $Y$.

We consider the classifier in the form of $\hat{Y} = \mathbb{I}(\omega^T(X - x_0) > 0)$ for some $\omega \in \mathbb{R}^d$ and $x_0 \in \mathcal{X}$, here $\mathbb{I}(\cdot)$ is the indicator function. It follows that

$$\Pr(\hat{Y} = 1, Y = 1) = \Pr(Y = 1)\Pr(\hat{Y} = 1 \mid Y = 1) =: \pi_1 \mathcal{P}_1(x_0),$$

where $\mathcal{P}_1(x_0) = \Pr(\omega^T(X - x_0) > 0 \mid Y = 1)$. In addition, $\Pr(\hat{Y} = 1, Y = 0) = \pi_0 \mathcal{P}_0(x_0)$ with $\mathcal{P}_0(x_0) = \Pr(\omega^T(X - x_0) > 0 \mid Y = 0)$ and $\Pr(\hat{Y} = 0, Y = 1) = \pi_1(1 - \mathcal{P}_1(x_0))$.

To study the impact of the imbalance ratio on the F1 score and compare its optimal decision boundary to the Bayesian decision boundary, we define a population version of the F1 score as

$$\mathcal{F}1(x_0) = \frac{2\pi_1 \mathcal{P}_1(x_0)}{\pi_1 \mathcal{P}_1(x_0) + \pi_0 \mathcal{P}_0(x_0) + \pi_1}. \tag{1}$$

**Lemma 2.** *The decision boundary that maximizes $\mathcal{F}1(x_0)$ for $x_0 \in \mathcal{X}$ is*

$$S^{\mathrm{F1}} = \left\{ x_0 \in \mathcal{X} : \frac{P_1(x_0)}{P_0(x_0)} = \frac{\mathcal{F}1(x_0)}{2 - \mathcal{F}1(x_0)} \frac{\pi_0}{\pi_1} \right\}.$$

The proof of Lemma 2 implies $\eta(x_0) = 0.5 \times \mathcal{F}1(x_0)$ for $x_0 \in S^{\mathrm{F}1}$, indicating that the probability of $x_0$ belongs to the positive class equals half of the maximum population F1 score $\mathcal{F}1(x_0)$ on the decision boundary. Consequently, the optimal F1 score decision boundary coincides with the optimal Bayesian decision boundary when $\mathcal{F}1(x_0) = 1$ for $x_0 \in S^{\mathrm{F}1}$. In such case, $\Pr(\hat{Y} = 1, Y = 0) = \Pr(\hat{Y} = 0, Y = 1) = 0$, indicating full separability between the two classes.

However, when class overlap exists such that $\mathcal{F}1(x_0) < 1$, we observe $P_1(x_0)/P_0(x_0) < P_1(x)/P_0(x)$ for $x_0 \in S^{\mathrm{F}1}$ and $x \in S$. This demonstrates that the optimal F1 score decision boundary deviates from the optimal Bayesian decision boundary that maximizes accuracy towards the majority class region. The discrepancy between $S$ and $S^{\mathrm{F}1}$ is directly associated with $\frac{\mathcal{F}1(x_0)}{2-\mathcal{F}1(x_0)}$, which becomes severe when the maximum F1 score decreases and approaches 0. In addition, $\frac{\mathcal{F}1(x_0)}{2-\mathcal{F}1(x_0)}$ is weighted by the imbalance ratio $\pi_0/\pi_1$, which further amplifies the deviation.

Lemma 2 indicates that for $x_0 \in S^{\mathrm{F}1}$, the maximum F1 score satisfies $\frac{\mathcal{F}1(x_0)}{2-\mathcal{F}1(x_0)} = \frac{P_1(x_0)}{P_0(x_0)}\frac{\pi_1}{\pi_0}$, which decreases as the $\pi_1/\pi_0$ decreases. Consequently, the optimal F1 score decreases as the imbalance ratio becomes more severe, limiting the best performance of classifiers in terms of the F1 score. In addition, optimizing accuracy under an imbalance ratio would further sacrifice the F1 score.

**Example 2.** *Figure 2 exhibits the relationship between the deviation of $S$ and $S^{\mathrm{F}1}$ with the imbalance ratio $\pi_0/\pi_1$ and the class overlap measured by $|\mu_1 - \mu_0|$ when $P_0 \sim N(\mu_0, \sigma^2)$ and $P_1 \sim N(\mu_1, \sigma^2)$. In Figure 2(a), where the data is balanced and the class overlap is small, the deviation between the optimal accuracy and the maximum F1 score is negligible, and the decision boundaries align closely. However, in Figure 2(b), for balanced data with a larger class overlap, the deviation between $S$ and $S^{\mathrm{F}1}$ increases noticeably. In Figure 2(c), with the same mean difference as in Figure 2(b), the deviation between $S$ and $S^{\mathrm{F}1}$ grows as the imbalance ratio increases. Notably, the maximum F1 score decreases as the imbalance ratio increases, and the F1 score is significantly lower at the Bayesian decision boundary.*

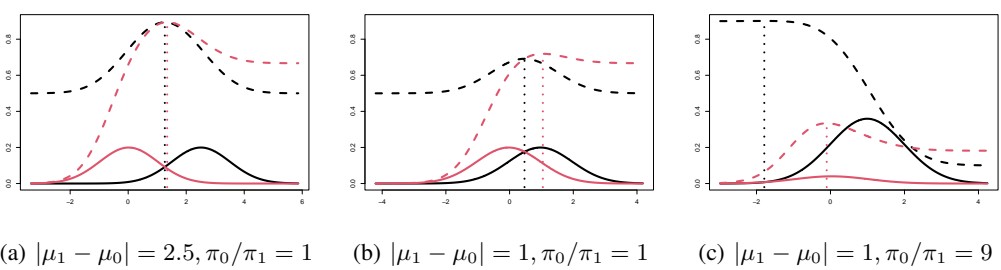

(a) $|\mu_1 - \mu_0| = 2.5, \pi_0/\pi_1 = 1$     (b) $|\mu_1 - \mu_0| = 1, \pi_0/\pi_1 = 1$     (c) $|\mu_1 - \mu_0| = 1, \pi_0/\pi_1 = 9$

Figure 2: Comparison of decision boundaries of $S$ (vertical black dashed line) and $S^{F1}$ (vertical red dashed line). The red solid line represents $\pi_1 P_1(x)$, and the black solid line represents $\pi_0 P_0(x)$. The black dashed curve represents the accuracy with respect to $x$, and the red dashed curve is $\mathcal{F}1(x)$.

## 4 LABEL-NOISE-BASED RE-BALANCING APPROACH

Motivated by the distortion of decision boundaries caused by the imbalance ratio, as observed in the last section, we propose a novel *Label-Noise-based Re-balancing (LNR)* approach. This method introduces beneficial asymmetric label noise to enhance the performance of classification algorithms in imbalanced learning in both binary and multi-class settings.

### 4.1 METHODOLOGY

The proposed LNR method consists of two steps: (1) introduce artificial label noises to the original data, and (2) train a classifier on the noisy data. We first focus on binary classification to motivate and justify the introduction of label noise, which plays a crucial role in correctly adjusting the decision boundary on the relabeled data. Extensions to the multi-class setting are explored in Section 4.3.

Suppose the original data $(X, Y) \sim \mathrm{P}$. Define $\rho(x) = \Pr(Y^* = 1 \mid X = x, Y = 0)$ as the probability of flipping a sample from the true label $Y = 0$ to the noisy label $Y^* = 1$, similarly, define $\gamma(x) = \Pr(Y^* = 0 \mid X = x, Y = 1)$. With the flipped labels, we obtain the noisy data pairs $(X, Y^*) \sim \mathrm{P}^*$. Consequently, $\eta^*(x) = \Pr(Y^* = 1 \mid X = x) = \eta(x)[1 - \gamma(x)] + [1 - \eta(x)]\rho(x)$.

The label noises discussed in Ahfock & McLachlan (2021); Cannings et al. (2020) are symmetric and proportional to classification difficulty $\eta(x)$, where both positive and negative samples are equally likely to be flipped on the decision boundary. The classifier remains consistent on such symmetric label noises, counterbalancing the distortion and leaving the decision boundary unchanged. For example, with $\rho(x) = \eta(x), \rho(x) + \gamma(x) = 1$, we have $\eta^*(x) = \eta(x)$, making the noise model harmless and preserving the decision boundary.

However, in imbalanced data, the overwhelming number of majority class samples invading the minority class region often results in significant misclassification of minority class samples. To achieve a higher F1 score by adjusting the decision boundary and reducing the imbalance ratio near the decision boundary, we consider an asymmetric label noise model that only flips majority class samples to minority class samples. This approach shifts the biased decision boundary toward the majority class region rather than neutralizing the shifts with symmetric noise. Since deeply invaded majority class samples cause greater harm, we aim for a noise model directly proportional to the extent of this invasion. Therefore, we propose the flipped rates $\rho(x)$ and $\gamma(x)$ satisfy:

$$\rho(x) \propto \eta(x), \quad \gamma(x) = 0. \tag{2}$$

In this model, the probability of majority class samples being flipped to minority class is proportional to their probability of being classified as minority class, which is intuitive. Setting $\gamma(x) = 0$ means that only majority class samples with $Y = 0$ are flipped at the rate $\rho(x)$. Using this label noise model, we can generate the noisy data, as outlined in Algorithm 1. As an example shown in Figure 3, the asymmetric label noise model pushes the biased decision boundary (left vertical dashed line) on imbalanced data distribution (solid curves) toward the majority region at the corrected decision boundary (right vertical dashed line) on the data distribution with label noises (dashed curves).

By introducing artificial label noises with a well-behaved flip-rate function $\rho(x)$, the deviation of the decision boundary caused by asymmetric label noise can correct the biased decision boundary on imbalanced data. The flipped majority class samples enrich the minority class while simultaneously reducing the number of the majority class. The Bayesian decision boundary that maximizes accuracy based on the relabeled sample $(X, Y^*)$ is $S^* = \{x^* \in \mathcal{X} : \eta^*(x^*) = 0.5\}$.

**Lemma 3.** *The optimal Bayesian decision boundary based on $(X, Y^*)$ is*

$$S^* = \left\{ x^* \in \mathcal{X} : \frac{P_1(x^*)}{P_0(x^*)} = [1 - 2\rho(x^*)] \frac{\pi_0}{\pi_1} \right\},$$

*conditional on that the flip-rate on the decision boundary satisfies $\rho(x^*) < 0.5$*

As $1 - 2\rho(x^*) < 1$, Lemma 3 indicates that the Bayesian decision boundary on noisy data shifts back to the majority class region. While this may sacrifice some overall accuracy, it improves the classification accuracy of minority class samples, thereby enhancing the F1 score. Furthermore, the decision boundary that maximizes accuracy on the noisy data coincides with the optimal F1 decision boundary on the original data when $\frac{\mathcal{F}1(x^*)}{2 - \mathcal{F}1(x^*)} = 1 - 2\rho(x^*)$.

Similar as in Section 3.2, consider the classifier $\hat{Y}^* = \mathbb{I}(\omega^T(X - x_0^*) > 0)$ for some $\omega \in \mathbb{R}^d$ and $x_0^* \in \mathcal{X}$ trained from the noisy data. Define the corresponding population F1 score as $\mathcal{F}1^*(x_0^*) = \frac{2\pi_1^* \mathcal{P}_1^*(x_0^*)}{\pi_1^* \mathcal{P}_1^*(x_0^*) + \pi_0^* \mathcal{P}_0^*(x_0^*) + \pi_1^*}$, where $\mathcal{P}_1^*(x_0^*) = \Pr(\omega^T(X - x_0^*) > 0 \mid Y^* = 1)$ and $\mathcal{P}_0^*(x_0^*) = \Pr(\omega^T(X - x_0^*) > 0 \mid Y^* = 0)$, $\pi_1^* = \Pr(Y^* = 1)$ and $\pi_0^* = \Pr(Y^* = 0) = 1 - \pi_1^*$. Then, the decision boundary that maximizes $\mathcal{F}1^*(x_0^*)$ based on relabeled sample $(X, Y^*)$ is

$$S^{\mathrm{F1}*} = \left\{ x_0^* \in \mathcal{X} : \frac{\mathcal{P}_1^*(x_0^*)}{\mathcal{P}_0^*(x_0^*)} = \frac{\mathcal{F}1^*(x_0^*)}{2 - \mathcal{F}1^*(x_0^*)} \frac{\pi_0^*}{\pi_1^*} \right\}.$$

Under the relabeled data, it is expected that $\mathcal{P}_1^*(x_0^*)/\mathcal{P}_0^*(x_0^*)$ and $\pi_1^*/\pi_0^*$ will increase compared to $P_1(x^*)/P_0(x^*)$ and $\pi_1/\pi_0$. Therefore, compared to the original decision boundary $S^{\mathrm{F1}}$ that maximizes $\mathcal{F}1(x_0)$, we expect an increase in $\mathcal{F}1^*(x_0^*)$, the maximum F1 score under the noisy data.

In summary, with an asymmetric instance-dependent noise model, the artificial label noise can correct the biased decision boundaries caused by class imbalance, improving the classification accuracy of minority class samples. The F1 score is expected to increase under the noisy data, demonstrating the effectiveness of the asymmetric label noise model in imbalanced classification.

## 4.2 CONSTRUCTING LABEL NOISE MODEL IN BINARY CLASSIFICATION

In practice, estimating the flip-rate $\rho(x)$, which is proportional to $\eta(x)$ in binary classification, becomes essential. We propose utilizing the estimated posterior probabilities $\hat{\eta}(x)$ obtained from an arbitrary classifier trained on imbalanced data to approximate $\eta(x)$, thereby constructing the label noise model $\rho(x)$. As discussed in Section 3.1, the decision boundary of the classifier trained with imbalanced data is biased from the optimal decision boundary, often underestimating $\eta(x)$. However, we can still leverage these estimated probabilities to capture the similarities to the minority class and construct an appropriate noise model using Algorithm 1.

Step 1. Train a classifier $C_f$ on the original imbalanced data and obtain the corresponding posterior probabilities $\hat{\eta}[i] = C_f(X[i])$. Although $\hat{\eta}[i]$ may be biased, the majority class samples with features similar to those of minority class samples still tend to exhibit relatively higher posterior probabilities of being classified into minority classes. As a result, these samples are expected to have relatively higher probabilities of being relabeled as the minority class.

Step 2. Employ $z$-score standardization on $\hat{\eta}[\text{Ind}_{MA}]$ to get $\mathcal{Z}[\text{Ind}_{MA}]$, where $\text{Ind}_{MA}$ is the index of majority class samples. This is used to magnify the differentiation in the posterior probabilities of the majority class samples. The deviation level, measured by $\mathcal{Z}[\text{Ind}_{MA}]$, ranks all majority class samples, proportional to the similarity with minority class samples.

Step 3. Rescale $\mathcal{Z}[\text{Ind}_{MA}]$ to $[0, 1]$ by calculating $\rho[\text{Ind}_{MA}] = \max(\texttt{tanh}(\mathcal{Z}[\text{Ind}_{MA}] - t_{flip}), 0)$, where the threshold $t_{flip}$ is a tunable parameter for selecting appropriate samples with high similarities to minority class features. Notably, $\texttt{tanh}(\mathcal{Z}[\text{Ind}_{MA}] - t_{flip}) > 0$ when $\mathcal{Z}[\text{Ind}_{MA}] > t_{flip}$, which zeros the flip rates of majority class samples below $t_{flip}$.

Step 4. Flip majority class samples into minority class according to their flip rate $\rho[\text{Ind}_{MA}]$.

Empirically, optimal $t_{flip}$ can be selected by cross-validation. See Section 5 and Appendix for more discussions on practical implementation.

---

**Algorithm 1:** Flip labels in binary classification with noise model $\rho(x)$

---

**Input:** Feature $X$ and labels $Y$, Classifier $C_f$;
**Parameters:** Threshold $t_{flip}$;
$\text{Ind}_{MA} \leftarrow \texttt{index}(Y == 0)$,
$\hat{\eta}, \mathcal{Z}, \rho \leftarrow \texttt{zeroVector}(size = N)$;
**for** $i \in 1 \rightarrow length(Y)$ **do**
  $\quad \hat{\eta}[i] \leftarrow C_f(X[i])$;
**end**
$\mu \leftarrow \texttt{mean}(\hat{\eta}[\text{Ind}_{MA}]), \quad \sigma \leftarrow \texttt{std}(\hat{\eta}[\text{Ind}_{MA}])$,
$\mathcal{Z}[\text{Ind}_{MA}] \leftarrow \frac{\hat{\eta}[\text{Ind}_{MA}] - \mu}{\sigma}$;
$\rho[\text{Ind}_{MA}] \leftarrow max(\texttt{tanh}(\mathcal{Z}[\text{Ind}_{MA}] - t_{flip}), 0)$;
$\mathcal{U} \leftarrow \texttt{random}(0, 1)$;
**if** $\rho[\text{Ind}_{MA}] > \mathcal{U}$ **then**
  $\quad Y[\text{Ind}_{MA}] \leftarrow 1$;
**end**
**Return:** $Y$

---

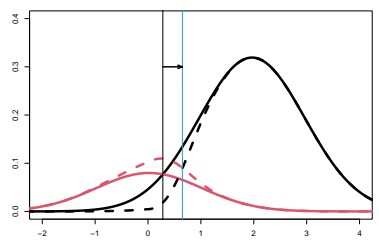

Figure 3: Asymmetric label noise shifts the biased decision boundary towards the majority class region, where the dashed lines are sample distributional densities after label flipping. The majority class samples deep into the minority region have the higher flip-rates $\rho(x)$.

## 4.3 EXTEND TO MULTI-CLASS CLASSIFICATION WITH ONLINE LABEL NOISE UPDATING

Unlike binary classification tasks that contain only one minority class, multi-class data may include multiple minority class samples. To make LNR adopt the various imbalance ratios between

classes, we weigh the flip rate of being flipped into different classes by the class prior weight vector $\theta_{\mathcal{C}}$, where $\mathcal{C}$ is the class vector. These weights are calculated based on the sample sizes $N_{\mathcal{C}}$, and $\max(\theta_{\mathcal{C}} - \theta_{\mathcal{C}}[y], 0)$ guarantees label flipping is only proceeded from majority class to minority class and preventing label flipping between classes with the same class prior. The label-flipping procedure for multi-class scenarios is detailed in Algorithm 2.

Due to that more complex neural networks are typically used in multi-class image classification tasks, training a separate flipping rate estimator is costly, unlike in binary classification scenarios. As our method allows for using any classifier as the flipping rate estimator, we directly utilize the model itself to calculate the posterior probabilities with $\texttt{softmax}(fc(w^t; X, Y))$ during training, where $fc(w^t; X, Y)$ is the output of the last fully-connected layer of model $w^t$ at epoch $t$. Similar to the Deferred Re-Weighting (DRW) approach (Cao et al., 2019), we postpone the introduction of label noise until the model has converged (after $T_D$ epochs) in the fine-tuning stage leveraging possible influence on the feature extraction process.

---

**Algorithm 2:** Constructing label noises during training at round $t$

---

**Input:** Feature $X$ and labels $Y$, and the model $w^t$ at current epoch $t$;
**Parameters:** Flipping threshold $t_{flip}$, Deferred epochs $T_D$;
$\mathrm{Q} \leftarrow \texttt{softmax}(fc(w^t; X, Y))$ where $fc(w^t; \cdot, \cdot)$ is the last fully-connected layer output of model $w^t$;
$\theta_{\mathcal{C}} \leftarrow 1 - \textbf{minMax}(N_{\mathcal{C}})$ where $N_{\mathcal{C}}$ counts the samples of each class;
$\mu \leftarrow \texttt{mean}(\mathrm{Q}), \gamma \leftarrow \texttt{std}(\mathrm{Q})$;
**for** $(x, y^t) \in (X, Y)$ **do**
    Calculate Z-score for sample $x$: $\mathcal{Z}_x \leftarrow \frac{\mathrm{Q}[x] - \mu}{\gamma}$;
    Get the flipping strength for class $c \in \mathcal{C}$: $\theta_{c=y} \leftarrow \max(\theta_{\mathcal{C}} - \theta_{\mathcal{C}}[y], 0)$ ;
    Calculate the class-wize flip-rate for sample $x$: $\mathcal{F}_x \leftarrow \max(\texttt{tanh}(\mathcal{Z}_x - t_{flip}), 0) \times \theta_{c=y}$;
    $\mathcal{U} \leftarrow \texttt{uniform}(\mathcal{F}_x)$;
    **if** $\mathcal{U}$ *contains 1 and* $t > T_D$ **then**
        $y^t \leftarrow \mathcal{C}[\texttt{indexOf}(\max(\mathcal{F}_x[\mathcal{U} == 1]))]$ where $\mathcal{C}$ is the class vector. ;
    **end**
    Update the model $w^t$ with $(x, y^t)$
**end**

---

It is important to note that the posterior probabilities produced by a multi-class model form a $C \times 1$ vector where $C$ is the number of classes. A majority class sample has $C - 1$ posterior probabilities corresponding to being classified as other classes, resulting in $C - 1$ distinct flipping rates. In the case of majority class samples exhibiting similar features to multiple minority classes, resulting in multiple high flip rates, we flip such majority samples into the minority class having the highest flip rate. Random flipping is employed in the event of a tie on flip rates.

## 5 EXPERIMENTAL EVALUATIONS

**Datasets.** *Imbalanced Binary classification:* The experimental comparisons were conducted on 32 datasets from the KEEL repository (Derrac et al., 2015) featuring a wide range of imbalance ratios from 1.82 to 49.6. Summary statistics of these datasets can be found in Table 5 in the Appendix. The criteria for selecting these 32 datasets are detailed in Appendix C.5, aimed at avoiding meaningless comparisons. *Imbalanced Multi-class classification:* We conducted experiments on two multi-class image datasets: CIFAR-10 and CIFAR-100. As the datasets are initially balanced, we artificially create step-wise and long-tailed imbalanced versions of these datasets with the imbalance ratio of $N_{max}/N_{min} = 100$, where $N_{max}$ and $N_{min}$ are the sample size of largest class and smallest class. The step-wise imbalance is created by removing 99% of samples from half of the classes in these datasets. The long-tailed imbalanced dataset is obtained by exponentially reducing samples per class, letting the ratio between the most frequent class and the least frequent class satisfy the imbalance ratio. The detailed information is in the Appendix C.5.

**Classifiers.** As the data-level methods can be coupled with many classifiers, we run the experiments with three commonly used classifiers on KEEL datasets: $k$-nearest neighbor (KNN) (Cover & Hart,

1967), classification and regression tree (CART) (Quinlan, 1986), and multi-layer perceptron (MLP) (Rumelhart & Williams, 1986). For multi-class datasets, the Resnet-32 is used for all methods. The classifier parameters and implement details are in Appendix C.2.

**Baseline and prior methods.** For binary classification, the *baseline* trains the classifier with the original imbalanced data. We compare our method, *LNR*, with well-known resampling method, including over-sampling methods: *SMOTE* (Chawla et al., 2002), *ADASYN* (He et al., 2008), and *Borderline* (Han et al., 2005); and under-sampling methods: *Random-Under-Sampling (RUS)*, *OSS* (Kubat & Matwin, 2000), and *CC* (ClusterCentroids) (Lin et al., 2017). Our method, *LNR*, adapts the final layer of the MLP classifier to include a soft-max function as the flip-rate estimator $C_f$, trained on the original data for KEEL datasets.

For image classification tasks, the baseline is the algorithm-level approach *LDAM-DRW* (Cao et al., 2019), and we compared our method with the data-level SOTA method *RSG* (Wang et al., 2021a) integrating with the baseline. We also show the improvement of our approach combined with a more recent algorithm-level approach *GCL* (Li et al., 2022b). Hyperparameters for all methods were selected via grid search within the ranges specified in Appendix C.2.

**Evaluation metrics.** To ensure a comprehensive evaluation, we report multiple widely used metrics, avoiding those that prioritize a single metric at the expense of others. For the binary classification on KEEL, we used **F1 score**, **G-mean**, and **AUC**. The F1 score and G-mean focus on the performance of classifying minority class samples, while AUC is more conservative, as it is sensitive to the error rate of the majority class samples. For step-wise imbalanced multi-class datasets, we report the average minority class accuracy $\text{Recall}_{\text{MI}}$, the average majority class precision $\text{Prec}_{\text{MA}}$, and the overall accuracy $\text{Acc}_{\text{overall}}$, as we expect to obtain higher recall on minority classes while maintaining high precision on majority classes. To evaluate the performance of long-tailed datasets, we compare their average accuracy of **Many-shot**, **Medium-shot**, and **Few-shot** groups of classes. Appendix C.7 shows the metrics definition and detailed evaluation settings.

## 5.1 RESULTS AND EVALUATIONS

**Comparison on step-wise imbalanced data.** Table 1 reports the comparison of our method with the SOTA data-level method RSG, combined with the LDAM-DRW on CIFAR-10/100 datasets. Our method maintains the highest accuracy for the minority class while also achieving superior precision for the majority class. It consistently outperforms the generative RSG method. Results demonstrate the significant improvement of LNR achieved by combining with multi-stage algorithm-level methods of LDAM-DRW and GCL. By selectively flipping the labels of samples near the decision boundary that share similar features with minority class samples, we minimize the data editing and achieve a more efficient correction of the biased decision boundary, which offers us the highest recalls of minority classes without sacrificing precision on majority classes and overall accuracy.

Table 1: Comparision on Step-wise CIFAR-10/100 Datasets in the format of (mean % ±std).

| | Step-wise Cifar-10 | | | Step-wise Cifar-100 | | |
|---|---|---|---|---|---|---|
| | $\text{Recall}_{\text{MI}}$ | $\text{Prec}_{\text{MA}}$ | $\text{Acc}_{\text{overall}}$ | $\text{Recall}_{\text{MI}}$ | $\text{Prec}_{\text{MA}}$ | $\text{Acc}_{\text{overall}}$ |
| LDAM | 66.41±0.2 | 73.71±0.1 | 77.47±0.06 | 19.80±0.02 | 52.19±0.06 | 45.23±0.03 |
| LDAM-RSG | 67.02±0.07 | 74.15±0.09 | 77.74±0.08 | 21.67±0.04 | 52.87±0.04 | 45.51±0.02 |
| **LDAM-LNR** | **75.06±0.09** | **80.01±0.07** | **78.12±0.03** | **25.84±0.06** | **56.99±0.08** | **45.63±0.02** |
| GCL | 56.78±0.08 | 69.39±0.06 | 74.80±0.04 | 5.48 ±0.03 | 41.39±0.06 | 43.87±0.07 |
| **GCL-LNR** | **72.22±0.05** | **77.71±0.07** | **80.8±0.02** | **26.48±0.03** | **55.59±0.04** | **46.20±0.03** |

**Comparison on Long-tail Cifar-10/100.** Due to the rarity of tail classes, people often have a greater interest in the accuracy of these tail classes (corresponding to few-shot accuracy). Table 2 indicates that our method achieves the highest overall and few-shot accuracy on long-tailed CIFAR-10/100, outperforming other approaches. It is notable that RSG primarily focuses on improving medium-shot accuracy, as it requires a sufficient number of seed samples to generate appropriate new samples. While RSG slightly outperforms in medium-shot accuracy, our LNR approach excels in few-shot and overall accuracy for both Cifar-10 and Cifar-100, demonstrating its strength in sparsely populated tail classes. When combined with GCL, LNR further boosts few-shot and over-

all accuracy, highlighting the remarkable compatibility and simplicity in flipping labels to enhance performance, unlike other data-level approaches that rely on complex sample generation networks.

Table 2: Comparision on Long-tailed CIFAR-10/100 Datasets in the format of (mean % ±std).

| | Long-tailed Cifar-10 | | | | Long-tailed Cifar-100 | | | |
|---|---|---|---|---|---|---|---|---|
| | **Many-shot** | **Medium-shot** | **Few-shot** | **Overall** | **Many-shot** | **Medium-shot** | **Few-shot** | **Overall** |
| LDAM | **82.62±0.06** | 76.12±0.1 | 75.01±0.1 | 78.39±0.03 | **62.21±0.05** | 43.28±0.08 | 20.83±0.03 | 42.98±0.03 |
| LDAM-RSG | 81.56±0.15 | **77.03±0.1** | 77.30±0.1 | 78.93±0.02 | 60.46±0.05 | **43.88±0.1** | 22.57±0.09 | 43.08±0.07 |
| **LDAM-LNR** | 81.17±0.08 | 76.42±0.01 | **79.83±0.1** | **79.34±0.01** | 61.04±0.03 | 43.36±0.02 | **24.11±0.04** | **43.58±0.02** |
| GCL | **88.60±0.04** | **79.57±0.01** | 70.08±0.2 | 80.55±0.03 | **67.16±0.03** | 46.63±0.06 | 13.57±0.06 | 43.90±0.03 |
| **GCL-LNR** | 88.20±0.04 | 79.50±0.07 | **77.60±0.2** | **82.41±0.03** | 57.11±0.07 | **51.38±0.07** | **25.02±0.09** | **45.48±0.02** |

**Evaluation on KEEL results.** The relative rankings shown in Figure 4 demonstrate that our method achieved the highest average performance across 32 datasets for all three classifiers in both F1 score and G-mean. In addition to these rankings, we calculated the Pearson correlation coefficients between the relative performance differences of our method and other methods, and the data distribution characteristics. The numeric results are detailed in Table 9 in Appendix C.7. Our method demonstrated superior F1 score and G-mean performance on data with higher class overlap, attributed to reduced class overlap through label flipping at decision boundaries. In highly imbalanced data settings, oversampling methods require synthesizing oversized samples to balance the data, which exacerbates generative errors and increases the error rate for majority class samples, thereby hampering AUC performance. On the contrary, LNR does not involve adding or removing samples, significantly enhancing F1 score and G-mean scores without compromising AUC performance.

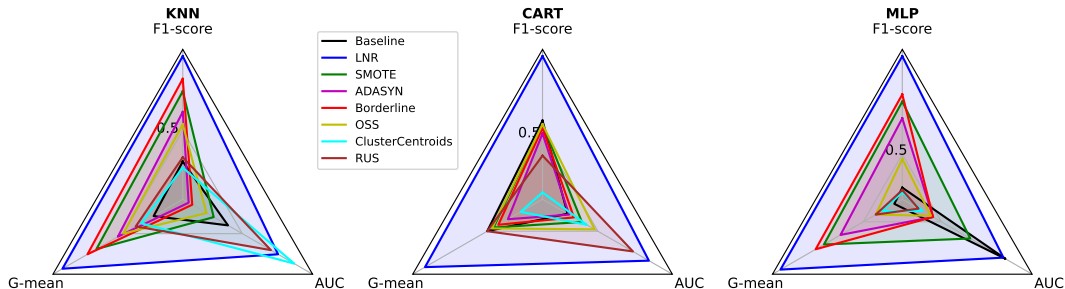

Figure 4: Relative ranking of methods from 1 (the best) to 0 (the lowest) on 32 KEEL datasets.

## 5.2 SENSITIVITY ANALYSIS

With an appropriate label noise model, LNR requires an adequate $t_{flip}$ to introduce label noises. When the label flip threshold $t_{flip}$ is too low, we observe poorer performance on MLP than the baseline, indicating that a low $t_{flip}$ leads to excessive noise being introduced. When $t_{flip}$ is too high, only a few samples are flipped until the performance gain approaches zero. The full sensitivity analysis and figures are detailed in Appendix C.3.

## 6 CONCLUSION

In this paper, we theoretically analyze the impact mechanisms of imbalanced data on classifiers. By investigating the influence of label noise on the data, we propose a novel method of artificially introducing label noise to enhance the performance of imbalanced learning. Traditional resampling algorithms suffer from information loss and the introduction of generative noise. In contrast, our LNR method balances classes by simply flipping the labels of majority class samples that avoid these issues. Through testing on simulated and real-world data, our method not only demonstrates superiority in the recognition performance of minority class samples but also shows comprehensive improvements across multiple metrics. In addition, our LNR approach can integrate seamlessly with any classifiers and other algorithm-level methods for imbalanced learning.

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

# A PROOFS

## A.1 PROOF OF LEMMA 1

*Proof.* By Bayes theorem,

$$\eta(x) = \Pr(Y = 1 \mid X = x) = \frac{\pi_1 P_1(x)}{\pi_1 P_1(x) + \pi_0 P_0(x)}.$$

Thus, $\eta(x) = 0.5$ is equivalent to $\frac{P_1(x)}{P_0(x)} = \frac{\pi_0}{\pi_1}$. $\qquad\square$

For Gaussian data with $P_0 \sim N(\mu_0, \Sigma)$ and $P_1 \sim N(\mu_1, \Sigma)$, $\frac{P_1(x)}{P_0(x)} = \frac{\pi_0}{\pi_1}$ is equivalent to

$$\frac{\exp\left\{-\frac{1}{2}(x - \mu_1)^T \Sigma^{-1}(x - \mu_1)\right\}}{\exp\left\{-\frac{1}{2}(x - \mu_0)^T \Sigma^{-1}(x - \mu_0)\right\}} = \frac{\pi_0}{\pi_1},$$

and this can be simplified to

$$\omega^T x + \beta = \ln\left(\frac{\pi_0}{\pi_1}\right)$$

with $\omega = \Sigma^{-1}(\mu_1 - \mu_0)$ and $\beta = \frac{1}{2}\mu_0^T \Sigma^{-1}\mu_0 - \frac{1}{2}\mu_1^T \Sigma^{-1}\mu_1$.

## A.2 PROOF OF LEMMA 2

*Proof.* The population version of the F1 score is defined as

$$\mathcal{F}1(x_0) = \frac{2\pi_1 \mathcal{P}_1(x_0)}{\pi_1 \mathcal{P}_1(x_0) + \pi_0 \mathcal{P}_0(x_0) + \pi_1}. \tag{3}$$

where $\mathcal{P}_1(x_0) = \Pr(\omega^T(X - x_0) > 0 \mid Y = 1)$, and $\mathcal{P}_0(x_0) = \Pr(\omega^T(X - x_0) > 0 \mid Y = 0)$.

Here, we denote the numerator as $A = 2\pi_1 \mathcal{P}_1(x_0)$ and the denominator as $B = \pi_1 \mathcal{P}_1(x_0) + \pi_0 \mathcal{P}_0(x_0) + \pi_1$. The $\operatorname{argmax}_{x_0 \in \mathcal{X}} \mathcal{F}1(x_0)$ can be found by letting $\frac{\partial \mathcal{F}1(x_0)}{\partial x_0} = 0$, that is,

$$\frac{\partial \mathcal{F}1(x_0)}{\partial x_0} = \frac{A'B - B'A}{B^2} = 0$$

where $A' = \frac{\partial A}{\partial x_0}$ and $B' = \frac{\partial B}{\partial x_0}$. As $B > 0$, we have

$$A'B = B'A. \tag{4}$$

Assume exchangeability of integration and differentiation, the Leibniz integral rule implies $\partial \mathcal{P}_1(x_0)/\partial x_0 = -P_1(x_0)\omega$ and $\partial \mathcal{P}_0(x_0)/\partial x_0 = -P_0(x_0)\omega$. By substituting

$$A' = -2\pi_1 P_1(x_0)\omega, \ B' = -\pi_1 P_1(x_0)\omega - \pi_0 P_0(x_0)\omega$$

into equation 4, we obtain that

$$-2\pi_1 P_1(x_0) \left(\pi_1 \mathcal{P}_1(x_0) + \pi_0 \mathcal{P}_0(x_0) + \pi_1\right)\omega = 2\pi_1 \mathcal{P}_1(x_0) \left(-\pi_1 P_1(x_0)\omega - \pi_0 P_0(x_0)\omega\right),$$

which is equivalent to

$$\frac{2\pi_1 P_1(x_0)}{\pi_1 P_1(x_0) + \pi_0 P_0(x_0)} = \mathcal{F}1(x_0). \tag{5}$$

Rearranging the equation results in

$$\frac{P_1(x_0)}{P_0(x_0)} = \frac{\mathcal{F}1(x_0)}{2 - \mathcal{F}1(x_0)} \frac{\pi_0}{\pi_1},$$

which indicates that the decision boundary that maximizes the population F1 score is of the form

$$S^{\mathrm{F1}} = \left\{ x_0 \in \mathcal{X} : \frac{P_1(x_0)}{P_0(x_0)} = \frac{\mathcal{F}1(x_0)}{2 - \mathcal{F}1(x_0)} \frac{\pi_0}{\pi_1} \right\}.$$

$\qquad\square$

### A.3 PROOF OF LEMMA 3

*Proof.* First,

$$
\begin{aligned}
\eta^*(x) & = \Pr(Y^* = 1 \mid X = x, Y = 1) \Pr(Y = 1 \mid X = x) \\
& \quad + \Pr(Y^* = 1 \mid X = x, Y = 0) \Pr(Y = 0 \mid X = x) \\
& = \eta(x) + \rho(x)[1 - \eta(x)] \\
& = \eta(x)[1 - \rho(x)] + \rho(x).
\end{aligned}
$$

As $x^*$ satisfies $\eta^*(x^*) = 0.5$ with the constrain of $\rho(\hat{x}^*) < 0.5$, we have $\eta(x^*)[1 - \rho(x^*)] + \rho(x^*) = 0.5$, which is equivalent to

$$
\frac{\pi_1 P_1(x^*)}{\pi_1 P_1(x^*) + \pi_0 P_0(x^*)} [1 - \rho(x^*)] + \rho(x^*) = \frac{1}{2}.
$$

It follows that

$$
\frac{P_1(x^*)}{P_0(x^*)} = [1 - 2\rho(x^*)] \frac{\pi_0}{\pi_1}.
$$

Thus, the optimal Bayesian decision boundary that maximizes the accuracy on the noisy dataset $(X, Y^*)$ is defined as:

$$
S^* = \left\{ x^* \in \mathcal{X} : \frac{P_1(x^*)}{P_0(x^*)} = [1 - 2\rho(x^*)] \frac{\pi_0}{\pi_1} \right\}.
$$

$\square$

For one-dimensional normal distributed data with $P_1 \sim N(\mu_1, \sigma^2)$ and $P_0 \sim N(\mu_0, \sigma^2)$, we further take the natural logarithm of both sides to specialize further $x^*$ with the terms of $(\mu_1, \mu_0, \sigma, \pi_1, \pi_0)$:

$$
\ln\left(\frac{\pi_1}{\pi_0}\right) - \frac{(x^* - \mu_1)^2}{2\sigma^2} + \frac{(x^* - \mu_0)^2}{2\sigma^2} = \ln(1 - 2\rho(\hat{x}^*)).
$$

After rearranging, we have the optimal Bayesian decision boundary that maximizes the accuracy on noisy dataset $(X, Y^*)$ as

$$
x^* = \frac{1}{2}(\mu_0 + \mu_1) + \ln\left(\frac{\pi_0}{\pi_1}\right) \frac{\sigma^2}{\mu_1 - \mu_0} + \ln(1 - 2\rho(x^*)) \frac{\sigma^2}{\mu_1 - \mu_0}.
$$

For the multivariate normal distribution with equal covariance $\Sigma$, the decision boundary has the form:

$$
2(\mu_1 - \mu_0)^T \Sigma^{-1} x^* = \mu_1^T \Sigma^{-1} \mu_1 - \mu_0^T \Sigma^{-1} \mu_0 + 2\ln\left(\frac{\pi_0}{\pi_1}\right) + 2\ln(1 - 2\rho(x^*)).
$$

## B IMPACT OF IMBALANCE RATIO ON DECISION BOUNDARY OF KNN CLASSIFIER

In this section, we investigate the impact of the imbalance ratio on the $k$-nearest neighbors (KNN) algorithm, which is a widely used classification method due to its non-parametric nature.

The KNN classifies a new sample based on the majority voting of its $k$ nearest neighbors in the training dataset. Denote $\mathcal{X}_{train}$ and $\mathcal{X}_{test}$ as the training and testing datasets, respectively. For each instance $x_{test} \in \mathcal{X}_{test}$, let $\mathcal{X}_{nn}(x_{test}; k) = \{x^{(1)}, \ldots, x^{(k)}\} \subset \mathcal{X}_{train}$ with corresponding labels $\{Y^{(1)}, \ldots, Y^{(k)}\}$ be its $k$-nearest neighbors with ascending Euclidean distances. Then, the KNN classifies $x_{test}$ to the positive class if $\frac{1}{k} \sum_{i=1}^{k} \mathbb{I}(Y^{(i)} = 1) \geq 0.5$.

We introduce the following setup to study the impact of the imbalance ratio on the KNN classifier. Denote the open Euclidean ball centered at $x \in \mathcal{X}_{test}$ with the radius $r_k(x) = \|x - x^{(k)}\|$ as $B_k(x) = \{x \in \mathcal{X} : \|x - x^{(k)}\| \leq r_k(x)\}$, where $x^{(k)} \in \mathcal{X}_{train}$ is the $k$-th nearest neighbor of $x$ and $\|\cdot\|$ is the Euclidean distance. Then, the decision boundary of the KNN is defined as $\{x_{knn} \in \mathcal{X} : \Pr(x \in B_k(x_{knn}), Y = 1) = \Pr(x \in B_k(x_{knn}), Y = 0)\}$.

**Lemma 4.** *The decision boundary of the KNN is*

$$S^{knn} = \left\{ x_{knn} \in \mathcal{X} : \frac{\int_{B_k(x_{knn})} P_1(x)dx}{\int_{B_k(x_{knn})} P_0(x)dx} = \frac{\pi_0}{\pi_1} \right\}.$$

*Proof.* The proof of this lemma follows immediately from the facts that $\Pr(x \in B_k(x_{knn}), Y = 1) = \int_{B_k(x_{knn})} P_1(x)dx$ and $\Pr(x \in B_k(x_{knn}), Y = 0) = \pi_0 \int_{B_k(x_{knn})} P_0(x)dx$. □

When the size of the training dataset $n_{train} \to \infty$, $B_k(x_{knn})$ degenerates to the point $x_{knn}$ and $\frac{\int_{B_k(x_{knn})} P_1(x)dx}{\int_{B_k(x_{knn})} P_0(x)dx}$ vanishes to $\frac{P_1(x_{knn})}{P_0(x_{knn})}$. In this case, the decision boundary of the KNN coincides with the Bayesian decision boundary, so the imbalance ratio has the same impact on the decision boundary of the KNN as for the Bayesian decision boundary. In the case of a finite training size, the KNN decision boundary may deviate from the Bayesian decision boundary as demonstrated in the following example.

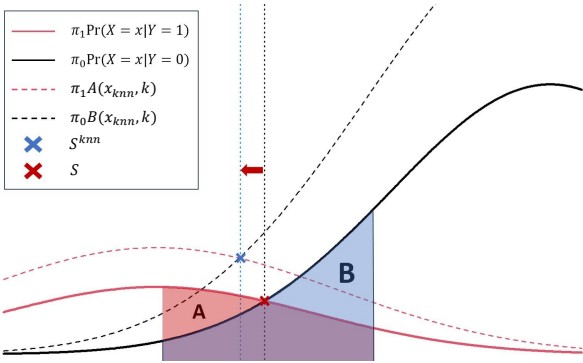

Figure 5: KNN biases the decision boundary from the optimal Bayesian decision boundary towards the minority class region when data is imbalanced. Area A and area B are calculated by assuming $r_k(x^*) = 1$ and substituting $x = x^*$ and $r_k(x^*) = 1$ into $A_1(x, k)$ and $A_0(x, k)$ respectively.

**Example 3.** *For one-dimensional data, $B_k(x_{knn})$ is an interval on the x-axis of length $2r_k(x_{knn})$. The theoretical decision boundary produced by the KNN classifier has the following form*

$$S^{knn} = \left\{ x_{knn} \in \mathcal{X} : \frac{A_1(x_{knn}, r)}{A_0(x_{knn}, r)} = \frac{\pi_0}{\pi_1} \right\}. \tag{6}$$

*where $A_1(x_{knn}, r) = \int_{x-r_k(x_{knn})}^{x+r_k(x_{knn})} P_1(x)dx$, and $A_0(x_{knn}, r) = \int_{x-r_k(x_{knn})}^{x+r_k(x_{knn})} P_0(x)dx$ represent the areas under the class-conditional probability density curves.*

*Let $P_1 \sim N(0, 1)$, $P_0 \sim N(2.5, 1)$ with $\pi_1 = 0.2$ and $\pi_0 = 0.8$ as considered in Figure 5, the Bayesian decision boundary is $S = x^* \in \mathcal{X} : x^* = 0.6955$ (the red cross). With $r_k(x^*) = 1$, we obtain area A: $\pi_1 A_1(0.6955, k) = 0.1147$ and area B: $\pi_0 A_0(0.6955, k) = 0.1674$, thus $A < B$ indicating that the decision boundary of KNN biases from the Bayesian decision boundary towards the minority class.*

*As the KNN decision boundary $S^{knn}$ does not have an explicit expression due to $r_k(x^*)$ being dependent on both $x^*$ and $k$, we numerically demonstrate the relation between $r_k(x^*)$ and the deviation of $x^*_{knn}$ and $x^*$ in Figure 8. From Figure 8, it is evident that the decision boundary $x^*_{knn}$ of KNN deviates from the optimal decision boundary $x^*$ as $r_k(x^*)$ increases on imbalanced data, while it consistently aligns with $x^*$ on balanced data for this example.*

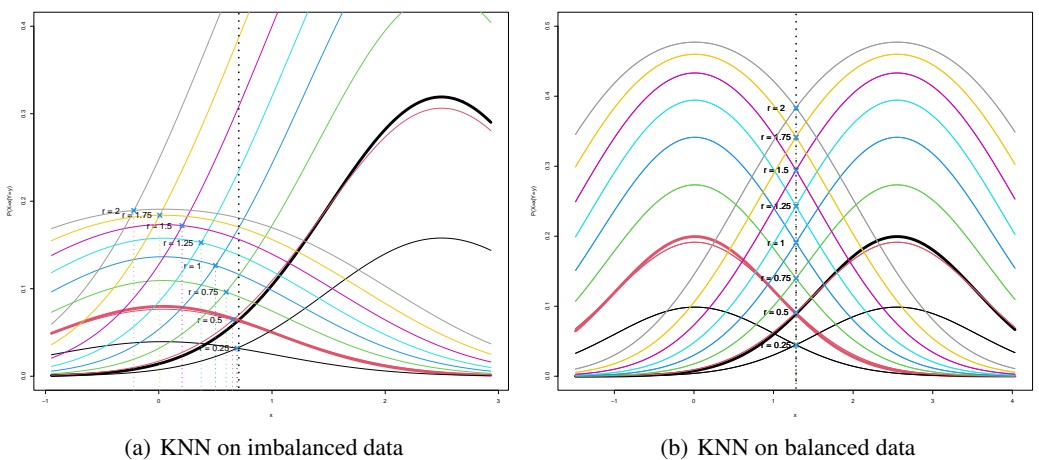

(a) KNN on imbalanced data        (b) KNN on balanced data

Figure 6: The thin curves in different colors represent $\pi_1 A(x, r_k)$ and $\pi_0 B(x, r_k)$ with different values of $r_k = r$. Their intersections (blue cross) are the decision boundaries produced by KNN with the corresponding $r$ value, and the optimal Bayesian decision boundary $x = x^*$ is shown with the vertical dashed black line.

The Bayesian decision boundary tends to be biased towards the minority class region in comparison to the optimal F1 score decision boundary, thereby contributing to a decrease in F1 score performance. Additionally, the decision boundary fitted by the KNN classifier exhibits a continued encroachment upon the minority class region, resulting in compromised accuracy and F1 score performance.

## C  EXPERIMENT DETAILS

### C.1  COMPUTE RESOURCES

**Hardware information.**    **Type of CPU:** One Intel(R) Core(TM) i7-10700K CPU @ 3.80GHz with 16 cores. **Type of GPU:** One NVIDIA GeForce RTX 4080 SUPER with memory of 16GB.

**Computational workload.**    The experiments on 8 synthetic datasets are repeated 30 rounds using 3 different classifiers with 8 methods. The total amount of training is $8 \times 30 \times 3 \times 8 = 5760$ for experiments on synthetic data. The experiment conducted on CIFAR-10/100 involves 5 methods in 2 different imbalanced settings. The results are collected from 10 rounds of training with 100 epochs. The total epochs of training is $5 \times 2 \times 10 \times 100 = 10000$. The experiments on 32 KEEL datasets are repeated in 100 rounds. The total amount of training is $8 \times 100 \times 3 \times 8 = 19200$.

Table 3: Hyperparameters of methods and classifiers

| Classifiers | parameter | value | |
|---|---|---|---|
| KNN | $k$ | [5,10] | |
| MLP | Hidden layer shape | 5x10x5 | |
| | Maximum training epochs | 800 on synthetic data, 2000 on KEEL data | |
| | Optimizer | Adam | |
| | Learning rate | 0.001 | |
| Methods | parameter | tuning range | steps |
| LNR | threshold $t_{flip}$ | [0,4] | 10 |
| | training epoch $e$ | [100,4000] | 3 |
| KNN-based resampling | $k$ | $[1,\min(20, N_{minority})]$ | 3 |
| OSS | $n_{oss}$ | [2,30] | 10 |
| All resampling methods | re-balance ratio $\gamma$ | $[\frac{\pi_1}{\pi_0}, 1]$ | 10 |

## C.2 METHOD PARAMETERS

The hyperparameters of classifiers are listed in Table 3. Re-sampling methods are compared and implemented using the "imbalanced-learn" package [1] in Python. Classifers KNN, CART, and MLP are implemented with the Python "scikit-learn" package [2], and parameters not listed are set to their default values. The Resnet-32 is implemented with Pytorch.

The parameters are selected within the tuning range in Table 3 by repeating 3 times of 4-fold cross-validation on reshuffled data with F1 score. The KNN-based resampling methods, including SMOTE, ADASYN, Borderline-SMOTE, and CC (ClusterCentroids), require the user to preset the value $k$ to synthetic or rank the samples.

We introduce deferred label noise after training epoch $T_D$ for multi-class image classification. We set $T_D = 80$ and $t_{flip} = 3.5$ for both imbalance settings on CIFAR-10/100 datasets. For all experiments, the Resnet-32 is trained up to 100 epochs using SDG (stochastic gradient descend) optimizer with a momentum of 0.9 and a learning rate of 0.1 for the first 60 epochs, 0.01 for the epochs 60-80, and 0.0001 after the 80 epochs.

## C.3 SENSITIVITY ANALYSIS

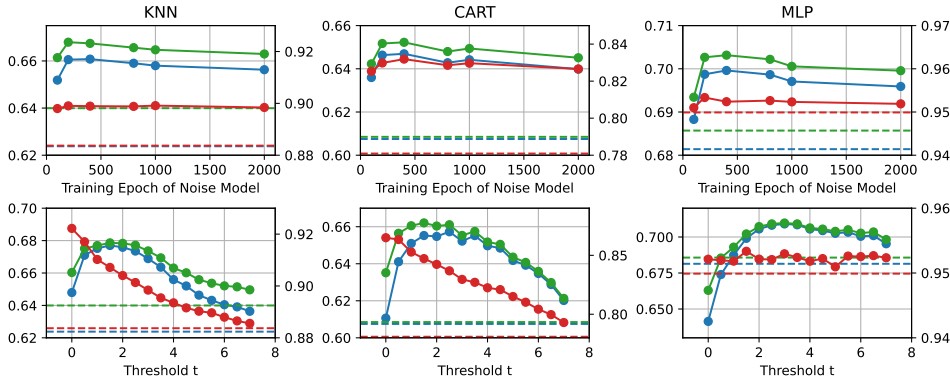

Figure 7: **Sensitivity Analysis.** The horizontal dashed lines represent the baseline performance in terms of F1 (blue), G-mean (green), and AUC (red). The right y-axis corresponds to the AUC values. The solid line denotes the performance of LNR. **Top:** Minimal impact of noise model training epochs on performance gains, with LNR consistently outperforming the baseline. When the number of training epochs is too low (not converged) or too high (overfitting), performance slightly declines. **Bottom:** When the label flip threshold $t$ is too low, we observe poorer performance on MLP compared to the baseline, indicating that a low $t$ leads to excessive noise being introduced. When $t$ is too high, only a few samples are flipped until the performance gain approaches zero. For KNN and CART, the AUC gradually decreases as $t$ increases but not sensitive to MLP.

Sensitivity analysis for each hyperparameter was conducted using simulated data while keeping other parameters constant. In the simulations on synthetic data, the minority class prior was set to 0.1, with a sample size of 500, a class mean distance of 2, and a dimensionality of 5. The results consistently showed that the performance gains from our LNR approach decrease if the threshold is either too low or too high. For KNN and CART, the classifiers always outperformed the baseline regardless of $t_{flip}$. However, for MLP, if $t_{flip}$ is too low, excessive label noise can cause performance to fall below baseline levels, while if is too high, performance can degrade to baseline levels.

The impact of the epoch parameter on performance is less sensitive. We observed that if the noise model is trained for too few epochs, the benefits of flipped labels are reduced. However, once the model converges, further training does not significantly enhance performance, as the model possesses enough capability to filter out detrimental label noise. As LNR is data-driven, the parameters can be optimized using standard cross-validation techniques like other data-level methods.

---

[1] https://github.com/scikit-learn-contrib/imbalanced-learn.git
[2] https://github.com/scikit-learn/scikit-learn

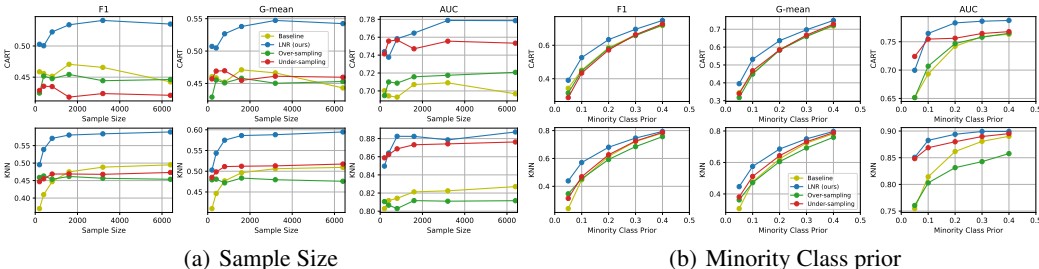

(a) Sample Size         (b) Minority Class prior

Figure 8: **The impact of data characteristics on performance**. For clarity, the average performance of over-sampling and under-sampling methods is reported. **a)** Our method demonstrates performance gains comparable to those of resampling methods when the sample size is 200. However, our method consistently outperforms the others as the sample size increases. **b)** As the data approaches a more balanced state, all methods exhibit similar performance; notably, our method's AUC shows improvement even with more balanced data. The performance gains of our method in terms of F1 and G-mean become more evident as the prior of the minority class decreases.

## C.4 CLASS FAIRNESS OF LNR ON LONG-TAILED DATA

For multi-classification problems, LNR significantly improves the performance in terms of overall accuracy and few-shot accuracy while maintaining class fairness. LNR flip samples are based on the similarity to the minority class, making these flips regulated and self-limiting. This prevents the model from having an unintended bias towards specific classes. To clearly demonstrate the harmfulness of LNR towards class fairness, we compared the average confusion matrices of GCL and GCL-LNR.

Table 4: Conn fusion matrix of GCL-LNR. The differences (higher than 10) compared to the confusion matrix of GCL are reported in the brackets. The horizontal axis represents the predicted value, whereas the vertical axis represents the true value.

| GCL-LNR (Differ) | 0 | 1 | 2 | 3 | 4 | 5 | 6 | 7 | 8 | 9 |
|---|---|---|---|---|---|---|---|---|---|---|
| 0 | **908 (-26)** | 10 | 24 | 8 | 4 | 2 | 2 | 4 | 20 (+13) | 19 (+16) |
| 1 | 6 | **978 (-4)** | 0 | 0 | 1 | 0 | 1 | 0 | 0 | 13 |
| 2 | 33 | 3 | **860 (-4)** | 28 | 20 | 21 | 23 | 8 | 0 | 4 |
| 3 | 23 | 4 | 50 | **782 (+18)** | 25 | 72 (-13) | 22 | 13 | 2 | 8 |
| 4 | 12 | 2 | 54 | 39 | **829 (+3)** | 16 | 18 | 25 | 2 | 2 |
| 5 | 15 | 2 | 42 | 140 | 26 | **736 (-1)** | 13 | 24 | 1 | 2 |
| 6 | 11 | 6 | 58 | 68 | 13 | 14 | **820 (-4)** | 8 | 1 | 1 |
| 7 | 22 | 3 | 33 | 50 (-20) | 48 | 50 | 6 | **778 (+22)** | 2 | 8 |
| 8 | 112 (-91) | 52 | 22 | 11 | 2 | 4 | 4 | 2 | **758 (+80)** | 33 (+20) |
| 9 | 30 (-42) | 148 (-48) | 4 | 8 | 2 | 0 | 3 | 4 | 10 | **792 (+102)** |

The bold numbers on the diagonal demonstrate that LNR achieves a more effective trade-off between the classification performance of the head class and the tail class, using only a small number of head class samples (reduced 16 true positives in head classes [0,1,2,3]) to achieve a significant

improvement in tail class performance (increasing 204 true positives in tail classes [7,8,9]). To provide a straightforward understanding of how LNR achieves the better decision boundary on the long-tailed CIFAR-10, we present the statistics of flipped labels below, where "→" means "flipped to":

{0: [→7: 2, →8: 41, →9: 44], 1:[→9: 6], 2: [], 3: [], 4: [], 5: [], 6: [], 7: [], 8: [], 9: []}

The long-tailed CIFAR-10 before and after flipping labels by LNR:

**Before**: 0: 5000, 1: 2997, 2: 1796, 3: 1077, 4: 645, 5: 387, 6: 232, 7: 139, 8: 83, 9: 50

**After**: 0: 4913, 1: 2991, 2: 1796, 3: 1077, 4: 645, 5: 387, 6: 232, 7: 141, 8: 124, 9: 100

These beneficial label noises effectively correct the classifier's decision boundaries with minimal data editing.

## C.5 EXPERIMENT DATASETS

**Synthetic data:** The simulations are conducted on the $d$-dimensional Gaussian distribution $N(\mu_r, \Sigma)$, with the mean of $\mu_r$ and the covariance matrix $\Sigma$ being the $d \times d$ identity matrix $\mathcal{I}_d$. We generate the simulated data with the setting: let $\Pr(Y = 1) = \pi_1 \in [0, 1]$ and $X \mid Y = r \sim N(\mu_r, \Sigma)$, where $\mu_1 = (\mu/2, 0, \ldots, 0)^T = -\mu_0 \in \mathbb{R}^d$. There are $2 \times 2 \times 2 = 8$ different distributions generated with different statistical parameters combinations of $|\mu_1 - \mu_0| \in \{1, 2\}$, $d \in \{5, 25\}$, and $\pi_1 \in \{0.1, 0.3\}$. The maximum training epoch of the MLP classifier is set to 800 as MLP converges faster on synthetic datasets. Due to the limited performance on high dimensional synthetic data with the constrained structure of the MLP classifier, we instead use the KNN classifier with $k = 50$ as the flip-rate estimator $C_f$ for LNR on $d = 25$ synthetic datasets.

**KEEL dataset:** The KEEL repository (Derrac et al., 2015) provides us with 99 binary imbalanced datasets. We select 32 datasets with the following criteria to avoid meaningless and inefficient comparisons on the datasets:

a) Baseline already achieves an F1-score over 90%, or all methods have an F1-score lower than 10%.

b) Datasets with insufficient minority class sample size in the test dataset (less than 10).

The imbalance ratios $\pi_0/\pi_1$ in the selected 32 datasets widely ranged from 1.82 to 49.6, providing a comprehensive comparison. We simply removed all categorical attributes to simplify data preprocessing and standardized the remaining numerical dimensions.

## C.6 KEEL DATASET CHARACTERISTICS

As we discussed in Section 3.2, the degree of decision boundary deviation is affected by both the imbalanced ratio and the class overlap. The indicator of the imbalance ratio is less insightful as the degree of class overlap may vary largely on datasets of the same imbalance ratio. To study the methods' strengths and weaknesses, we summarise the characteristics of the dataset based on the categories proposed by Napierala & Stefanowski (2016), which suggests assigning each minority class sample to one of the following groups based on the number of minority class neighbors within the 5-nearest-neighbors: **Safe:** $4+$, **Border:** $2-3$, **Rare:** 1, **Outlier:** 0. The proportion of each group describes the distribution of minorities in the imbalanced dataset, especially Border and Rare reflecting the extent of class overlap, as a complementary metric to the imbalanced ratio.

## C.7 EVALUATION SETTINGS AND RESULTS

**Settings.** We employed various seeds to generate 30 sets of synthetic training and testing datasets, with the sample size $N_{train} = 500$ and $N_{test} = 2000$ for all 8 distributions. Due to the limited sample size for the KEEL datasets, we use different seeds to partition the data into 100 sets of training and testing datasets at a ratio of 7:3, ensuring at least 10 minority class samples in each testing set. With 30 rounds on synthetic data and 100 rounds on KEEL datasets, we obtain statistically significant results, and the mean and the standard deviation are reported in Appendix C.7.

Table 5: KEEL(Derrac et al., 2015) Data statistics and characteristics: $d$ is the number of dimensions, $N_1$ and $N_0$ denote the number of minority and majority class samples.

| index | Dataset Name | $d$ | $(N, N_1, N_0)$ | IR: $\pi_0/\pi_1$ | (S,B,R,O) |
|---|---|---|---|---|---|
| | statistics | | | | |
| 1 | *abalone-17_vs_7-8-9-10* | 7 | (2338, 58, 2280) | 39.31 | (0.02, 0.22, 0.33, 0.43) |
| 2 | *abalone-19_vs_10-11-12-13* | 7 | (1622, 32, 1590) | 49.69 | (0.0, 0.0, 0.28, 0.72) |
| 3 | *abalone9-18* | 7 | (731, 42, 689) | 16.4 | (0.02, 0.4, 0.17, 0.4) |
| 4 | *ecoli1* | 7 | (336, 77, 259) | 3.36 | (0.53, 0.32, 0.08, 0.06) |
| 5 | *ecoli2* | 7 | (336, 52, 284) | 5.46 | (0.77, 0.15, 0.0, 0.08) |
| 6 | *ecoli3* | 7 | (336, 35, 301) | 8.6 | (0.29, 0.49, 0.09, 0.14) |
| 7 | *flare-F* | 9 | (1066, 43, 1023) | 23.79 | (0.05, 0.26, 0.4, 0.3) |
| 8 | *glass-0-1-2-3_vs_4-5-6* | 9 | (214, 51, 163) | 3.2 | (0.67, 0.22, 0.06, 8.06) |
| 9 | *glass0* | 9 | (214, 70, 144) | 2.06 | (0.56, 0.36, 0.03, 0.06) |
| 10 | *glass1* | 9 | (214, 76, 138) | 1.82 | (0.46, 0.3, 0.16, 0.08) |
| 11 | *haberman* | 3 | (306, 81, 225) | 2.78 | (0.05, 0.47, 0.32, 0.16) |
| 12 | *kr-vs-k-three_vs_eleven* | 3 | (2935, 81, 2854) | 35.23 | (0.62, 0.22, 0.09, 0.07) |
| 13 | *kr-vs-k-zero-one_vs_draw* | 3 | (2901, 105, 2796) | 26.63 | (0.65, 0.23, 0.06, 0.07) |
| 14 | *led7digit-0-2-4-5-6-7-8-9_vs_1* | 7 | (443, 37, 406) | 10.97 | (0.19, 0.65, 0.14, 0.03) |
| 15 | *newthyroid2* | 5 | (215, 35, 180) | 5.14 | (0.69, 0.31, 0.0, 0.0) |
| 16 | *page-blocks0* | 10 | (5472, 559, 4913) | 8.79 | (0.7, 0.17, 0.07, 0.06) |
| 17 | *pima* | 8 | (768, 268, 500) | 1.87 | (0.3, 0.45, 0.17, 0.09) |
| 18 | *vehicle1* | 18 | (846, 217, 629) | 2.9 | (0.24, 0.57, 0.15, 0.05) |
| 19 | *vehicle3* | 18 | (846, 212, 634) | 2.99 | (0.16, 0.51, 0.26, 0.06) |
| 20 | *yeast-0-2-5-6_vs_3-7-8-9* | 8 | (1004, 99, 905) | 9.14 | (0.35, 0.3, 0.15, 0.19) |
| 21 | *yeast-0-2-5-7-9_vs_3-6-8* | 8 | (1004, 99, 905) | 9.14 | (0.68, 0.17, 0.05, 0.1) |
| 22 | *yeast-0-3-5-9_vs_7-8* | 8 | (506, 50, 456) | 9.12 | (0.16, 0.28, 0.22, 0.34) |
| 23 | *yeast-0-5-6-7-9_vs_4* | 8 | (528, 51, 477) | 9.35 | (0.08, 0.43, 0.18, 0.31) |
| 24 | *yeast-1-2-8-9_vs_7* | 8 | (947, 30, 917) | 30.57 | (0.0, 0.27, 0.23, 0.5) |
| 25 | *yeast-1-4-5-8_vs_7* | 8 | (693, 30, 663) | 22.1 | (0.0, 0.07, 0.43, 0.5) |
| 26 | *yeast-1_vs_7* | 7 | (459, 30, 429) | 14.3 | (0.07, 0.33, 0.27, 0.33) |
| 27 | *yeast-2_vs_4* | 8 | (514, 51, 463) | 9.08 | (0.55, 0.22, 0.06, 0.18) |
| 28 | *yeast1* | 8 | (1484, 429, 1055) | 2.46 | (0.21, 0.45, 0.22, 0.11) |
| 29 | *yeast3* | 8 | (1484, 163, 1321) | 8.1 | (0.56, 0.26, 0.08, 0.1) |
| 30 | *yeast4* | 8 | (1484, 51, 1433) | 28.1 | (0.06, 0.35, 0.2, 0.39) |
| 31 | *yeast5* | 8 | (1484, 44, 1440) | 32.73 | (0.34, 0.5, 0.11, 0.05) |
| 32 | *yeast6* | 8 | (1484, 35, 1449) | 41.4 | (0.37, 0.23, 0.11, 0.29) |

**Metrics.** The performance on binary classification is evaluated in three different metrics: F1 score, G-mean (Geometric Mean), and AUC (Area under the Curve). Different to the definition of F1, which mainly focuses on the true positives, G-mean also considers the true negatives with the following definition:

$$\text{G-mean} = \sqrt{\frac{\text{TP}}{\text{TP} + \text{FN}} \times \frac{\text{TN}}{\text{TN} + \text{FP}}}.$$

Compared to accuracy, AUC evaluates the classification performance of a classifier by calculating the area under the ROC (Receiver Operating Characteristic) curve. The ROC curve takes into account both recall (TPR) and FPR (False Positive Rate), defined as:

$$\text{AUC} = \int_0^1 \text{TPR}(\text{FPR}^{-1}(t))dt$$

where $\text{FPR} = \text{FP}/(\text{FP} + \text{TN})$. A classifier that correctly classifies positive samples as positive while maintaining a low error rate for negative samples will have a higher AUC. Unlike F1 and G-mean, which are more sensitive to the prediction performance of minority class samples, AUC is a more conservative metric for comprehensively reflecting the performance of a classifier. Methods that significantly increase the number of minority class samples to improve F1 score or G-mean often come at the cost of higher errors in negative samples (FP). Therefore, we desire a re-balancing method that should consider these three metrics comprehensively.

For multi-class image classification, we use different metrics to evaluate the performance on step-wise and long-tailed datasets. On the step-wise CIFAR-10/100, due to half of classes are removed 99% of samples forms 5 and 50 minority classes, we evaluated model by the average accuracy of minority classes denoted as $\text{Recall}_{\text{MI}}$ and the average precision of majority classes denoted as $\text{Preci}_{\text{MA}}$, as well as the overall accuracy. For long-tailed CIFAR-10, we report the average accuracy across *Many-shot* ($N_c > 1000$), *Many-shot* ($100 \geq N_c \geq 200$), and *Few-shot* ($N_c < 2000$), where $N_C = [5000, 2997, 1796, 1077, 645, 387, 232, 139, 83, 50]$. For long-tailed CIFAR-100, we define the *Many-shot* ($N_c > 200$), *Many-shot* ($200 \geq N_c \geq 20$), and *Few-shot* ($N_c < 20$).

**Results on synthetic datasets.** The 8 synthetic datasets vary in degrees of class overlap, imbalance ratio, and dimensionality. Table 6 presents the case of $\pi_1 = 0.1$, $d = 25$, $\mu_D = 2$. Our approach consistently outperformed other methods across all datasets for both F1 score and G-mean, achieving the best AUC performance for most datasets. Through the relative performance ranking shown in Figure 9, our approach achieves the highest ranks across all metric classifiers, demonstrating its overall superiority over resampling methods on all metrics. The synthetic Gaussian distributional dataset results with 3 classifiers are reported in Table 10, where each row represents the (average ± std) result of 30 rounds on the distribution with different statistical settings. The win-times on synthetic datasets are reported in Table 7. The numerical results are reported in Table 8.

Table 6: Results on synthetic data with $\pi_1 = 0.1$, $d = 25$, $\mu_D = 2$ (average % ±std).

| Metric | Classifier | baseline | LNR | Over-sampling | | | Under-sampling | | |
|--------|-----------|----------|-----|-------|--------|------------|-----|----|-----|
| | | | | SMOTE | ADASYN | Borderline | OSS | CC | RUS |
| F1 score | KNN | 15.72±0.05 | **40.88±0.04** | 31.63±0.03 | 31.15±0.03 | 33.4 ±0.03 | 24.94±0.07 | 27.1 ±0.07 | 35.74±0.04 |
| | CART | 43.72±0.05 | **49.52±0.03** | 40.56±0.05 | 39.48±0.05 | 40.51±0.05 | 42.59±0.06 | 36.3 ±0.04 | 38.94±0.05 |
| | MLP | 44.32±0.05 | **47.82±0.03** | 42.68±0.05 | 42.35±0.04 | 44.01±0.04 | 43.39±0.04 | 28.84±0.02 | 30.44±0.04 |
| G-mean | KNN | 23.89±0.05 | **43.28±0.03** | 37.86±0.02 | 37.5±0.02 | 38.28 ±0.03 | 29.07±0.06 | 33.54±0.06 | 41.48±0.03 |
| | CART | 43.88±0.05 | **50.5 ±0.03** | 41.08±0.05 | 39.86±0.05 | 40.79±0.05 | 43.07±0.05 | 43.73±0.03 | 44.29±0.05 |
| | MLP | 44.55±0.04 | **48.87±0.03** | 42.95±0.05 | 42.62±0.04 | 44.18±0.04 | 43.62±0.04 | 35.61±0.02 | 36.26±0.04 |
| AUC | KNN | 71.06±0.03 | **81.53±0.03** | 73.98±0.02 | 73.52±0.03 | 74.5 ±0.02 | 73.33±0.03 | 74.41±0.03 | 80.29±0.03 |
| | CART | 69.04±0.03 | 75.58±0.03 | 68.67±0.03 | 67.88±0.03 | 67.96±0.04 | 69.72±0.03 | **75.77±0.03** | 75.45±0.04 |
| | MLP | 82.52±0.03 | **84.6 ±0.02** | 79.94±0.04 | 79.77±0.04 | 80.25±0.04 | 81.13±0.03 | 73.5 ±0.03 | 74.01±0.05 |

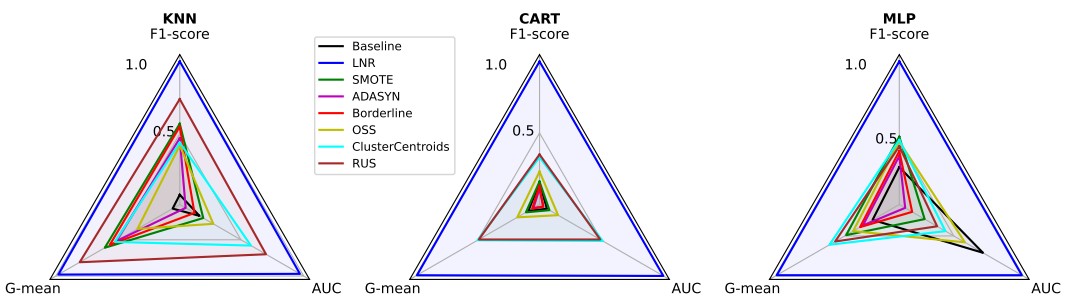

Figure 9: Relative ranking of methods from 1 (the best) to 0 (the lowest) on 8 synthetic datasets.

**Results on KEEL datasets.** The comparison results on the KEEL datasets are reported in Tables 11–19 in the format of (average% ± std) of 100 rounds for each dataset, demonstrating the superiority of our method on KEEL datasets. The win-times of each methods for each classifier on each metric are summarized below and the highest value of each column are in bold:

In order to provide a deeper understanding of the strengths and weaknesses of our method, we calculate the relative performance difference as an extension to the relative rankings. The relative performance differences between our method and re-sampling methods are calculated by: 1) First we min-max normalize the result of all methods on each dataset and obtain the relative performances. 2) Minus the relative performance of our method to calculate the differences. We then calculate the Pearson correlation coefficients between the relative performance differences and the minority class sample distribution characteristics {safe, border, rare, outlier} is shown in Table 9. Coefficients higher than $0.25$ or lower than $-0.25$ are highlighted with bold fonts.

Table 7: Win-times comparison on **8 Synthetic datasets / 32 KEEL datasets**.

| methods | KNN | | | CART | | | MLP | | |
|---|---|---|---|---|---|---|---|---|---|
| | F1 | G-mean | AUC | F1 | G-mean | AUC | F1 | G-mean | AUC |
| baseline | 0/ 2 | 0/ 2 | 0/ 1 | 0/ 4 | 0/ 4 | 0/ 1 | 0/ 2 | 0/ 2 | 2/**13** |
| LNR | **8/16** | **6/16** | **7**/ 9 | **8/21** | **8/18** | **8/13** | **8/19** | **8/16** | **6**/ 5 |
| SMOTE | 0/ 5 | 0/ 5 | 0/ 2 | 0/ 3 | 0/ 3 | 0/ 1 | 0/ 2 | 0/ 1 | 0/ 7 |
| ADASYN | 0/ 3 | 0/ 2 | 0/ 0 | 0/ 2 | 0/ 2 | 0/ 2 | 0/ 3 | 0/ 3 | 0/ 2 |
| Borderline | 0/ 4 | 0/ 4 | 0/ 0 | 0/ 1 | 0/ 1 | 0/ 0 | 0/ 4 | 0/ 4 | 0/ 2 |
| OSS | 0/ 2 | 0/ 2 | 0/ 0 | 0/ 1 | 0/ 1 | 0/ 2 | 0/ 2 | 0/ 2 | 0/ 1 |
| CC | 0/ 0 | 0/ 1 | 0/**19** | 0/ 0 | 0/ 2 | 0/ 2 | 0/ 0 | 0/ 3 | 0/ 2 |
| RUS | 0/ 0 | 2/ 0 | 1/ 1 | 0/ 0 | 0/ 1 | 0/11 | 0/ 0 | 0/ 1 | 0/ 0 |

Table 8: Relative performance ranking on 8 synthetic and 32 KEEL datasets

| Classifiers | Metrics | baseline | **LNR** | Over-sampling | | Under-sampling | | | |
|---|---|---|---|---|---|---|---|---|---|
| | | | | SMOTE | ADASYN | Borderline | OSS | CC | RUS |
| | | | | **Synthetic Gaussian datasets** | | | | | |
| | F1 | 0.08 | **1.** | 0.57 | 0.47 | 0.55 | 0.42 | 0.44 | 0.74 |
| KNN | G-mean | 0.07 | **0.98** | 0.61 | 0.52 | 0.57 | 0.35 | 0.53 | 0.81 |
| | AUC | 0.17 | **0.97** | 0.2 | 0.06 | 0.13 | 0.28 | 0.58 | 0.7 |
| | F1 | 0.15 | **1.** | 0.16 | 0.11 | 0.13 | 0.23 | 0.33 | 0.35 |
| CART | G-mean | 0.09 | **0.99** | 0.11 | 0.05 | 0.06 | 0.18 | 0.5 | 0.49 |
| | AUC | 0.06 | **1.** | 0.08 | 0.03 | 0.03 | 0.15 | 0.51 | 0.49 |
| | F1 | 0.34 | **1.** | 0.53 | 0.41 | 0.44 | 0.5 | 0.51 | 0.47 |
| MLP | G-mean | 0.3 | **0.99** | 0.49 | 0.37 | 0.39 | 0.44 | 0.61 | 0.57 |
| | AUC | 0.71 | **0.99** | 0.29 | 0.15 | 0.2 | 0.58 | 0.44 | 0.38 |
| | | | | **KEEL datasets** | | | | | |
| | F1 | 0.35 | **0.86** | 0.69 | 0.59 | 0.75 | 0.53 | 0.32 | 0.37 |
| KNN | G-mean | 0.33 | **0.84** | 0.65 | 0.53 | 0.7 | 0.5 | 0.4 | 0.43 |
| | AUC | 0.42 | 0.7 | 0.34 | 0.2 | 0.22 | 0.3 | **0.79** | 0.66 |
| | F1 | 0.58 | **0.93** | 0.56 | 0.51 | 0.54 | 0.56 | 0.19 | 0.39 |
| CART | G-mean | 0.5 | **0.89** | 0.46 | 0.37 | 0.43 | 0.47 | 0.29 | 0.5 |
| | AUC | 0.31 | **0.82** | 0.4 | 0.32 | 0.35 | 0.48 | 0.43 | 0.72 |
| | F1 | 0.36 | **0.91** | 0.72 | 0.65 | 0.75 | 0.48 | 0.34 | 0.35 |
| MLP | G-mean | 0.35 | **0.9** | 0.69 | 0.61 | 0.73 | 0.44 | 0.41 | 0.44 |
| | AUC | **0.81** | 0.8 | 0.64 | 0.46 | 0.46 | 0.45 | 0.41 | 0.39 |

# D EXTENSION TO RELATED WORKS

## D.1 IMPACT OF RESAMPLING ON CLASSIFIER'S DECISION BOUNDARY

Resampling is a data-level technique that does not alter the optimization objective of the classifier. Numerous research studies (Chawla et al., 2002; He et al., 2008; Han et al., 2005) have shown that classifiers optimized based on overall accuracy when applied to resampled data can achieve better evaluation metrics for minority class samples, such as F1, G-means, and AUC.

As mechanisms that imbalanced data bias decision boundary, the essence of how resampling methods enhance classifier performances to shift decision boundaries that maximize accuracy towards those that maximize F1 by balancing the distribution of samples.

We denote the rebalanced class-conditional distribution $\tilde{X} \mid Y = c$ for $c = 1, 0$ as $\tilde{P}_1$ and $\tilde{P}_0$. The optimal Bayesian decision boundary $x_r^*$ on the re-sampled class distribution $\tilde{P}_1, \tilde{P}_0$ has the following form:

Table 9: Pearson Correlation between data distribution characteristics and relative performance differences

| Metrics | Characteristics | baseline | Over-sampling | | | Under-sampling | | |
|---|---|---|---|---|---|---|---|---|
| | | | SMOTE | ADASYN | Borderline | OSS | CC | RUS |
| | | | with **KNN** classifier | | | | | |
| F1-score | Dimension | **0.3** | 0.16 | 0.01 | 0.08 | 0.1 | -0.13 | -0.12 |
| | IR | -0.03 | -0.08 | -0.14 | -0.13 | 0.16 | 0.17 | 0.34 |
| | Safe | **-0.77** | -0.18 | 0.06 | -0.03 | **-0.71** | -0.06 | 0.04 |
| | Border | **0.26** | 0.16 | 0.04 | 0.17 | **0.26** | 0.18 | -0.03 |
| | Rare | **0.68** | 0.08 | -0.13 | 0.02 | **0.68** | 0.01 | -0.12 |
| | Outlier | **0.48** | 0.07 | -0.04 | -0.11 | **0.39** | -0.05 | 0.04 |
| G-mean | Dimension | **0.32** | 0.14 | -0.01 | 0.06 | 0.12 | -0.1 | -0.1 |
| | IR | -0.02 | -0.07 | -0.1 | -0.08 | 0.09 | 0.04 | 0.13 |
| | Safe | **-0.76** | -0.23 | -0.02 | -0.12 | **-0.63** | 0.09 | 0.22 |
| | Border | **0.3** | 0.16 | 0.04 | 0.12 | **0.31** | 0.25 | 0.08 |
| | Rare | **0.69** | 0.15 | -0.06 | 0.13 | **0.63** | -0.16 | -0.32 |
| | Outlier | **0.43** | 0.1 | 0.02 | -0.01 | **0.27** | -0.23 | -0.19 |
| AUC | Dimension | 0.1 | **0.31** | 0.24 | **0.31** | 0.21 | **0.27** | 0.47 |
| | IR | **0.49** | 0.21 | 0.05 | 0.1 | **0.3** | -0.23 | -0.18 |
| | Safe | **-0.56** | **-0.53** | **-0.33** | **-0.43** | **-0.57** | **-0.28** | -0.3 |
| | Border | -0.06 | 0.23 | 0.17 | 0.22 | 0.1 | **0.31** | 0.22 |
| | Rare | **0.44** | **0.39** | 0.2 | **0.26** | **0.6** | **0.3** | 0.29 |
| | Outlier | **0.59** | **0.33** | 0.2 | **0.27** | **0.38** | -0.03 | 0.08 |
| | | | with **CART** classifier | | | | | |
| F1-score | Dimension | **0.4** | **0.29** | **0.26** | **0.27** | **0.31** | 0.01 | -0.15 |
| | IR | **-0.3** | -0.15 | -0.16 | -0.15 | -0.13 | 0.18 | 0.48 |
| | Safe | -0.25 | **-0.59** | **-0.57** | **-0.42** | -0.14 | -0.12 | -0.26 |
| | Border | 0.23 | **0.27** | 0.18 | 0.21 | 0.05 | -0.04 | -0.05 |
| | Rare | **0.32** | **0.63** | **0.59** | **0.42** | 0.16 | 0.07 | 0.15 |
| | Outlier | -0.03 | 0.24 | **0.3** | 0.17 | 0.06 | 0.17 | 0.32 |
| G-mean | Dimension | **0.41** | **0.27** | 0.24 | **0.26** | **0.33** | 0.03 | -0.07 |
| | IR | **-0.32** | -0.06 | -0.08 | -0.12 | -0.1 | -0.14 | -0.05 |
| | Safe | **-0.27** | **-0.66** | **-0.6** | **-0.49** | -0.21 | **0.28** | 0.24 |
| | Border | **0.32** | **0.31** | 0.22 | 0.23 | 0.14 | 0.24 | 0.31 |
| | Rare | 0.29 | **0.63** | **0.53** | **0.46** | 0.21 | **-0.36** | -0.33 |
| | Outlier | -0.05 | **0.31** | **0.37** | 0.24 | 0.06 | **-0.38** | -0.4 |
| AUC | Dimension | **0.39** | **0.41** | **0.34** | **0.36** | **0.32** | 0.14 | 0.05 |
| | IR | -0.24 | -0.04 | -0.05 | -0.07 | 0.02 | **-0.4** | -0.56 |
| | Safe | -0.03 | **-0.58** | **-0.41** | **-0.36** | -0.2 | 0.17 | 0.29 |
| | Border | 0.16 | 0.23 | 0.1 | 0.14 | 0.12 | 0.22 | 0.26 |
| | Rare | 0.04 | **0.52** | **0.36** | **0.32** | **0.27** | -0.1 | -0.19 |
| | Outlier | -0.13 | **0.31** | **0.29** | 0.2 | 0.01 | **-0.37** | -0.52 |
| | | | with **MLP** classifier | | | | | |
| F1-score | Dimension | **0.28** | 0.04 | 0.11 | 0.12 | **0.31** | 0.01 | 0.02 |
| | IR | -0.05 | 0.08 | -0.02 | 0.03 | 0.03 | 0.14 | 0.3 |
| | Safe | **-0.75** | -0.05 | 0.11 | 0.02 | **-0.46** | 0.05 | 0.04 |
| | Border | **0.38** | -0.24 | **-0.29** | -0.2 | -0.16 | -0.09 | -0.09 |
| | Rare | **0.63** | 0.13 | -0.03 | 0.06 | **0.48** | -0.03 | -0.16 |
| | Outlier | **0.39** | 0.2 | 0.09 | 0.1 | **0.5** | 0.02 | 0.11 |
| G-mean | Dimension | **0.31** | 0.15 | 0.14 | 0.15 | **0.29** | 0.03 | -0.01 |
| | IR | -0.04 | -0.02 | -0.04 | 0.03 | 0.02 | 0.01 | 0.15 |
| | Safe | **-0.74** | -0.11 | 0.06 | -0.03 | **-0.48** | 0.18 | 0.18 |
| | Border | **0.42** | -0.18 | **-0.26** | -0.21 | -0.1 | -0.02 | -0.01 |
| | Rare | **0.59** | 0.14 | -0.02 | 0.09 | **0.45** | -0.16 | -0.32 |
| | Outlier | **0.37** | 0.21 | 0.13 | 0.16 | **0.49** | -0.14 | -0.06 |
| AUC | Dimension | -0.25 | -0.18 | -0.15 | 0.01 | -0.11 | 0.03 | 0.1 |
| | IR | -0.07 | **0.42** | **0.35** | **0.28** | -0.08 | 0.12 | 0.39 |
| | Safe | 0.05 | **-0.5** | -0.17 | **-0.27** | 0.13 | **-0.43** | -0.56 |
| | Border | -0.21 | **-0.28** | **-0.38** | **-0.36** | 0.2 | 0.22 | 0.19 |
| | Rare | 0.05 | **0.47** | 0.25 | **0.35** | -0.2 | **0.36** | 0.35 |
| | Outlier | 0.08 | **0.67** | **0.39** | **0.47** | -0.22 | 0.21 | 0.44 |

$$S^r = \{x_r^* \in \mathcal{X} : \frac{P_1(x)}{P_0(x)} = \gamma \frac{\pi_0}{\pi_1}\}$$

where $\gamma$ is the resampling rate, over-sampling methods set $\gamma \geq 1$, and under-sampling methods set $\gamma \in [0, 1]$. When $\gamma = \frac{\pi_1}{\pi_0}$, the corresponding class ratio after re-sampling is fully balanced.

However, the fully balanced data often does not yield optimal performance for classifiers. Therefore, the concept of "optimal degree of balance" has been empirically studied (He & Garcia, 2009; Estabrooks et al., 2004; Weiss & Provost, 2003). (Estabrooks et al., 2004) suggests hybridizing under-sampling with over-sampling is efficient for tuning the degree of balance, avoiding excessively removing or adding samples.

### D.2    INFORMED UNDER-SAMPLING

Tomek-links (Tomek, 1976) first finds the nearest neighbor (1-Nearest Neighbour) for each sample as links. If the nearest neighbor and the sample come from different classes, the majority class sample in this link is removed from the data. This simple and efficient cleaning rule prioritizes the removal of samples in class-overlapping regions, thereby reducing the impact of imbalanced data on the classification boundary.

Instead of using 1-NN, the ENN (Edited Nearest Neighbours) method (Wilson & Martinez, 2000) calculates $k$-nearest neighboring samples for each majority class sample to obtain smoother guidance from the data distribution and removes samples that are inconsistent with their nearest neighbors. ENN exclusively removes the noisy and ambiguous instances along the decision boundary. However, it's insufficient to balance the data with large imbalance ratio by merely removing the majority class samples around the borderline. Therefore, the CNNs (Condensed Nearest Neighbors) (Hart, 1968) was originally proposed to reduce the memory requirements for the KNN algorithm to mitigate the gap. CNNs retained all minority class samples and only the majority class samples that can not be classified correctly with the rest of the data. It aims to find the smallest subset of the most informative data and remove the redundant majority class samples far from the decision boundary. The CNNs algorithm can be run iteratively to achieve the desired size of under-sampled data.

However, CNNs is extremely sensitive to noise, as noisy samples are more likely to be misclassified and retained by CNNs. In advance, the **OSS** (One-Sided Selection) (Kubat & Matwin, 2000) method first removes redundant examples from the majority class that are far from the decision boundary using CNNs and then cleans the noisy samples retained by CNNs using the Tomek-links. (Li et al., 2022a) has argued that data cleaning should be done before the reduction of the majority class. Therefore, NCR (Li et al., 2022a) first applies ENN to obtain data with less noise before reducing the redundant majority class samples.

However, the methods mentioned above involve reducing class imbalance by removing samples, which often leads to under-sampled majority class samples no longer being representative of the original ones. Instead of discarding majority class samples, the **CC** (ClusterCentroids) (Lin et al., 2017) approach reduces class imbalance by replacing the original majority class dataset with centroids of the K-means. By adjusting the number of centroids, CC can effectively reduce the majority class samples while ensuring that the synthetic samples remain representative of the original data, achieving higher AUC performance.

### D.3    SMOTE-BASED OVER-SAMPLING

**SMOTE.** For each instance $x_i$ in the minority class, SMOTE searches its $k$-nearest neighbors in the same class. One neighbor is randomly selected, denoted as $x_i'$, and the seed samples are $x_i$ and $x_i'$. The new sample $x_{new}$ is then generated at:

$$X_{new} = x_i + (x_i' - x_i)\delta$$

where $\delta$ is a random number between [0,1]. Figure (a) demonstrates the idea of SMOTE.

The drawback of SMOTE is obvious. The synthetic samples do not remain in the same distribution as the seed samples, which introduces generative errors to the original distribution (Karystinos & Pados, 2000). SMOTE also uses outliers/noise as the seed samples, which increases the generative errors undesirably. For instance, three outliers on the right of Figure 10 generate unwanted new samples.

**Borderline-SMOTE.** Borderline-SMOTE is proposed in (Han et al., 2005) to address this problem, in which only the minority samples near the borderline are over-sampled. Borderline-SMOTE divides all minority samples into 3 groups: SAFE, DANGER, and NOISE. For each instance $x_i$ in the minority class, borderline-SMOTE searches its $m$-nearest neighbors from the entire data. The

number of minority samples among the $m$-nearest neighbors is denoted by $m'$. If all neighbors of $x_i$ are from the majority class, in this case, $m = m'$, then $x_i$ is considered as NOISE and eliminated from the generating steps. In the case of $m > m' > \frac{m}{2}$, namely, the number of majority nearest neighbors is larger than the number of its minority ones, $x_i$ is added to the DANGER set. Only the minority samples in the DANGER set are used as seed samples. The generating mechanism is similar to SMOTE. Figure 10(b) shows the synthetic samples generated by borderline-SMOTE.

The borderline-SMOTE naturally focuses on the difficult-to-learn samples, namely the borderline minority samples mixture with the majority samples. However, the threshold provides a rigid boundary when defining the DANGER set. The generating process is still blinked without further guidance.

**ADASYN.** ADASYN (He et al., 2008) followed a similar idea of forcing the classifier to focus on the samples lying on the borderline but with a much softer boundary. ADASYN finds the impurity of the neighborhood by normalizing $r_i = \delta_i / k$, where the $\delta_i$ is the number of majority samples in the $k$-nearest neighbors. Then, the normalized $r'_i = r_i / \sum_i^{m_s} r_i$ is a density distribution that sums to 1. The number of synthetic data examples that need to be generated for each minority example $x_i$ is calculated by $g_i = r'_i \times (m_l - m_s) \times \beta$, where $m_l$ and $m_s$ is the number of observations of majority and minority classes, respectively, $\beta$ is a number from [0,1] that specifies the desired balance level after generating the synthetic data. The generating procedure is the same as SMOTE. ADASYN provides impurity as guidance while generating the synthetic data. However, the noise samples are used as seed instances with a higher weight as the impurity becomes 1. Figure 10(c) represents the synthetic samples produced by ADASYN.

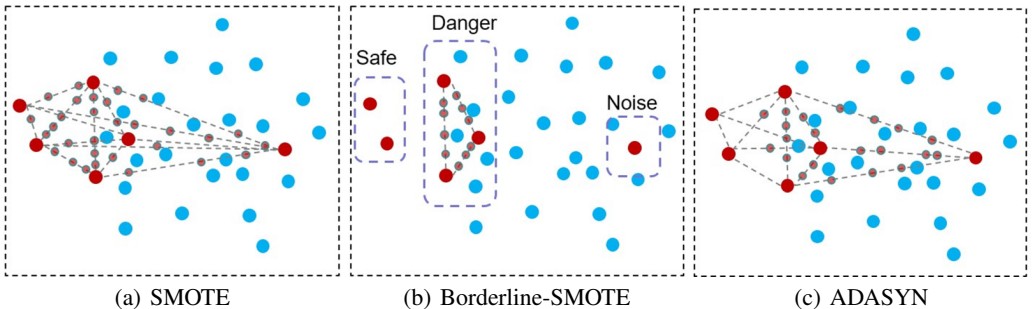

| (a) SMOTE | (b) Borderline-SMOTE | (c) ADASYN |

Figure 10: Informed Over-sampling methods. The larger red dots represent the original seed sample in the minority class. The smaller red dots demonstrate the synthetic samples by three methods.

Figures 10(a) and 10(c) show that SMOTE and ADASYN generate more outliers that deviate from the original distribution by leveraging noise from the minority class samples on the right side of the images. Borderline addresses this issue by dividing the data into three groups to mitigate the problem; however, it can be seen that they still encounter similar challenges: the newly generated samples do not adhere to the distribution of the minority class samples. Borderline-SMOTE and ADASYN tend to synthesize samples on the overlapping region, likely enlarging the overlaps between the minority and majority classes. As a result, the decision boundary may become blurred and distorted.

Table 10: Comparison on synthetic datasets with KNN, CART and MLP classifiers.

| Metric | Settings $\pi_1,\mu_D,d$ | baseline | LNR | Over-sampling SMOTE | ADASYN | Borderline | Under-sampling OSS | CC | RUS |
|---|---|---|---|---|---|---|---|---|---|
| | | | | with **KNN** classifier | | | | | |
| F1-score | 0.1,1,5 | 8.54±0.03 | **29.82±0.03** | 22.05±0.02 | 21.5±0.02 | 23.24±0.03 | 12.85±0.04 | 12.85±0.04 | 24.94±0.02 |
| | 0.1,1,25 | 2.0±0.01 | **21.8±0.03** | 19.47±0.01 | 19.5±0.01 | 19.86±0.02 | 5.19±0.03 | 5.19±0.03 | 20.73±0.03 |
| | 0.1,2,5 | 43.38±0.06 | **55.2±0.04** | 44.21±0.03 | 42.68±0.04 | 48.07±0.04 | 47.76±0.05 | 47.76±0.05 | 44.86±0.04 |
| | 0.1,2,25 | 15.72±0.05 | **40.88±0.04** | 31.63±0.03 | 31.15±0.03 | 33.4±0.03 | 24.94±0.07 | 24.94±0.07 | 35.74±0.04 |
| | 0.3,1,5 | 39.7±0.03 | **54.19±0.02** | 47.45±0.02 | 47.23±0.01 | 46.67±0.02 | 45.99±0.02 | 45.99±0.02 | 50.38±0.02 |
| | 0.3,1,25 | 27.61±0.03 | **48.88±0.02** | 45.42±0.02 | 45.1±0.01 | 45.08±0.01 | 36.49±0.03 | 36.49±0.03 | 44.05±0.03 |
| | 0.3,2,5 | 71.14±0.03 | **73.64±0.02** | 70.6±0.02 | 67.82±0.02 | 68.3±0.02 | 72.66±0.02 | 72.66±0.02 | 72.58±0.02 |
| | 0.3,2,25 | 54.75±0.04 | **66.1±0.02** | 60.11±0.02 | 58.13±0.02 | 58.3±0.02 | 60.91±0.04 | 60.91±0.04 | 64.6±0.02 |
| G-mean | 0.1,1,5 | 12.12±0.04 | 31.26±0.03 | 24.57±0.02 | 24.07±0.02 | 24.45±0.03 | 14.83±0.04 | **32.86±0.02** | 31.4±0.03 |
| | 0.1,1,25 | 3.98±0.02 | 23.87±0.03 | 25.43±0.02 | 25.69±0.02 | 24.29±0.02 | 6.97±0.03 | 8.22±0.04 | **26.42±0.03** |
| | 0.1,2,5 | 46.52±0.05 | **56.04±0.04** | 46.8±0.03 | 45.28±0.04 | 49.29±0.04 | 49.04±0.05 | 51.12±0.05 | 50.49±0.03 |
| | 0.1,2,25 | 23.89±0.05 | **43.28±0.03** | 37.86±0.02 | 37.5±0.02 | 38.28±0.03 | 29.07±0.06 | 33.54±0.06 | 41.48±0.03 |
| | 0.3,1,5 | 40.5±0.02 | **55.38±0.02** | 48.2±0.02 | 48.42±0.01 | 47.51±0.02 | 46.09±0.02 | 54.36±0.02 | 51.63±0.02 |
| | 0.3,1,25 | 29.17±0.03 | **51.57±0.03** | 49.19±0.03 | 49.02±0.02 | 48.73±0.02 | 36.73±0.03 | 31.98±0.03 | 45.34±0.03 |
| | 0.3,2,5 | 71.32±0.03 | **74.15±0.02** | 71.07±0.02 | 68.76±0.02 | 69.08±0.02 | 72.83±0.02 | 72.62±0.01 | 73.3±0.02 |
| | 0.3,2,25 | 56.22±0.03 | **67.83±0.02** | 63.71±0.02 | 62.23±0.02 | 62.42±0.02 | 61.16±0.04 | 59.37±0.03 | 65.98±0.03 |
| AUC | 0.1,1,5 | 60.2±0.02 | **69.2±0.02** | 60.18±0.02 | 59.94±0.02 | 60.98±0.02 | 60.76±0.03 | 68.02±0.03 | 66.35±0.03 |
| | 0.1,1,25 | 54.93±0.02 | **61.36±0.03** | 55.74±0.02 | 55.65±0.02 | 56.33±0.03 | 55.75±0.02 | 57.04±0.03 | 59.14±0.03 |
| | 0.1,2,5 | 81.19±0.02 | 87.1±0.03 | 80.69±0.02 | 79.74±0.02 | 80.93±0.02 | 82.34±0.02 | **89.67±0.01** | 88.16±0.02 |
| | 0.1,2,25 | 71.06±0.03 | **81.53±0.03** | 73.98±0.02 | 73.52±0.03 | 74.5±0.02 | 73.33±0.03 | 74.41±0.03 | 80.29±0.03 |
| | 0.3,1,5 | 65.27±0.01 | **71.54±0.02** | 63.95±0.02 | 62.81±0.02 | 63.11±0.02 | 65.92±0.01 | 67.96±0.02 | 67.15±0.02 |
| | 0.3,1,25 | 57.36±0.01 | **64.54±0.02** | 56.55±0.03 | 55.12±0.02 | 55.87±0.02 | 57.7±0.01 | 58.9±0.01 | 59.22±0.02 |
| | 0.3,2,5 | 87.83±0.02 | **89.63±0.01** | 86.49±0.02 | 84.21±0.01 | 84.06±0.02 | 88.35±0.02 | 89.38±0.01 | 89.34±0.01 |
| | 0.3,2,25 | 78.82±0.02 | **85.49±0.01** | 81.36±0.02 | 79.58±0.03 | 79.64±0.03 | 79.79±0.03 | 80.22±0.02 | 83.31±0.02 |
| | | | | with **CART** classifier | | | | | |
| F1-score | 0.1,1,5 | 19.13±0.04 | **27.94±0.03** | 19.22±0.02 | 19.5±0.03 | 18.83±0.03 | 19.22±0.03 | 23.07±0.03 | 22.21±0.02 |
| | 0.1,1,25 | 16.3±0.03 | **24.07±0.02** | 17.6±0.03 | 17.02±0.03 | 16.7±0.03 | 17.3±0.02 | 21.66±0.02 | 21.29±0.02 |
| | 0.1,2,5 | 44.77±0.05 | **52.49±0.04** | 44.49±0.05 | 42.96±0.06 | 44.43±0.05 | 43.96±0.05 | 41.99±0.06 | 39.6±0.05 |
| | 0.1,2,25 | 43.72±0.05 | **49.52±0.03** | 40.56±0.05 | 39.48±0.05 | 40.51±0.05 | 42.59±0.06 | 36.3±0.04 | 38.94±0.05 |
| | 0.3,1,5 | 41.07±0.02 | **52.0±0.02** | 42.24±0.02 | 42.64±0.02 | 42.16±0.02 | 43.98±0.02 | 45.33±0.02 | 46.14±0.02 |
| | 0.3,1,25 | 40.11±0.02 | **50.01±0.02** | 40.53±0.03 | 40.84±0.02 | 40.48±0.03 | 41.32±0.02 | 45.38±0.02 | 44.6±0.02 |
| | 0.3,2,5 | 65.66±0.02 | **71.68±0.02** | 66.03±0.02 | 65.0±0.02 | 65.93±0.02 | 66.88±0.02 | 66.32±0.02 | 67.12±0.03 |
| | 0.3,2,25 | 64.18±0.02 | **69.7±0.02** | 64.02±0.02 | 63.39±0.02 | 63.38±0.03 | 64.75±0.02 | 64.63±0.02 | 65.37±0.02 |
| G-mean | 0.1,1,5 | 19.22±0.04 | **30.48±0.03** | 19.76±0.02 | 19.91±0.03 | 19.05±0.03 | 19.58±0.03 | 28.41±0.03 | 28.39±0.02 |
| | 0.1,1,25 | 16.42±0.03 | 27.33±0.04 | 18.09±0.03 | 17.48±0.03 | 16.98±0.03 | 17.81±0.02 | **28.68±0.02** | 26.68±0.03 |
| | 0.1,2,5 | 44.91±0.05 | **53.83±0.03** | 44.97±0.05 | 43.49±0.06 | 44.68±0.05 | 44.51±0.05 | 46.82±0.05 | 45.28±0.04 |
| | 0.1,2,25 | 43.88±0.05 | **50.5±0.03** | 41.08±0.05 | 39.86±0.05 | 40.79±0.05 | 43.07±0.05 | 43.73±0.03 | 44.29±0.05 |
| | 0.3,1,5 | 41.11±0.02 | **53.42±0.02** | 42.38±0.02 | 42.88±0.02 | 42.26±0.02 | 44.29±0.02 | 46.29±0.02 | 47.29±0.03 |
| | 0.3,1,25 | 40.15±0.02 | **51.55±0.02** | 40.73±0.03 | 41.01±0.02 | 40.63±0.03 | 41.57±0.03 | 46.55±0.02 | 45.61±0.02 |
| | 0.3,2,5 | 65.69±0.02 | **72.29±0.02** | 66.13±0.02 | 65.13±0.02 | 66.02±0.02 | 67.09±0.02 | 66.86±0.02 | 67.73±0.03 |
| | 0.3,2,25 | 64.21±0.02 | **70.24±0.02** | 64.15±0.02 | 63.51±0.02 | 63.51±0.03 | 64.92±0.02 | 65.36±0.02 | 65.88±0.02 |
| AUC | 0.1,1,5 | 55.11±0.02 | **62.86±0.02** | 55.06±0.02 | 55.22±0.02 | 54.89±0.02 | 55.1±0.02 | 59.5±0.03 | 58.74±0.03 |
| | 0.1,1,25 | 53.44±0.02 | **60.01±0.03** | 53.9±0.02 | 53.54±0.02 | 53.41±0.02 | 53.66±0.02 | 58.2±0.03 | 57.3±0.03 |
| | 0.1,2,5 | 69.66±0.03 | **78.42±0.03** | 71.22±0.03 | 70.35±0.04 | 70.26±0.03 | 70.93±0.04 | 77.12±0.03 | 76.46±0.03 |
| | 0.1,2,25 | 69.04±0.03 | 75.58±0.03 | 68.67±0.03 | 67.88±0.03 | 67.96±0.04 | 69.72±0.03 | **75.77±0.03** | 75.45±0.04 |
| | 0.3,1,5 | 57.62±0.01 | **64.11±0.01** | 57.81±0.02 | 57.76±0.01 | 57.93±0.02 | 58.64±0.02 | 58.36±0.02 | 58.87±0.02 |
| | 0.3,1,25 | 57.0±0.02 | **62.16±0.02** | 56.18±0.02 | 56.58±0.02 | 56.31±0.02 | 56.79±0.02 | 58.06±0.02 | 57.63±0.02 |
| | 0.3,2,5 | 75.54±0.02 | **80.98±0.02** | 76.0±0.02 | 75.25±0.02 | 75.89±0.02 | 76.77±0.01 | 76.6±0.01 | 77.32±0.02 |
| | 0.3,2,25 | 74.48±0.02 | **79.31±0.01** | 74.51±0.02 | 74.03±0.02 | 74.02±0.02 | 75.09±0.02 | 75.32±0.02 | 75.81±0.02 |
| | | | | with **MLP** classifier | | | | | |
| F1-score | 0.1,1,5 | 7.94±0.08 | **30.36±0.03** | 23.0±0.03 | 22.9±0.03 | 23.04±0.03 | 14.79±0.09 | 27.05±0.02 | 25.66±0.02 |
| | 0.1,1,25 | 16.91±0.04 | **24.66±0.02** | 18.99±0.03 | 19.52±0.03 | 18.29±0.03 | 16.97±0.04 | 21.31±0.02 | 20.96±0.02 |
| | 0.1,2,5 | 54.57±0.04 | **57.46±0.04** | 47.51±0.04 | 45.43±0.05 | 50.95±0.05 | 55.24±0.04 | 48.86±0.04 | 45.52±0.04 |
| | 0.1,2,25 | 44.32±0.05 | **47.82±0.03** | 42.68±0.05 | 42.35±0.04 | 44.01±0.04 | 43.39±0.04 | 28.84±0.02 | 30.44±0.04 |
| | 0.3,1,5 | 45.19±0.03 | **54.57±0.02** | 51.84±0.03 | 52.37±0.02 | 51.93±0.02 | 51.13±0.03 | 53.66±0.02 | 53.94±0.02 |
| | 0.3,1,25 | 38.83±0.03 | **49.7±0.02** | 43.2±0.02 | 43.64±0.02 | 42.93±0.02 | 41.73±0.02 | 44.64±0.02 | 44.79±0.02 |
| | 0.3,2,5 | 74.34±0.02 | **74.87±0.02** | 74.6±0.02 | 72.14±0.02 | 72.12±0.02 | 74.76±0.02 | 74.58±0.02 | 74.48±0.01 |
| | 0.3,2,25 | 64.5±0.03 | **68.61±0.02** | 65.12±0.02 | 64.71±0.02 | 64.67±0.03 | 64.07±0.03 | 62.69±0.02 | 62.62±0.03 |
| G-mean | 0.1,1,5 | 10.57±0.09 | 31.85±0.03 | 25.22±0.03 | 25.39±0.03 | 23.94±0.03 | 16.82±0.08 | **32.26±0.02** | 31.53±0.02 |
| | 0.1,1,25 | 17.03±0.04 | 27.33±0.03 | 19.5±0.02 | 20.15±0.03 | 18.57±0.0 3 | 17.21±0.04 | **27.93±0.02** | 27.22±0.02 |
| | 0.1,2,5 | 55.74±0.04 | **58.02±0.04** | 49.29±0.04 | 47.57±0.05 | 51.86±0.05 | 55.77±0.04 | 52.72±0.03 | 50.13±0.03 |
| | 0.1,2,25 | 44.55±0.04 | **48.87±0.03** | 42.95±0.05 | 42.62±0.04 | 44.18±0.04 | 43.62±0.04 | 35.61±0.02 | 36.26±0.04 |
| | 0.3,1,5 | 46.37±0.03 | **55.37±0.02** | 52.45±0.03 | 53.42±0.02 | 52.76±0.02 | 51.31±0.03 | 54.64±0.02 | 54.75±0.02 |
| | 0.3,1,25 | 38.97±0.03 | **51.01±0.02** | 43.44±0.02 | 44.01±0.02 | 43.15±0.02 | 41.88±0.03 | 45.63±0.03 | 45.69±0.03 |
| | 0.3,2,5 | 74.48±0.02 | **75.08±0.02** | 74.83±0.02 | 72.85±0.02 | 72.76±0.02 | 74.82±0.02 | 74.88±0.02 | 74.8±0.01 |
| | 0.3,2,25 | 64.56±0.03 | **69.14±0.02** | 65.23±0.02 | 64.82±0.02 | 64.81±0.03 | 64.22±0.03 | 63.53±0.02 | 63.31±0.03 |
| AUC | 0.1,1,5 | **71.4±0.03** | 71.31±0.03 | 62.85±0.04 | 63.04±0.04 | 64.61±0.04 | 69.73±0.04 | 69.32±0.03 | 67.45±0.03 |
| | 0.1,1,25 | 61.51±0.04 | **64.07±0.03** | 58.48±0.04 | 58.53±0.03 | 57.73±0.04 | 59.13±0.04 | 60.73±0.03 | 59.31±0.04 |
| | 0.1,2,5 | **90.28±0.01** | 89.9±0.02 | 85.39±0.03 | 84.09±0.04 | 86.77±0.03 | 89.65±0.01 | 88.72±0.03 | 87.74±0.02 |
| | 0.1,2,25 | 82.52±0.03 | **84.6±0.02** | 79.94±0.04 | 79.77±0.04 | 80.25±0.04 | 81.13±0.03 | 73.5±0.03 | 74.01±0.05 |
| | 0.3,1,5 | 73.19±0.02 | **73.24±0.02** | 70.83±0.02 | 70.01±0.02 | 70.15±0.0 2 | 73.0±0.02 | 72.13±0.02 | 72.25±0.02 |
| | 0.3,1,25 | 59.87±0.03 | **65.94±0.01** | 60.88±0.02 | 61.04±0.02 | 60.76±0.02 | 60.62±0.03 | 60.28±0.02 | 60.61±0.03 |
| | 0.3,2,5 | 91.2±0.01 | **91.22±0.01** | 90.68±0.01 | 89.64±0.02 | 89.34±0.01 | 91.01±0.01 | 90.9±0.01 | 90.79±0.01 |
| | 0.3,2,25 | 81.84±0.03 | **85.69±0.02** | 81.81±0.02 | 81.13±0.03 | 81.41±0.03 | 8 0.8±0.03 | 80.56±0.02 | 80.42±0.03 |

Table 11: **F1 score** with **KNN** classifier on KEEL datasets

| Dataset | baseline | LNR | Over-sampling | | | Under-sampling | | |
|---|---|---|---|---|---|---|---|---|
| | | | SMOTE | ADASYN | Borderline | OSS | CC | RUS |
| 1 | 9.75±0.09 | **39.12±0.08** | 33.18±0.06 | 32.05±0.06 | 34.72±0.07 | 15.63±0.09 | 25.67±0.05 | 19.22±0.04 |
| 2 | 0.17±0.02 | **15.19±0.07** | 13.13±0.06 | 13.17±0.06 | 11.02±0.08 | 0.35±0.02 | 8.75±0.03 | 7.34±0.02 |
| 3 | 26.11±0.14 | **43.89±0.11** | 36.06±0.07 | 35.0±0.07 | 42.49±0.1 | 31.88±0.13 | 33.79±0.07 | 29.54±0.05 |
| 4 | 77.55±0.06 | 78.29±0.06 | **79.57±0.05** | 77.64±0.05 | 78.6±0.05 | 78.47±0.06 | 78.22±0.05 | 78.13±0.05 |
| 5 | **88.72±0.05** | 88.57±0.05 | 82.39±0.05 | 77.62±0.06 | 80.75±0.06 | 86.17±0.06 | 84.21±0.06 | 76.32±0.08 |
| 6 | 60.99±0.1 | **69.53±0.08** | 62.69±0.08 | 61.12±0.07 | 63.1±0.09 | 66.67±0.09 | 58.27±0.08 | 56.72±0.08 |
| 7 | 23.22±0.12 | **32.73±0.08** | 29.6±0.07 | 29.72±0.07 | 29.74±0.07 | 22.68±0.12 | 16.58±0.04 | 23.3±0.05 |
| 8 | 78.36±0.08 | 80.06±0.07 | 85.53±0.05 | **85.84±0.05** | 85.25±0.06 | 83.59±0.08 | 82.86±0.06 | 81.98±0.06 |
| 9 | 72.1±0.05 | 72.17±0.05 | **73.47±0.05** | 72.37±0.05 | 72.85±0.05 | 71.19±0.06 | 66.98±0.05 | 70.93±0.05 |
| 10 | 65.72±0.07 | 68.51±0.06 | 68.52±0.06 | **69.65±0.06** | 69.43±0.06 | 66.27±0.07 | 65.31±0.06 | 67.79±0.06 |
| 11 | 35.23±0.07 | **47.75±0.07** | 43.48±0.06 | 43.54±0.07 | 43.12±0.06 | 37.7±0.08 | 39.29±0.07 | 45.54±0.06 |
| 12 | 65.11±0.09 | 63.66±0.08 | **72.64±0.06** | 65.0±0.08 | 71.17±0.07 | 64.27±0.1 | 29.04±0.04 | 28.22±0.05 |
| 13 | 77.01±0.06 | 66.98±0.07 | **78.78±0.07** | 60.15±0.05 | 62.49±0.05 | 56.62±0.2 | 44.23±0.04 | 38.79±0.07 |
| 14 | 77.05±0.09 | **77.45±0.07** | 67.07±0.09 | 68.49±0.1 | 68.46±0.12 | 54.9±0.28 | 54.44±0.07 | 59.83±0.14 |
| 15 | 86.21±0.09 | 89.92±0.07 | **94.25±0.05** | 93.83±0.05 | 93.72±0.05 | 90.63±0.06 | 91.77±0.05 | 92.2±0.06 |
| 16 | 82.72±0.02 | **83.35±0.02** | 81.71±0.02 | 79.93±0.02 | 82.66±0.02 | 83.09±0.02 | 79.22±0.02 | 77.21±0.02 |
| 17 | 60.36±0.04 | **66.88±0.03** | 63.54±0.04 | 63.58±0.03 | 63.37±0.03 | 62.02±0.04 | 61.63±0.04 | 64.05±0.03 |
| 18 | 52.66±0.05 | **64.24±0.04** | 61.62±0.03 | 61.83±0.03 | 61.9±0.03 | 57.65±0.05 | 60.9±0.04 | 60.57±0.04 |
| 19 | 47.38±0.05 | **63.13±0.03** | 58.41±0.04 | 59.71±0.03 | 58.79±0.03 | 53.64±0.05 | 60.11±0.04 | 59.31±0.03 |
| 20 | 57.96±0.07 | **61.61±0.06** | 49.79±0.04 | 45.57±0.04 | 55.17±0.05 | 59.6±0.07 | 42.41±0.06 | 48.41±0.06 |
| 21 | **83.07±0.05** | 82.74±0.04 | 69.12±0.04 | 66.1±0.04 | 76.65±0.04 | 83.0±0.05 | 72.02±0.06 | 71.15±0.05 |
| 22 | 35.86±0.09 | **42.58±0.09** | 37.48±0.06 | 36.5±0.06 | 41.16±0.07 | 37.73±0.1 | 34.31±0.06 | 34.85±0.06 |
| 23 | 33.27±0.14 | 47.76±0.09 | 45.6±0.06 | 44.28±0.06 | **48.52±0.07** | 46.52±0.1 | 41.28±0.06 | 42.69±0.05 |
| 24 | 5.48±0.09 | 18.46±0.09 | 23.88±0.07 | 21.75±0.08 | **30.48±0.1** | 11.35±0.12 | 15.24±0.04 | 15.34±0.05 |
| 25 | 1.52±0.05 | 17.42±0.08 | 19.68±0.06 | **20.2±0.06** | 18.29±0.09 | 4.21±0.07 | 14.44±0.03 | 13.9±0.03 |
| 26 | 13.4±0.14 | 33.24±0.1 | 38.32±0.08 | 35.12±0.09 | **44.6±0.09** | 28.01±0.15 | 31.59±0.07 | 29.22±0.07 |
| 27 | 75.89±0.07 | 77.0±0.07 | 73.9±0.07 | 72.58±0.07 | 74.4±0.08 | **79.32±0.06** | 73.97±0.06 | 75.16±0.07 |
| 28 | 53.73±0.03 | **59.79±0.02** | 56.92±0.03 | 56.41±0.02 | 56.83±0.02 | 54.61±0.03 | 57.76±0.02 | 57.99±0.03 |
| 29 | 75.66±0.04 | **77.96±0.04** | 70.65±0.04 | 68.71±0.04 | 69.73±0.04 | 76.64±0.04 | 63.95±0.04 | 66.9±0.04 |
| 30 | 18.39±0.1 | 39.83±0.09 | 35.38±0.07 | 35.33±0.07 | **41.48±0.08** | 27.62±0.12 | 25.04±0.04 | 26.66±0.04 |
| 31 | 63.81±0.09 | **69.99±0.07** | 68.95±0.06 | 69.31±0.07 | 68.4±0.07 | 66.51±0.08 | 43.87±0.05 | 44.28±0.06 |
| 32 | 54.57±0.1 | 56.62±0.1 | 40.29±0.06 | 39.92±0.07 | 52.68±0.09 | **57.75±0.1** | 24.68±0.05 | 26.98±0.05 |

Table 12: **G-mean** with **KNN** classifier on KEEL datasets

| Dataset | baseline | LNR | Over-sampling | | | Under-sampling | | |
|---|---|---|---|---|---|---|---|---|
| | | | SMOTE | ADASYN | Borderline | OSS | CC | RUS |
| 1 | 16.8±0.13 | **39.98±0.08** | 35.02±0.06 | 33.64±0.06 | 35.21±0.07 | 22.78±0.11 | 33.9±0.04 | 29.95±0.04 |
| 2 | 0.3±0.03 | **17.72±0.08** | 15.47±0.07 | 15.52±0.07 | 11.33±0.08 | 0.62±0.04 | 16.2±0.04 | 15.69±0.03 |
| 3 | 35.35±0.15 | **45.11±0.11** | 37.61±0.07 | 36.57±0.08 | 43.2±0.1 | 38.32±0.14 | 36.68±0.07 | 34.93±0.06 |
| 4 | 78.02±0.06 | 78.64±0.05 | **79.93±0.05** | 78.19±0.05 | 79.1±0.05 | 78.69±0.06 | 78.83±0.05 | 78.78±0.05 |
| 5 | **88.81±0.05** | 88.69±0.05 | 83.0±0.05 | 78.55±0.05 | 81.38±0.05 | 86.49±0.06 | 84.75±0.06 | 77.84±0.07 |
| 6 | 62.17±0.1 | **70.18±0.08** | 64.55±0.08 | 62.69±0.08 | 64.34±0.08 | 67.19±0.09 | 62.77±0.06 | 61.79±0.06 |
| 7 | 26.92±0.13 | **34.65±0.08** | 30.26±0.07 | 30.39±0.07 | 30.49±0.07 | 27.14±0.12 | 28.36±0.04 | 33.2±0.05 |
| 8 | 79.32±0.07 | 80.48±0.07 | 85.69±0.05 | **86.03±0.05** | 85.45±0.06 | 83.95±0.07 | 83.33±0.06 | 82.19±0.06 |
| 9 | 72.54±0.05 | 73.13±0.05 | **74.56±0.05** | 73.77±0.04 | 74.3±0.04 | 72.03±0.05 | 70.54±0.04 | 72.52±0.05 |
| 10 | 66.4±0.07 | 69.0±0.06 | 68.77±0.06 | **70.14±0.06** | 69.78±0.06 | 66.71±0.07 | 66.65±0.05 | 68.16±0.06 |
| 11 | 37.84±0.07 | **48.14±0.06** | 43.93±0.07 | 44.18±0.07 | 43.5±0.06 | 38.22±0.07 | 40.41±0.08 | 46.43±0.07 |
| 12 | 66.82±0.08 | 64.25±0.08 | **73.29±0.06** | 65.47±0.08 | 71.97±0.07 | 65.6±0.09 | 40.94±0.04 | 40.01±0.04 |
| 13 | 78.91±0.05 | 67.4±0.07 | **79.76±0.07** | 64.25±0.04 | 66.2±0.04 | 59.88±0.17 | 52.64±0.03 | 48.16±0.05 |
| 14 | 77.38±0.09 | **77.69±0.07** | 68.46±0.08 | 69.54±0.1 | 69.46±0.11 | 56.93±0.26 | 60.03±0.06 | 63.78±0.11 |
| 15 | 87.24±0.08 | 90.3±0.06 | **94.38±0.05** | 94.05±0.04 | 93.91±0.05 | 90.98±0.06 | 92.03±0.05 | 92.43±0.06 |
| 16 | 82.96±0.02 | **83.4±0.02** | 82.06±0.02 | 80.23±0.02 | 82.83±0.02 | 83.22±0.02 | 79.75±0.02 | 78.29±0.02 |
| 17 | 60.61±0.04 | **67.69±0.03** | 63.83±0.04 | 64.18±0.03 | 63.83±0.03 | 62.1±0.04 | 61.84±0.04 | 64.46±0.04 |
| 18 | 52.86±0.05 | **65.51±0.04** | 63.14±0.03 | 63.75±0.03 | 63.62±0.03 | 57.85±0.05 | 62.39±0.04 | 62.75±0.04 |
| 19 | 48.44±0.05 | **64.48±0.03** | 59.51±0.04 | 61.26±0.03 | 59.98±0.03 | 53.79±0.04 | 61.01±0.04 | 60.92±0.03 |
| 20 | 60.02±0.06 | **61.87±0.06** | 52.3±0.04 | 48.07±0.04 | 56.01±0.05 | 60.92±0.07 | 47.18±0.04 | 51.33±0.05 |
| 21 | **83.32±0.04** | 82.83±0.04 | 70.91±0.04 | 68.01±0.04 | 77.09±0.04 | 83.17±0.05 | 73.54±0.05 | 72.73±0.04 |
| 22 | 41.13±0.09 | **43.72±0.09** | 39.69±0.06 | 38.77±0.06 | 41.99±0.07 | 41.09±0.09 | 40.14±0.06 | 39.82±0.06 |
| 23 | 38.2±0.14 | 48.45±0.09 | 47.46±0.07 | 46.0±0.06 | **49.31±0.07** | 48.85±0.1 | 47.4±0.05 | 48.01±0.05 |
| 24 | 9.2±0.15 | 20.45±0.1 | 27.28±0.08 | 24.81±0.09 | **31.51±0.1** | 17.44±0.17 | 24.3±0.04 | 22.71±0.05 |
| 25 | 2.66±0.08 | 18.92±0.09 | 22.55±0.07 | 22.99±0.07 | 18.76±0.09 | 6.78±0.12 | **24.16±0.04** | 22.47±0.04 |
| 26 | 20.34±0.19 | 34.85±0.1 | 40.83±0.08 | 37.38±0.09 | **45.5±0.09** | 35.28±0.17 | 37.91±0.07 | 35.26±0.07 |
| 27 | 77.69±0.06 | 77.68±0.06 | 74.43±0.07 | 73.05±0.07 | 74.72±0.07 | **80.08±0.06** | 74.93±0.06 | 75.75±0.07 |
| 28 | 54.17±0.03 | **61.14±0.03** | 57.53±0.03 | 57.44±0.02 | 57.62±0.02 | 54.75±0.03 | 60.98±0.02 | 58.96±0.03 |
| 29 | 76.07±0.03 | **78.14±0.04** | 71.86±0.03 | 69.83±0.04 | 70.42±0.04 | 76.82±0.04 | 67.64±0.03 | 69.79±0.04 |
| 30 | 25.64±0.12 | 40.62±0.09 | 37.77±0.07 | 37.52±0.07 | **42.05±0.08** | 33.46±0.13 | 35.17±0.04 | 36.27±0.03 |
| 31 | 65.42±0.09 | **70.98±0.06** | 70.87±0.06 | 70.4±0.07 | 70.09±0.06 | 67.23±0.08 | 52.98±0.04 | 53.26±0.05 |
| 32 | 56.51±0.1 | 57.36±0.1 | 44.41±0.06 | 43.74±0.07 | 54.07±0.09 | **58.86±0.1** | 35.55±0.04 | 37.02±0.05 |

Table 13: **AUC** with **KNN** classifier on KEEL datasets

| Dataset | baseline | **LNR** | Over-sampling | | | Under-sampling | | |
|---|---|---|---|---|---|---|---|---|
| | | | SMOTE | ADASYN | Borderline | OSS | CC | RUS |
| 1 | 76.65±0.06 | 87.59±0.05 | 76.36±0.05 | 76.22±0.05 | 74.72±0.05 | 75.14±0.05 | **88.8±0.03** | 87.88±0.04 |
| 2 | 61.62±0.07 | **72.77±0.08** | 63.32±0.07 | 63.33±0.07 | 60.68±0.06 | 61.02±0.06 | 67.22±0.08 | 66.47±0.07 |
| 3 | 73.9±0.06 | **80.04±0.06** | 73.46±0.05 | 73.11±0.05 | 74.02±0.06 | 75.72±0.07 | 79.71±0.06 | 79.37±0.05 |
| 4 | 93.26±0.03 | 92.75±0.03 | 92.55±0.03 | 91.46±0.03 | 91.49±0.03 | 93.55±0.03 | **94.37±0.02** | 93.58±0.03 |
| 5 | 95.41±0.03 | 95.87±0.03 | 94.81±0.03 | 93.51±0.03 | 93.79±0.03 | 95.98±0.03 | **96.22±0.03** | 95.48±0.03 |
| 6 | 90.39±0.04 | 92.06±0.04 | 89.44±0.04 | 88.31±0.04 | 88.45±0.04 | 91.58±0.04 | **93.11±0.03** | 92.34±0.04 |
| 7 | 76.73±0.07 | 77.5±0.07 | 74.84±0.06 | 74.9±0.06 | 75.39±0.05 | 69.66±0.05 | 75.12±0.06 | **84.29±0.04** |
| 8 | 94.9±0.03 | 95.52±0.02 | 95.36±0.03 | 95.32±0.03 | 95.54±0.03 | 94.91±0.04 | **96.3±0.03** | 95.1±0.03 |
| 9 | 86.41±0.04 | 85.46±0.04 | **86.66±0.04** | 86.15±0.04 | 86.03±0.04 | 84.92±0.04 | 83.92±0.04 | 84.24±0.04 |
| 10 | **82.17±0.05** | 82.16±0.05 | 82.08±0.05 | 82.04±0.05 | 82.0±0.05 | 79.83±0.05 | 80.12±0.05 | 81.85±0.05 |
| 11 | 63.83±0.05 | **67.21±0.05** | 62.14±0.05 | 61.6±0.05 | 62.16±0.05 | 61.79±0.05 | 55.33±0.06 | 62.51±0.06 |
| 12 | 93.88±0.03 | 95.36±0.03 | 95.13±0.03 | 93.43±0.03 | 94.87±0.03 | 93.87±0.04 | **97.56±0.01** | 96.57±0.01 |
| 13 | 93.47±0.05 | 93.28±0.05 | 93.19±0.04 | 96.33±0.02 | 95.39±0.03 | 87.32±0.04 | **97.32±0.01** | 97.02±0.01 |
| 14 | 90.2±0.04 | 90.37±0.04 | 89.0±0.04 | 89.07±0.04 | 89.12±0.04 | 86.4±0.13 | **95.45±0.03** | 93.35±0.05 |
| 15 | 99.57±0.01 | 99.41±0.01 | **99.59±0.01** | 99.52±0.01 | 99.41±0.01 | 99.48±0.01 | 99.58±0.0 | 99.57±0.01 |
| 16 | 95.53±0.01 | 95.74±0.01 | 95.52±0.01 | 95.18±0.01 | 95.32±0.01 | 95.69±0.01 | **97.6±0.01** | 97.16±0.01 |
| 17 | 77.72±0.03 | **79.38±0.03** | 75.78±0.03 | 75.18±0.03 | 75.05±0.03 | 76.12±0.03 | 75.35±0.03 | 76.92±0.03 |
| 18 | 81.74±0.02 | **82.94±0.02** | 80.77±0.02 | 80.77±0.02 | 80.57±0.02 | 80.83±0.03 | 80.53±0.03 | 79.81±0.03 |
| 19 | 81.26±0.02 | **82.42±0.02** | 79.47±0.02 | 79.36±0.02 | 79.07±0.02 | 79.85±0.02 | 81.42±0.02 | 79.99±0.02 |
| 20 | 83.35±0.04 | **84.17±0.03** | 82.44±0.03 | 80.3±0.03 | 82.63±0.03 | 82.79±0.03 | 83.24±0.04 | 82.46±0.04 |
| 21 | 93.84±0.02 | 93.99±0.02 | 92.94±0.02 | 91.84±0.02 | 92.96±0.02 | 93.83±0.02 | **94.32±0.02** | 93.78±0.02 |
| 22 | 72.94±0.06 | 75.18±0.06 | 72.61±0.05 | 71.68±0.05 | 72.9±0.05 | 72.01±0.06 | **75.55±0.05** | 74.21±0.05 |
| 23 | 78.46±0.06 | 80.94±0.05 | 78.03±0.04 | 77.31±0.04 | 77.18±0.05 | 80.21±0.05 | **85.6±0.03** | 84.94±0.04 |
| 24 | 68.99±0.07 | 73.35±0.07 | 70.61±0.07 | 70.22±0.07 | 69.37±0.07 | 69.32±0.07 | **75.72±0.07** | 70.99±0.08 |
| 25 | 63.51±0.07 | 64.99±0.07 | 64.61±0.07 | 64.7±0.07 | 64.06±0.07 | 63.14±0.07 | **67.07±0.07** | 64.01±0.07 |
| 26 | 73.91±0.06 | 77.12±0.05 | 74.1±0.07 | 73.16±0.06 | 74.68±0.06 | 73.81±0.06 | **78.95±0.06** | 76.07±0.06 |
| 27 | 91.62±0.04 | 93.15±0.03 | 92.6±0.03 | 91.47±0.03 | 90.9±0.03 | 95.38±0.03 | **97.05±0.01** | 96.37±0.02 |
| 28 | 78.04±0.02 | **78.67±0.02** | 74.32±0.02 | 73.0±0.02 | 73.06±0.02 | 75.61±0.02 | 75.02±0.02 | 75.8±0.02 |
| 29 | 93.53±0.02 | 95.52±0.02 | 92.83±0.02 | 91.4±0.02 | 91.22±0.02 | 94.22±0.02 | **96.67±0.01** | 96.51±0.01 |
| 30 | 78.14±0.06 | 82.78±0.05 | 77.35±0.05 | 77.29±0.05 | 76.76±0.05 | 78.45±0.05 | **90.18±0.03** | 89.79±0.03 |
| 31 | 96.56±0.03 | **98.27±0.02** | 96.48±0.02 | 96.39±0.02 | 96.38±0.03 | 97.14±0.02 | 98.09±0.01 | 97.71±0.01 |
| 32 | 86.83±0.05 | 88.55±0.05 | 85.65±0.05 | 85.42±0.05 | 86.2±0.05 | 86.96±0.05 | **92.84±0.04** | 91.89±0.04 |

Table 14: **F1 score** with **CART** classifier on KEEL datasets

| Dataset | baseline | **LNR** | Over-sampling | | | Under-sampling | | |
|---|---|---|---|---|---|---|---|---|
| | | | SMOTE | ADASYN | Borderline | OSS | CC | RUS |
| 1 | 28.61±0.09 | **35.51±0.07** | 24.22±0.08 | 23.36±0.07 | 24.12±0.09 | 26.04±0.08 | 18.45±0.04 | 16.03±0.03 |
| 2 | 6.97±0.08 | **10.32±0.06** | 8.76±0.07 | 8.89±0.07 | 7.08±0.07 | 6.58±0.06 | 7.97±0.02 | 7.26±0.02 |
| 3 | 33.29±0.11 | **36.29±0.08** | 28.07±0.09 | 26.85±0.09 | 31.68±0.1 | 31.59±0.1 | 23.55±0.05 | 23.77±0.06 |
| 4 | 74.28±0.06 | **76.71±0.06** | 75.73±0.07 | 75.55±0.06 | 74.29±0.06 | 75.61±0.06 | 75.44±0.07 | 75.81±0.06 |
| 5 | 73.83±0.07 | 75.79±0.07 | **76.27±0.06** | 74.05±0.07 | 73.94±0.08 | 71.92±0.08 | 67.61±0.08 | 66.14±0.08 |
| 6 | 55.97±0.13 | **60.24±0.1** | 51.71±0.12 | 52.56±0.12 | 51.27±0.12 | 55.72±0.12 | 59.2±0.09 | 56.66±0.09 |
| 7 | 23.25±0.1 | **32.35±0.09** | 20.44±0.1 | 20.27±0.09 | 22.05±0.09 | 24.11±0.1 | 10.35±0.03 | 24.31±0.06 |
| 8 | 85.14±0.06 | **87.58±0.06** | 85.28±0.06 | 83.98±0.07 | 85.35±0.07 | 78.33±0.1 | 83.58±0.07 | 84.74±0.06 |
| 9 | 70.88±0.06 | 71.61±0.07 | 71.51±0.07 | 70.53±0.06 | 72.13±0.06 | **72.59±0.06** | 68.87±0.07 | 71.13±0.07 |
| 10 | 63.32±0.08 | 65.68±0.08 | 63.55±0.08 | **66.17±0.07** | 65.24±0.08 | 65.11±0.07 | 65.4±0.07 | 64.12±0.08 |
| 11 | 37.15±0.07 | **48.8±0.06** | 37.94±0.08 | 37.56±0.08 | 38.18±0.08 | 40.92±0.07 | 40.85±0.06 | 44.37±0.07 |
| 12 | **78.36±0.06** | 73.98±0.05 | 71.32±0.06 | 71.39±0.06 | 70.56±0.06 | 76.65±0.08 | 33.87±0.09 | 50.86±0.08 |
| 13 | **82.91±0.05** | 70.2±0.06 | 63.46±0.06 | 61.72±0.05 | 63.16±0.05 | 67.22±0.16 | 58.32±0.04 | 48.73±0.06 |
| 14 | 76.99±0.07 | 74.71±0.08 | **77.3±0.07** | 76.97±0.08 | 77.11±0.08 | 75.18±0.08 | 46.9±0.08 | 56.99±0.1 |
| 15 | 87.68±0.06 | 87.86±0.07 | 92.27±0.06 | **92.55±0.05** | 92.16±0.05 | 86.99±0.1 | 82.02±0.1 | 86.41±0.09 |
| 16 | 83.7±0.02 | **84.6±0.02** | 83.12±0.02 | 80.7±0.02 | 82.45±0.02 | 82.8±0.03 | 66.82±0.04 | 76.24±0.03 |
| 17 | 57.01±0.05 | **64.35±0.03** | 58.38±0.04 | 57.49±0.04 | 57.05±0.04 | 58.52±0.04 | 56.46±0.04 | 60.28±0.04 |
| 18 | 51.85±0.04 | **58.73±0.04** | 53.51±0.04 | 53.66±0.05 | 52.78±0.05 | 53.35±0.04 | 54.65±0.04 | 57.13±0.04 |
| 19 | 51.7±0.05 | **59.4±0.04** | 52.88±0.05 | 52.61±0.05 | 52.45±0.05 | 53.62±0.04 | 54.13±0.05 | 55.18±0.04 |
| 20 | 49.01±0.07 | **54.53±0.06** | 48.76±0.07 | 45.45±0.06 | 48.47±0.08 | 47.45±0.07 | 27.68±0.05 | 38.24±0.05 |
| 21 | 73.66±0.05 | **75.13±0.05** | 74.73±0.06 | 72.41±0.06 | 74.15±0.05 | 69.46±0.06 | 37.73±0.07 | 57.73±0.07 |
| 22 | 35.41±0.1 | **36.02±0.08** | 31.54±0.1 | 28.9±0.08 | 30.17±0.09 | 31.76±0.1 | 24.05±0.04 | 29.58±0.05 |
| 23 | 40.75±0.09 | **47.78±0.08** | 42.61±0.09 | 40.01±0.1 | 41.73±0.09 | 43.59±0.09 | 32.41±0.06 | 38.92±0.07 |
| 24 | **23.69±0.12** | 19.1±0.09 | 14.33±0.08 | 14.97±0.09 | 17.7±0.1 | 21.09±0.1 | 9.9±0.02 | 11.63±0.04 |
| 25 | 11.82±0.09 | **13.91±0.08** | 11.12±0.07 | 10.42±0.08 | 12.74±0.08 | 13.22±0.08 | 12.17±0.02 | 12.92±0.03 |
| 26 | **37.61±0.12** | 32.78±0.1 | 27.59±0.11 | 24.68±0.12 | 28.67±0.11 | 33.57±0.11 | 23.85±0.04 | 24.35±0.08 |
| 27 | 72.04±0.07 | 72.59±0.07 | **73.13±0.08** | 70.52±0.08 | 70.54±0.08 | 67.53±0.07 | 68.77±0.08 | 72.73±0.07 |
| 28 | 50.46±0.03 | **57.79±0.03** | 50.98±0.03 | 50.89±0.03 | 50.8±0.04 | 51.74±0.03 | 53.51±0.03 | 52.86±0.03 |
| 29 | 68.73±0.05 | **76.47±0.04** | 71.24±0.05 | 69.79±0.04 | 69.76±0.05 | 70.0±0.05 | 62.36±0.06 | 67.41±0.04 |
| 30 | 30.4±0.09 | **36.97±0.08** | 30.26±0.09 | 28.75±0.09 | 28.85±0.09 | 30.76±0.09 | 16.71±0.06 | 24.04±0.04 |
| 31 | 66.19±0.1 | 68.31±0.08 | 68.48±0.09 | 69.24±0.09 | **69.48±0.08** | 61.56±0.11 | 55.91±0.1 | 51.74±0.1 |
| 32 | 41.96±0.11 | **47.48±0.09** | 44.67±0.11 | 40.48±0.11 | 45.22±0.11 | 38.11±0.11 | 11.37±0.06 | 20.25±0.05 |

Table 15: **G-mean** with **CART** classifier on KEEL datasets

| Dataset | baseline | LNR | Over-sampling | | | Under-sampling | | |
|---|---|---|---|---|---|---|---|---|
| | | | SMOTE | ADASYN | Borderline | OSS | CC | RUS |
| 1 | 28.91±0.09 | **37.02±0.08** | 24.49±0.08 | 23.64±0.07 | 24.34±0.09 | 26.53±0.08 | 26.61±0.04 | 25.34±0.04 |
| 2 | 7.07±0.08 | 12.23±0.07 | 9.05±0.07 | 9.22±0.07 | 7.23±0.08 | 6.82±0.06 | 14.67±0.04 | **15.17±0.04** |
| 3 | 33.71±0.11 | **37.26±0.09** | 28.36±0.09 | 27.08±0.09 | 32.23±0.1 | 32.46±0.1 | 28.62±0.06 | 30.74±0.07 |
| 4 | 74.51±0.06 | **77.06±0.06** | 75.9±0.07 | 75.76±0.06 | 74.49±0.06 | 75.85±0.06 | 75.81±0.07 | 76.35±0.06 |
| 5 | 74.25±0.07 | 76.23±0.06 | **76.63±0.06** | 74.39±0.07 | 74.23±0.08 | 72.82±0.07 | 68.85±0.08 | 68.13±0.07 |
| 6 | 56.92±0.13 | 60.96±0.1 | 52.37±0.12 | 53.25±0.12 | 51.95±0.12 | 56.59±0.12 | **60.99±0.09** | 59.45±0.08 |
| 7 | 25.63±0.1 | **34.26±0.09** | 21.25±0.1 | 20.93±0.1 | 22.74±0.09 | 25.93±0.1 | 21.69±0.03 | 32.67±0.05 |
| 8 | 85.49±0.06 | **87.81±0.06** | 85.59±0.06 | 84.28±0.07 | 85.63±0.07 | 78.97±0.09 | 83.89±0.07 | 85.05±0.06 |
| 9 | 71.18±0.06 | 72.02±0.07 | 71.81±0.06 | 70.82±0.06 | 72.54±0.06 | **72.88±0.06** | 69.45±0.07 | 71.58±0.07 |
| 10 | 63.62±0.08 | 65.99±0.08 | 63.88±0.08 | **66.48±0.07** | 65.51±0.08 | 65.62±0.07 | 65.82±0.07 | 64.5±0.08 |
| 11 | 37.38±0.07 | **49.54±0.06** | 38.1±0.08 | 37.72±0.07 | 38.36±0.08 | 41.32±0.07 | 42.43±0.06 | 45.59±0.07 |
| 12 | **78.88±0.05** | 74.87±0.04 | 71.95±0.06 | 71.98±0.06 | 71.24±0.06 | 77.19±0.07 | 44.89±0.07 | 58.07±0.06 |
| 13 | **84.21±0.04** | 70.64±0.06 | 66.67±0.04 | 65.64±0.04 | 66.7±0.04 | 69.37±0.15 | 64.16±0.03 | 56.51±0.05 |
| 14 | 77.31±0.07 | 75.22±0.08 | **77.67±0.07** | 77.37±0.07 | 77.49±0.08 | 75.77±0.08 | 53.31±0.06 | 60.6±0.08 |
| 15 | 87.95±0.06 | 88.16±0.06 | 92.41±0.06 | **92.71±0.05** | 92.34±0.05 | 87.68±0.09 | 83.41±0.09 | 86.92±0.08 |
| 16 | 83.74±0.02 | **84.7±0.02** | 83.16±0.02 | 80.74±0.02 | 82.47±0.02 | 82.91±0.03 | 69.36±0.03 | 77.57±0.03 |
| 17 | 57.1±0.05 | **64.96±0.03** | 58.46±0.04 | 57.59±0.04 | 57.13±0.04 | 58.69±0.04 | 56.67±0.04 | 60.61±0.04 |
| 18 | 51.97±0.04 | **59.44±0.05** | 53.58±0.04 | 53.73±0.05 | 52.89±0.05 | 53.54±0.04 | 55.72±0.04 | 58.26±0.04 |
| 19 | 51.81±0.05 | **60.38±0.04** | 52.98±0.05 | 52.71±0.05 | 52.54±0.05 | 53.86±0.04 | 55.31±0.05 | 56.34±0.04 |
| 20 | 49.32±0.07 | **55.0±0.06** | 49.06±0.07 | 45.71±0.06 | 48.71±0.06 | 47.83±0.07 | 35.91±0.05 | 43.62±0.04 |
| 21 | 73.83±0.05 | **75.5±0.05** | 74.92±0.06 | 72.64±0.06 | 74.31±0.05 | 69.98±0.06 | 46.06±0.05 | 61.46±0.06 |
| 22 | 35.73±0.1 | **36.82±0.08** | 31.91±0.1 | 29.26±0.08 | 30.45±0.09 | 32.25±0.1 | 32.9±0.05 | 34.84±0.05 |
| 23 | 41.2±0.09 | **48.5±0.08** | 43.01±0.09 | 40.4±0.1 | 42.16±0.09 | 44.25±0.09 | 39.88±0.06 | 43.41±0.07 |
| 24 | **24.18±0.12** | 20.31±0.09 | 14.63±0.08 | 15.32±0.09 | 18.05±0.1 | 21.76±0.1 | 19.18±0.04 | 19.26±0.05 |
| 25 | 12.08±0.09 | 15.06±0.08 | 11.37±0.08 | 10.62±0.08 | 12.92±0.08 | 13.73±0.09 | **21.99±0.04** | 20.6±0.05 |
| 26 | **38.18±0.12** | 33.67±0.1 | 28.07±0.11 | 24.99±0.12 | 29.19±0.11 | 34.46±0.11 | 32.44±0.05 | 30.63±0.08 |
| 27 | 72.45±0.07 | 73.01±0.07 | **73.55±0.08** | 70.9±0.08 | 71.1±0.07 | 70.52±0.06 | 70.89±0.07 | 73.09±0.07 |
| 28 | 50.5±0.03 | **58.81±0.03** | 51.04±0.03 | 50.95±0.03 | 50.84±0.04 | 51.87±0.03 | 55.16±0.03 | 53.73±0.03 |
| 29 | 68.86±0.05 | **76.95±0.04** | 71.36±0.05 | 69.93±0.04 | 69.86±0.05 | 70.2±0.05 | 65.7±0.05 | 69.63±0.03 |
| 30 | 30.8±0.09 | **38.13±0.08** | 30.59±0.09 | 29.08±0.09 | 29.19±0.1 | 31.16±0.09 | 27.67±0.06 | 33.14±0.05 |
| 31 | 66.73±0.1 | 69.04±0.08 | 68.91±0.09 | 69.71±0.08 | **70.01±0.08** | 62.22±0.11 | 60.96±0.08 | 58.02±0.08 |
| 32 | 42.54±0.11 | **48.97±0.09** | 45.49±0.11 | 41.12±0.11 | 45.83±0.11 | 39.04±0.12 | 22.69±0.05 | 30.39±0.05 |

Table 16: **AUC** with **CART** classifier on KEEL datasets

| Dataset | baseline | LNR | Over-sampling | | | Under-sampling | | |
|---|---|---|---|---|---|---|---|---|
| | | | SMOTE | ADASYN | Borderline | OSS | CC | RUS |
| 1 | 62.85±0.05 | 72.47±0.06 | 61.97±0.05 | 61.45±0.05 | 60.98±0.05 | 63.12±0.05 | 74.83±0.05 | **74.89±0.05** |
| 2 | 52.45±0.04 | 57.64±0.06 | 53.92±0.04 | 54.09±0.05 | 52.6±0.04 | 52.54±0.04 | 61.73±0.08 | **61.8±0.08** |
| 3 | 63.71±0.06 | **68.64±0.06** | 61.82±0.05 | 61.03±0.05 | 63.05±0.05 | 65.25±0.06 | 66.11±0.06 | 68.44±0.07 |
| 4 | 82.33±0.04 | 85.27±0.04 | 83.57±0.05 | 83.57±0.05 | 82.46±0.04 | 84.36±0.05 | 84.64±0.05 | **85.42±0.04** |
| 5 | 84.23±0.05 | **86.4±0.05** | 86.19±0.05 | 84.76±0.05 | 84.52±0.06 | 85.83±0.05 | 84.06±0.05 | 84.26±0.05 |
| 6 | 74.04±0.08 | 79.37±0.07 | 72.69±0.08 | 73.06±0.07 | 72.03±0.07 | 77.02±0.08 | **82.51±0.06** | 82.4±0.06 |
| 7 | 64.23±0.09 | 73.74±0.1 | 63.72±0.09 | 63.35±0.1 | 63.77±0.1 | 64.48±0.09 | 55.67±0.08 | **77.43±0.07** |
| 8 | 89.1±0.05 | **91.81±0.05** | 89.53±0.05 | 88.7±0.06 | 89.63±0.05 | 86.07±0.07 | 88.26±0.05 | 90.62±0.05 |
| 9 | 77.11±0.05 | 77.5±0.05 | 77.59±0.05 | 76.83±0.05 | 78.14±0.05 | **78.31±0.05** | 74.78±0.06 | 77.03±0.05 |
| 10 | 70.72±0.06 | 71.97±0.06 | 70.76±0.06 | **72.71±0.05** | 72.04±0.06 | 70.86±0.06 | 71.37±0.06 | 70.23±0.07 |
| 11 | 56.22±0.04 | **62.71±0.05** | 56.45±0.05 | 56.22±0.05 | 56.59±0.05 | 57.06±0.05 | 53.7±0.05 | 57.83±0.06 |
| 12 | 95.71±0.03 | **97.0±0.03** | 96.17±0.03 | 96.09±0.03 | 96.11±0.03 | 96.1±0.03 | 92.78±0.02 | 96.42±0.02 |
| 13 | **99.29±0.0** | 98.82±0.0 | 99.23±0.0 | 99.15±0.01 | 99.2±0.0 | 98.45±0.01 | 99.01±0.0 | 97.55±0.01 |
| 14 | 88.28±0.05 | 87.94±0.05 | **88.79±0.05** | 88.77±0.05 | 88.61±0.06 | 88.6±0.05 | 82.94±0.05 | 85.71±0.05 |
| 15 | 91.59±0.04 | 93.18±0.04 | 94.8±0.04 | **94.98±0.04** | 94.61±0.04 | 94.4±0.04 | 93.5±0.04 | 93.84±0.05 |
| 16 | 90.6±0.01 | 93.01±0.01 | 91.32±0.01 | 89.96±0.01 | 90.41±0.02 | 91.79±0.01 | 90.53±0.01 | **93.48±0.01** |
| 17 | 66.32±0.03 | **70.74±0.03** | 67.02±0.03 | 66.01±0.03 | 65.8±0.03 | 66.59±0.03 | 64.58±0.03 | 67.5±0.03 |
| 18 | 66.9±0.03 | **72.06±0.03** | 67.99±0.03 | 68.09±0.03 | 67.56±0.04 | 67.8±0.03 | 68.71±0.03 | 70.83±0.03 |
| 19 | 67.22±0.03 | **73.32±0.04** | 68.0±0.04 | 67.79±0.03 | 67.69±0.03 | 68.49±0.03 | 68.91±0.04 | 69.81±0.03 |
| 20 | 71.39±0.04 | **76.59±0.04** | 72.18±0.04 | 70.06±0.04 | 71.37±0.04 | 72.02±0.05 | 64.33±0.04 | 73.68±0.04 |
| 21 | 85.67±0.04 | **88.61±0.03** | 86.26±0.04 | 85.69±0.04 | 85.83±0.04 | 85.66±0.04 | 76.17±0.04 | 86.43±0.03 |
| 22 | 63.38±0.05 | **65.27±0.06** | 61.89±0.06 | 60.26±0.05 | 60.77±0.05 | 62.24±0.06 | 57.03±0.07 | 63.97±0.05 |
| 23 | 66.57±0.06 | 72.64±0.06 | 68.17±0.05 | 66.89±0.07 | 67.37±0.06 | 70.04±0.06 | 69.05±0.06 | **72.65±0.06** |
| 24 | 60.66±0.07 | 59.76±0.06 | 56.05±0.05 | 56.64±0.06 | 57.45±0.06 | 60.67±0.06 | 59.3±0.07 | **61.3±0.08** |
| 25 | 53.5±0.05 | 55.84±0.06 | 53.33±0.05 | 52.78±0.05 | 54.03±0.05 | 54.68±0.05 | 57.72±0.06 | **58.25±0.07** |
| 26 | 65.67±0.06 | 65.17±0.07 | 61.09±0.06 | 59.37±0.07 | 61.23±0.06 | 65.56±0.07 | **66.99±0.06** | 64.76±0.08 |
| 27 | 83.21±0.05 | 84.5±0.05 | 84.85±0.06 | 83.08±0.05 | 84.98±0.05 | **91.27±0.04** | 90.02±0.04 | 85.93±0.05 |
| 28 | 64.57±0.02 | **69.38±0.02** | 64.82±0.02 | 64.68±0.02 | 64.66±0.03 | 65.09±0.02 | 65.14±0.02 | 65.11±0.03 |
| 29 | 82.02±0.03 | **90.02±0.03** | 84.43±0.03 | 83.7±0.03 | 83.03±0.04 | 84.29±0.03 | 88.34±0.02 | 89.59±0.02 |
| 30 | 63.47±0.05 | 71.58±0.05 | 64.71±0.04 | 63.99±0.05 | 62.94±0.05 | 65.01±0.05 | 72.15±0.09 | **78.74±0.06** |
| 31 | 81.08±0.06 | 87.63±0.06 | 83.63±0.06 | 84.11±0.06 | 83.94±0.06 | 82.18±0.07 | 93.35±0.04 | **93.72±0.03** |
| 32 | 70.93±0.06 | 78.97±0.07 | 74.56±0.07 | 71.76±0.07 | 73.45±0.07 | 71.48±0.07 | 71.71±0.05 | **81.01±0.05** |

Table 17: **F1 score** with **MLP** classifier on KEEL datasets

| Dataset | baseline | LNR | Over-sampling | | | Under-sampling | | |
| | | | SMOTE | ADASYN | Borderline | OSS | CC | RUS |
|---|---|---|---|---|---|---|---|---|
| 1 | 26.47±0.12 | **44.35±0.08** | 37.11±0.07 | 36.81±0.06 | 38.36±0.07 | 32.04±0.13 | 27.03±0.04 | 24.62±0.05 |
| 2 | 0.0±0.0 | **18.4±0.07** | 13.8±0.07 | 12.75±0.07 | 10.29±0.07 | 0.0±0.0 | 10.04±0.03 | 8.93±0.02 |
| 3 | 44.83±0.17 | **52.22±0.08** | 48.33±0.08 | 45.94±0.09 | 51.69±0.09 | 43.47±0.21 | 40.43±0.07 | 36.91±0.06 |
| 4 | 75.16±0.07 | 76.18±0.06 | **77.05±0.05** | 76.63±0.06 | 76.75±0.05 | 75.48±0.07 | 76.83±0.04 | 76.13±0.05 |
| 5 | 82.01±0.06 | **82.46±0.06** | 81.27±0.07 | 76.19±0.07 | 76.7±0.08 | 80.06±0.06 | 78.89±0.06 | 75.15±0.07 |
| 6 | 40.44±0.31 | **66.92±0.08** | 66.48±0.09 | 66.17±0.08 | 66.07±0.09 | 60.85±0.11 | 61.47±0.07 | 59.16±0.08 |
| 7 | 21.49±0.11 | **31.02±0.1** | 27.75±0.06 | 26.45±0.07 | 27.55±0.07 | 20.89±0.12 | 16.94±0.06 | 25.17±0.05 |
| 8 | 86.35±0.05 | 86.15±0.06 | 87.07±0.06 | 87.02±0.06 | **88.18±0.05** | 83.64±0.07 | 85.72±0.06 | 87.44±0.05 |
| 9 | 51.91±0.31 | 71.02±0.08 | 69.46±0.1 | **72.28±0.07** | 69.79±0.1 | 59.72±0.27 | 69.43±0.07 | 70.44±0.06 |
| 10 | 63.87±0.07 | 64.83±0.06 | 64.16±0.07 | 65.06±0.07 | **66.22±0.07** | 63.52±0.08 | 64.27±0.07 | 65.75±0.07 |
| 11 | 29.54±0.12 | **50.44±0.07** | 48.06±0.08 | 48.31±0.07 | 48.0±0.08 | 43.26±0.08 | 40.48±0.07 | 47.47±0.08 |
| 12 | **68.6±0.07** | 66.18±0.05 | 63.01±0.06 | 62.55±0.05 | 64.46±0.05 | 67.3±0.09 | 50.6±0.05 | 45.61±0.07 |
| 13 | **79.55±0.05** | 70.58±0.07 | 53.64±0.05 | 55.36±0.06 | 56.74±0.05 | 69.39±0.16 | 51.48±0.06 | 49.38±0.06 |
| 14 | 45.53±0.34 | 70.19±0.09 | 72.12±0.09 | 70.46±0.1 | 71.65±0.1 | **73.86±0.11** | 58.99±0.1 | 56.48±0.11 |
| 15 | 94.3±0.04 | 95.24±0.04 | 95.02±0.03 | 95.31±0.04 | **95.4±0.03** | 93.34±0.05 | 93.31±0.05 | 94.07±0.05 |
| 16 | 82.79±0.02 | 83.65±0.02 | 81.57±0.02 | 76.76±0.03 | 80.04±0.02 | **84.15±0.02** | 77.06±0.03 | 77.85±0.02 |
| 17 | 61.88±0.04 | **67.36±0.03** | 64.76±0.04 | 64.84±0.03 | 64.85±0.03 | 63.81±0.04 | 61.75±0.04 | 64.79±0.04 |
| 18 | 71.16±0.04 | **73.2±0.03** | 72.2±0.04 | 71.87±0.03 | 71.46±0.03 | 71.74±0.04 | 71.43±0.03 | 70.93±0.04 |
| 19 | 62.11±0.05 | **67.23±0.04** | 66.9±0.04 | 66.66±0.04 | 67.18±0.04 | 62.68±0.08 | 65.18±0.04 | 64.96±0.04 |
| 20 | 54.87±0.1 | **59.41±0.06** | 47.81±0.06 | 40.61±0.06 | 50.05±0.08 | 55.88±0.08 | 37.26±0.05 | 38.22±0.05 |
| 21 | 78.8±0.06 | **79.8±0.05** | 72.59±0.05 | 65.8±0.08 | 69.04±0.1 | 78.71±0.05 | 59.3±0.08 | 65.77±0.09 |
| 22 | 32.23±0.1 | **40.73±0.09** | 35.86±0.08 | 31.63±0.09 | 38.56±0.1 | 31.22±0.13 | 31.61±0.06 | 30.49±0.07 |
| 23 | 44.73±0.12 | **50.94±0.08** | 48.17±0.1 | 47.13±0.1 | 48.43±0.09 | 45.08±0.13 | 41.05±0.06 | 38.67±0.08 |
| 24 | 0.0±0.0 | 16.27±0.08 | 16.75±0.09 | 15.27±0.09 | **22.28±0.13** | 1.5±0.04 | 12.14±0.04 | 11.28±0.04 |
| 25 | 0.0±0.0 | **18.43±0.09** | 14.18±0.08 | 14.0±0.08 | 13.09±0.09 | 0.1±0.01 | 15.63±0.05 | 13.42±0.03 |
| 26 | 8.76±0.15 | 33.8±0.09 | 29.75±0.08 | **39.72±0.1** | 25.94±0.14 | 30.1±0.06 | 26.53±0.07 | 28.76±0.11 |
| 27 | 69.77±0.08 | 71.14±0.08 | 68.47±0.07 | **71.24±0.08** | 70.84±0.08 | 67.36±0.08 | 64.6±0.07 | 61.41±0.08 |
| 28 | 55.49±0.03 | **60.34±0.03** | 59.53±0.03 | 59.0±0.02 | 59.16±0.02 | 58.09±0.03 | 58.9±0.02 | 59.75±0.02 |
| 29 | 78.32±0.04 | **78.37±0.04** | 75.25±0.03 | 73.47±0.04 | 75.53±0.04 | 78.23±0.04 | 69.79±0.04 | 69.39±0.04 |
| 30 | 10.44±0.13 | **41.68±0.07** | 36.45±0.08 | 36.55±0.07 | 38.78±0.07 | 19.53±0.15 | 24.72±0.04 | 22.67±0.04 |
| 31 | 60.93±0.11 | 66.58±0.08 | **74.58±0.07** | 74.19±0.07 | 73.77±0.08 | 67.37±0.1 | 52.98±0.08 | 45.92±0.07 |
| 32 | 49.7±0.16 | **58.02±0.1** | 46.87±0.1 | 45.48±0.1 | 54.62±0.11 | 49.75±0.15 | 26.67±0.07 | 22.41±0.05 |

Table 18: **G-mean** with **MLP** classifier on KEEL datasets

| Dataset | baseline | LNR | Over-sampling | | | Under-sampling | | |
| | | | SMOTE | ADASYN | Borderline | OSS | CC | RUS |
|---|---|---|---|---|---|---|---|---|
| 1 | 31.86±0.13 | **45.44±0.08** | 40.6±0.07 | 40.53±0.06 | 40.25±0.07 | 35.08±0.14 | 35.94±0.04 | 34.36±0.04 |
| 2 | 0.0±0.0 | **20.99±0.08** | 15.88±0.08 | 14.55±0.08 | 10.62±0.08 | 0.0±0.0 | 17.33±0.04 | 17.66±0.04 |
| 3 | 47.07±0.18 | **53.11±0.08** | 49.47±0.08 | 47.01±0.09 | 52.21±0.09 | 44.74±0.22 | 46.3±0.06 | 43.42±0.06 |
| 4 | 75.52±0.07 | 76.73±0.06 | 77.5±0.05 | 77.25±0.06 | 77.44±0.05 | 75.77±0.07 | **77.63±0.04** | 77.16±0.05 |
| 5 | 82.24±0.06 | **82.73±0.06** | 81.55±0.07 | 76.7±0.07 | 77.11±0.08 | 80.42±0.06 | 79.6±0.06 | 76.35±0.07 |
| 6 | 40.92±0.31 | **67.82±0.08** | 67.24±0.08 | 66.9±0.08 | 66.79±0.09 | 61.3±0.11 | 63.87±0.07 | 62.48±0.07 |
| 7 | 26.88±0.12 | 32.6±0.1 | 30.15±0.07 | 28.9±0.07 | 29.85±0.08 | 25.56±0.13 | 24.64±0.06 | **33.28±0.05** |
| 8 | 86.56±0.05 | 86.37±0.06 | 87.28±0.06 | 87.22±0.06 | **88.35±0.05** | 84.04±0.07 | 85.92±0.05 | 87.74±0.05 |
| 9 | 52.19±0.32 | 71.64±0.08 | 70.09±0.1 | **73.03±0.07** | 70.7±0.1 | 60.08±0.27 | 70.03±0.07 | 70.98±0.06 |
| 10 | 64.33±0.07 | 65.13±0.06 | 64.42±0.07 | 65.51±0.07 | **66.52±0.07** | 64.05±0.07 | 64.77±0.06 | 66.14±0.07 |
| 11 | 34.05±0.11 | **50.69±0.07** | 48.41±0.08 | 48.85±0.07 | 48.38±0.07 | 44.05±0.08 | 42.18±0.07 | 48.17±0.08 |
| 12 | **69.45±0.06** | 66.84±0.05 | 66.88±0.05 | 65.98±0.04 | 68.09±0.04 | 68.24±0.09 | 57.81±0.04 | 53.89±0.05 |
| 13 | **81.33±0.04** | 70.94±0.06 | 59.41±0.04 | 60.61±0.04 | 61.87±0.04 | 71.22±0.15 | 57.92±0.04 | 56.22±0.04 |
| 14 | 45.94±0.34 | 70.61±0.09 | 72.67±0.09 | 71.04±0.1 | 72.16±0.1 | **74.31±0.11** | 61.41±0.09 | 59.95±0.1 |
| 15 | 94.42±0.04 | 95.41±0.04 | 95.11±0.03 | 95.41±0.04 | **95.49±0.03** | 93.52±0.05 | 93.59±0.04 | 94.28±0.05 |
| 16 | 82.97±0.02 | 83.69±0.02 | 82.29±0.02 | 78.08±0.03 | 80.81±0.02 | **84.2±0.02** | 78.03±0.02 | 79.13±0.02 |
| 17 | 62.12±0.04 | **67.97±0.03** | 65.02±0.04 | 65.3±0.03 | 65.25±0.03 | 63.95±0.04 | 61.94±0.04 | 65.25±0.04 |
| 18 | 71.28±0.04 | **73.91±0.03** | 72.49±0.04 | 72.3±0.03 | 71.91±0.03 | 72.04±0.04 | 72.0±0.03 | 71.73±0.04 |
| 19 | 62.31±0.05 | **68.17±0.04** | 67.31±0.04 | 67.1±0.04 | 67.65±0.04 | 63.07±0.07 | 66.0±0.04 | 65.98±0.04 |
| 20 | 56.8±0.09 | **59.73±0.06** | 48.89±0.06 | 42.18±0.06 | 50.57±0.07 | 57.06±0.08 | 41.4±0.04 | 43.0±0.05 |
| 21 | 79.19±0.05 | **79.94±0.05** | 72.92±0.05 | 66.3±0.08 | 69.36±0.1 | 79.01±0.05 | 61.82±0.07 | 67.78±0.08 |
| 22 | 37.12±0.09 | **41.32±0.09** | 36.55±0.08 | 32.3±0.1 | 39.21±0.1 | 32.61±0.13 | 35.08±0.07 | 34.97±0.07 |
| 23 | 46.79±0.12 | **51.6±0.08** | 48.65±0.1 | 47.58±0.1 | 48.87±0.09 | 46.21±0.12 | 44.05±0.06 | 42.76±0.07 |
| 24 | 0.0±0.0 | 18.09±0.09 | 17.89±0.09 | 16.36±0.09 | **23.63±0.13** | 1.79±0.05 | 19.57±0.05 | 18.63±0.05 |
| 25 | 0.0±0.0 | 19.33±0.09 | 14.83±0.08 | 14.61±0.04 | 13.9±0.1 | 0.1±0.01 | **21.75±0.06** | 20.38±0.05 |
| 26 | 9.76±0.16 | 36.43±0.1 | 31.83±0.08 | **41.08±0.11** | 26.77±0.15 | 36.99±0.06 | 33.3±0.07 | 29.85±0.11 |
| 27 | 70.63±0.08 | 71.56±0.08 | 68.93±0.07 | **71.71±0.08** | 71.32±0.08 | 67.92±0.08 | 66.34±0.07 | 63.68±0.08 |
| 28 | 56.16±0.03 | 60.79±0.03 | 60.29±0.03 | 60.34±0.02 | 60.43±0.02 | 58.27±0.03 | **61.04±0.02** | 60.76±0.02 |
| 29 | 78.42±0.04 | **78.6±0.04** | 75.82±0.03 | 74.14±0.04 | 75.93±0.04 | 78.37±0.04 | 72.07±0.04 | 71.61±0.04 |
| 30 | 12.52±0.15 | **42.27±0.07** | 38.54±0.08 | 38.64±0.08 | 39.47±0.07 | 21.23±0.16 | 32.73±0.05 | 31.65±0.04 |
| 31 | 61.89±0.11 | 67.53±0.08 | **74.92±0.07** | 74.56±0.07 | 74.13±0.08 | 67.93±0.1 | 58.84±0.08 | 53.74±0.06 |
| 32 | 52.58±0.16 | **58.58±0.1** | 48.4±0.1 | 46.77±0.1 | 55.06±0.11 | 51.87±0.14 | 34.56±0.06 | 32.06±0.05 |

Table 19: **AUC** with **MLP** classifier on KEEL datasets

| Dataset | baseline | LNR | Over-sampling | | | Under-sampling | | |
| | | | SMOTE | ADASYN | Borderline | OSS | CC | RUS |
|---|---|---|---|---|---|---|---|---|
| 1 | **92.5±0.02** | 92.24±0.03 | 89.24±0.05 | 89.12±0.05 | 88.18±0.05 | 90.38±0.04 | 91.48±0.04 | 91.19±0.04 |
| 2 | **79.15±0.06** | 78.32±0.06 | 70.78±0.08 | 70.43±0.07 | 68.96±0.09 | 78.62±0.06 | 72.97±0.05 | 72.24±0.06 |
| 3 | **91.61±0.03** | 91.18±0.04 | 88.8±0.05 | 88.29±0.05 | 90.67±0.04 | 90.64±0.04 | 90.31±0.03 | 89.0±0.04 |
| 4 | 93.75±0.03 | 93.68±0.03 | **94.27±0.02** | 94.04±0.02 | 94.0±0.02 | 93.0±0.03 | 94.01±0.02 | 93.83±0.02 |
| 5 | 94.94±0.03 | **95.52±0.03** | 95.35±0.03 | 94.45±0.03 | 93.65±0.03 | 94.71±0.03 | 95.48±0.03 | 94.91±0.03 |
| 6 | **91.21±0.05** | 90.85±0.05 | 90.46±0.04 | 90.12±0.05 | 90.28±0.05 | 88.4±0.05 | 89.54±0.04 | 89.56±0.04 |
| 7 | **84.62±0.05** | 83.95±0.06 | 72.55±0.08 | 70.23±0.09 | 75.84±0.08 | 81.81±0.07 | 70.75±0.08 | 81.56±0.07 |
| 8 | 96.62±0.03 | 96.6±0.03 | 97.25±0.03 | 97.21±0.03 | **97.7±0.03** | 95.76±0.03 | 95.56±0.04 | 97.05±0.03 |
| 9 | 76.43±0.14 | 82.25±0.06 | 83.99±0.06 | **84.16±0.05** | 81.87±0.08 | 78.04±0.14 | 81.94±0.05 | 81.92±0.05 |
| 10 | 75.21±0.05 | 75.25±0.05 | 75.26±0.05 | 75.35±0.06 | **76.05±0.05** | 73.33±0.06 | 75.56±0.05 | 74.92±0.06 |
| 11 | 66.32±0.05 | **67.94±0.05** | 66.83±0.06 | 66.1±0.06 | 66.52±0.07 | 65.88±0.06 | 56.31±0.07 | 64.09±0.08 |
| 12 | 98.79±0.0 | 98.78±0.0 | **98.93±0.01** | 98.46±0.01 | 98.73±0.01 | 98.7±0.01 | 98.7±0.0 | 98.38±0.01 |
| 13 | 98.56±0.01 | 98.57±0.01 | **98.71±0.0** | 97.6±0.01 | 98.53±0.01 | 97.19±0.03 | 98.6±0.01 | 98.39±0.01 |
| 14 | 93.18±0.03 | 93.24±0.03 | 93.73±0.03 | **93.99±0.03** | 93.86±0.03 | 90.97±0.05 | 92.35±0.03 | 92.18±0.04 |
| 15 | **99.89±0.0** | 99.87±0.0 | 99.86±0.0 | 99.86±0.0 | 99.87±0.0 | 99.64±0.01 | **99.89±0.0** | 99.82±0.0 |
| 16 | 98.12±0.01 | 98.11±0.01 | **98.52±0.0** | 97.86±0.01 | 97.87±0.01 | 98.13±0.01 | 97.94±0.01 | 98.05±0.01 |
| 17 | 79.94±0.03 | **80.48±0.03** | 78.43±0.03 | 77.87±0.03 | 77.31±0.03 | 79.05±0.03 | 76.55±0.03 | 78.07±0.03 |
| 18 | 90.88±0.02 | 90.47±0.02 | **90.96±0.02** | 89.94±0.02 | 89.23±0.02 | 90.45±0.02 | 89.97±0.02 | 89.27±0.03 |
| 19 | **88.29±0.02** | 87.52±0.02 | 88.1±0.02 | 87.67±0.02 | 88.08±0.02 | 85.98±0.05 | 86.61±0.02 | 86.21±0.02 |
| 20 | **83.13±0.04** | 82.15±0.04 | 79.59±0.04 | 77.41±0.04 | 79.86±0.05 | 82.64±0.04 | 76.89±0.04 | 78.63±0.04 |
| 21 | **93.14±0.03** | 92.9±0.03 | 91.89±0.03 | 90.74±0.03 | 90.11±0.04 | 92.89±0.03 | 90.63±0.03 | 92.64±0.03 |
| 22 | **75.1±0.06** | 74.29±0.07 | 71.19±0.05 | 68.97±0.07 | 72.3±0.07 | 71.76±0.07 | 69.19±0.07 | 69.25±0.06 |
| 23 | 82.02±0.05 | 81.43±0.06 | **82.2±0.07** | 81.74±0.07 | 80.97±0.06 | 78.79±0.06 | 78.35±0.05 | 77.88±0.06 |
| 24 | 71.04±0.07 | **72.42±0.07** | 68.17±0.08 | 67.31±0.07 | 67.02±0.08 | 70.46±0.08 | 67.27±0.07 | 65.72±0.08 |
| 25 | 63.88±0.06 | 66.21±0.07 | 64.37±0.07 | 63.99±0.07 | 60.52±0.07 | 62.66±0.07 | **66.55±0.07** | 62.4±0.07 |
| 26 | 76.47±0.06 | 78.45±0.06 | 76.34±0.05 | 79.14±0.06 | 74.53±0.06 | **79.95±0.05** | 74.65±0.07 | 75.44±0.06 |
| 27 | 91.66±0.04 | 91.05±0.05 | 91.02±0.04 | 91.31±0.05 | 91.16±0.05 | 90.98±0.05 | **92.35±0.04** | 91.85±0.04 |
| 28 | **79.64±0.02** | 79.37±0.02 | 78.49±0.02 | 77.93±0.02 | 77.5±0.02 | 79.43±0.02 | 77.52±0.02 | 78.69±0.02 |
| 29 | 97.13±0.01 | **97.26±0.01** | 96.31±0.01 | 95.56±0.01 | 95.74±0.02 | 96.84±0.01 | 96.58±0.01 | 96.38±0.01 |
| 30 | **87.09±0.04** | 86.62±0.04 | 85.54±0.05 | 86.08±0.05 | 84.54±0.05 | 84.0±0.05 | 84.29±0.05 | 82.39±0.05 |
| 31 | 98.37±0.01 | 98.16±0.01 | **98.4±0.01** | 98.29±0.01 | 98.2±0.01 | 97.88±0.02 | 97.68±0.01 | 97.59±0.01 |
| 32 | **92.34±0.04** | 92.28±0.04 | 89.45±0.04 | 90.47±0.03 | 90.88±0.04 | 90.58±0.05 | 88.94±0.04 | 89.41±0.04 |

