# OpenReview forum: "Learning Imbalanced Data with Beneficial Label Noise"
_ICLR.cc/2025/Conference — Submitted to ICLR 2025_

### Official Review · Reviewer_kaNw · 2024-10-21

**Soundness:** 3
**Presentation:** 3
**Contribution:** 2
**Rating:** 6
**Confidence:** 4

**Summary:**

The paper introduces the Label-Noise-based Rebalancing (LNR) method to address both binary and multi-class imbalanced classification by using an asymmetric label noise model. Unlike conventional data-level methods, which often suffer from information loss and generative errors, LNR rebalances data by flipping the labels of majority class samples. This technique improves the recognition of minority class samples and can be seamlessly integrated with any classifier or algorithm-level methods. Additionally, the authors provide a theoretical analysis of how label noise affects classifiers, proposing that controlled noise can boost imbalanced learning without the downsides of traditional resampling methods. Experimental results demonstrate the superiority of LNR.

**Strengths:**

1. This paper presents a simple and effective method for addressing data imbalance, and the article is highly readable, making it easy to follow the idea.

2. This paper provides a theoretical analysis of the proposed method and validates the effectiveness of the proposed label-flipping approach.

3. The authors conducted extensive experiments to validate the effectiveness of the proposed method.

**Weaknesses:**

This paper may still have some shortcomings in terms of theoretical analysis and experimental results.

1. The label-flipping method is merely considered a trick, essentially no different from the Mixup strategy [1], and it may disrupt the original data distribution and the feature extraction process [2].

[1] H.  Zhang, M.  Cisse, Y. N.  Dauphin, D. Lopez-Paz,   Mixup: Beyond Empirical Risk Minimization, ICLR 2017.

[2] B.  Zhou,  Q.  Cui,   X.  Wei,   Z. Chen,  BBN: Bilateral-Branch Network with Cumulative Learning for Long-Tailed Visual Recognition, CVPR 2020.

2. While the theoretical analysis strengthens the persuasive power of the label-flipping technique, it is focused on binary classification, and whether it can be correctly extended to multi-class problems requires further discussion and analysis.

3. Since GCL is used as the baseline, but the experimental strategy differs significantly from the original GCL paper, there are some doubts regarding the validity of these experimental results.

4. By manually manipulating labels, the model might develop an unintended bias towards specific classes, especially in cases where the balance is artificially created through extensive relabeling. This could skew predictions and reduce the fairness of the model across different classes. It would be better if the authors could provide the confusion matrix result on CIFAR10-LT datasets.

**Questions:**

See Weaknesses.

---

### Official Review · Reviewer_vGZe · 2024-10-29

**Soundness:** 2
**Presentation:** 3
**Contribution:** 2
**Rating:** 5
**Confidence:** 4

**Summary:**

This paper proposes a new method called Label-Noise-based Re-balancing (LNR) to address the data imbalance problem by introducing label noise in the majority class samples in a classification task. The method aims to adjust the decision boundary and thus improve the classification accuracy of the minority class. The authors provide a detailed theoretical analysis of how label noise can mitigate the effects of class imbalance on decision boundaries. Extensive experimental results are reported in the main text and appendices, which seem to indicate that LNR demonstrates strong performance across a wide range of datasets and is applicable to various classifiers.

**Strengths:**

1) This paper explores a new method to handle imbalanced data in classification tasks by introducing asymmetric label noise, rather than relying solely on traditional resampling methods or algorithm-level adjustments. This noise-based method avoids information loss or generative errors.
2) The paper presents comprehensive mathematical proofs to demonstrate how class imbalance influences decision boundaries and how the proposed method helps to mitigate these effects. Overall, the paper is well-organized, with clear and accessible writing.

**Weaknesses:**

1) Regarding the class imbalance problem, more recent studies can be easily found in Google Scholar or DBLP and should be reviewed and compared (if possible). However, this paper completely ignores the studies after 2022, which cannot strongly support the contributions and innovations of this paper.
2) Regarding the method, the idea of training a classifier after dividing majority class samples near the decision boundary into the minority class to improve the classifier’s performance on the minority class lacks novelty. Additionally, the proposed method still depends on an parameter $t_{flip}$ to complete the division, without an automatic strategy for optimal division, limiting the proposed method’s contribution.
3) Regarding the experimental validation, the chosen experimental metrics and experimental results are not convincing enough. Mislabeling some majority class samples near the decision boundary as minority class samples naturally increases the number of samples classified as the minority class. This, in turn, unsurprisingly boosts the recall of the minority class and the precision of the majority class. These results can not effectively validate the proposed method. Conversely, the improvement of the proposed method is very limited in terms of the metrics that reflect the overall performance (such as "Acc_{overall}" in Table 1 and "Overall" in Table 2).

**Questions:**

In addition to the weaknesses I have already listed, I have the following questions:
1) Since the proposed method is intended to be effective across various classifiers, why is MLP specifically chosen as the flip-rate estimator $C_f$ (see "Our method, LNR, adapts the final layer of the MLP classifier to include a soft-max function as the flip-rate estimator $C_f$" in lines 433-434)? This setup could make KNN’s performance improvement rely on MLP’s performance. Table 5 in the appendix raises this concern, as MLP’s original performance is even better than the improved KNN results.
2) What is the difference between "method", "approach", and "methodology" in this paper, and in what context are these different terms used to describe the proposed method LNR (such as "this method" in line 23, "our LNR approach" in lines 23-24, and "a novel methodology" in line 147)?
3) Does the proposed LNR have any limitations, and if so, in what context do they arise? Limitations of the method should be noted and discussed.

---

> ### Author Response · Authors · 2024-11-23
> **Official Comment by Authors**
>
> Thank you for your thoughtful reviews. We have carefully addressed your concerns and questions in our response. We would appreciate it if you could let us know whether our response has adequately resolved your concerns. We look forward to continuing the discussion.

---

> > ### Comment · Reviewer_vGZe · 2024-11-24
> >
> > Dear authors,
> >
> > I have read your rebuttal and appreciate your efforts in addressing my suggestions. While some of my concerns have been adequately addressed, others remain unaddressed:
> >
> > (1) In your rebuttal, you only responded to W2 and W3, leaving W1, Q1, Q2, and Q3 unaddressed (where W and Q denote Weaknesses and Questions). These unaddressed concerns, particularly Q1 and W1, require attention.
> >
> > (2) Based on your rebuttal, I still find no substantial distinction between your interpretation of "relabel majority class samples that closely resemble the minority class as minority class samples" and my understanding of "dividing majority class samples near the decision boundary into the minority class". This will inevitably improve the recall of the minority class and the precision of the majority class. Therefore, the experimental results on this phenomenon are not sufficient to prove that the LNR proposed in this paper is effective.
> >
> > (3) Regarding the experimental results, your interpretation requires more rigor. While you correctly note that GCL-LNR achieved an overall accuracy exceeding other methods by more than 3%, it's equally important to acknowledge that the performance difference between LDAM-LNR and LDAM-RSG is minimal in both Step-wise Cifar-10 (78.12% vs 77.74%) and Step-wise Cifar-100 (45.63% vs 45.51%). The experiments should clearly demonstrate the effectiveness of LNR as a whole, not just GCL-LNR specifically.
> >
> > Overall, I appreciate the efforts made in the rebuttal. However, your current responses are not sufficient for me to change my final decision.
> >
> > Best regards,
> > PC vGZe

---

> > > ### Author Response · Authors · 2024-11-26
> > >
> > > Dear Reviewer vGZe,
> > >
> > > Thank you for your prompt feedback, and we appreciate your outlining the unaddressed concerns. We have carefully addressed your remaining concerns and prepared detailed responses. Could you kindly review them at your earliest convenience?
> > >
> > > We hope our responses have satisfactorily addressed your key concerns. If anything remains unclear or requires further clarification, please do not hesitate to let us know, and we will address it promptly.
> > >
> > > We look forward to your feedback.
> > >
> > > Best regards,
> > >
> > > Authors of Submission 9243

---

> ### Author Response · Authors · 2024-11-24
>
> Dear reviewer vGZe,
>
> We sincerely appreciate your prompt response.
>
> * In the previous "Response to **questions** of reviewer vGZe" section, we have addressed Q1 and Q3. We have posted two comments that address "Weakness" and "Questions" separately.
>
> * For Q2, we humbly consider this a matter of language usage, as many other works employ similar phrasing. To clarify our intent, in our paper, ‘method’ and ‘approach’ refer to specific algorithms, while ‘methodology’ denotes the theoretical derivation process behind the algorithms, explicitly referring to the strategy of "using label noise to correct the decision boundary." **We will carefully double-check the usage of these three words and make sure they are used consistently in the revised version.**
>
> * Regarding W1, we have not been able to find any of the latest data-level SOTA methods for comparison since 2023. There are many algorithm-level methods, and they are being updated rapidly; however, our approach can be integrated with them while ensuring the safety of LNR. The results from LDAM and GCL demonstrate that LNR can significantly enhance the performance of other algorithm-level methods. We would be very grateful if you could provide some references to the latest data-level methods for us to study and compare.
>
> **Regarding your unaddressed concern W3, we would like to clarify further**
> 1. The illustration in Figure 1 provides a straightforward understanding of how LNR selects majority class samples rather than merely dividing them. For instance, some “4”s and “8”s are highly similar to the minority class “9,” and the overlap of these similar features negatively impacts the classification accuracy for “9.” LNR transfers these features, classified as minority class samples, thereby enriching the feature space of “9” and reducing the influence of majority class samples by relabeling these “4”s and “8”s, which minimizes the impact on the majority classes. Notably, the comparison of the confusion matrices for GCL and LNR on the long-tailed CIFAR-10 dataset reveals that LNR requires only 26 true positives in head classes to achieve 204 true positives in tail classes with just 93 flipped labels (Appendix C.4).
>
> 2. In step-wise imbalanced multiclass datasets, half of the classes—specifically, 5 minority classes in step-wise CIFAR-10 and 50 minority classes in step-wise CIFAR-100—are classified as minority classes with a 100:1 imbalance ratio. Improving performance in these minority classes without degrading overall accuracy is already challenging when learning from highly imbalanced datasets. This situation highlights the difficulty of substantially improving both minority class accuracy and overall accuracy simultaneously.
>
> 3. Regarding the effectiveness of LNR in overall accuracy, the performance improvement of LDAM-RSG over LDAM is also relatively modest on the step-wise CIFAR-10 dataset (77.47\% vs. 77.74\%), and the step-wise CIFAR-100 (45.23\% vs. 45.51\%), The improvement in LDAM delivered by LNR is more significant than RSG: on the step-wise CIFAR-10 dataset (77.47\% vs. **78.12**\%), and the step-wise CIFAR-100 (45.23\% vs. **45.63**\%).  The overall accuracy improvement serves as a supplementary criterion on the step-wise imbalance setting, ensuring that enhancing minority class classification performance does not negatively impact overall accuracy, and LNR also outperforms all other methods.
>
> 4. The scarcity of minority class samples makes their identification even more critical in practical applications. LNR significantly enhances the classification performance of minority class samples compared to the previous state-of-the-art (SOTA) on the step-wise CIFAR-10 dataset (67.02% vs. **75.06%**) and the step-wise CIFAR-100 dataset (21.67% vs. **25.84%**), while preserving the majority class with minimal data editing compared to other methods. Although the overall accuracy shows relatively small improvements over RSG in the step-wise results, LNR's performance in minority classes significantly outperforms RSG and other methods, highlighting the effectiveness of LNR and our primary contributions in addressing the **imbalanced learning problem**.
>
> Thank you for raising these insightful questions, and we welcome any further inquiries you may have regarding our work.

---

> ### Author Response · Authors · 2024-11-30
> **Kindly Remainder - Discussion Period Ends Soon**
>
> Dear Reviewer vGZe,
>
> We appreciate your outlining the unaddressed concerns. We have carefully addressed your remaining concerns and prepared detailed responses. Could you kindly review them at your earliest convenience?
>
> We hope our responses have satisfactorily addressed your key concerns. If anything remains unclear or requires further clarification, please do not hesitate to let us know, and we will address it promptly.
>
> We look forward to your feedback.
>
> Best regards,
>
> Authors of Submission 9243

---

> > ### Comment · Reviewer_vGZe · 2024-12-01
> >
> > Dear Authors of Submission 9243,
> > I appreciate your efforts in addressing my concerns, I have raised my score to 5.
> >
> > vGZe

---

> > > ### Author Response · Authors · 2024-12-02
> > >
> > > Dear reviewer vGZe,
> > >
> > > We sincerely thank the reviewer vGZe for your efforts in reviewing our work and providing us with precious feedback. We are pleased to have been able to address your concerns and truly hope that if you have any further questions, you will feel free to reach out to us. We will do our utmost to clarify any remaining concerns you may have.
> > >
> > > Best wishes,
> > >
> > >  Authors of Submission 9243

---

### Official Review · Reviewer_M9Zh · 2024-10-29

**Soundness:** 3
**Presentation:** 3
**Contribution:** 3
**Rating:** 6
**Confidence:** 4

**Summary:**

The paper presents a novel approach called Label-Noise-based Rebalancing (LNR) to address the challenges posed by imbalanced data and label noise in classification tasks. The authors argue that traditional methods, such as resampling, often lead to information loss and generative errors. They propose that introducing beneficial label noise can help adjust biased decision boundaries, thereby improving classifier performance, particularly for minority classes. The LNR method employs an asymmetric label noise model that selectively flips labels of majority class samples to minority classes, aiming to enhance the performance of classifiers in both binary and multi-class settings. The authors validate their approach through extensive experiments on synthetic and real-world datasets, demonstrating its effectiveness compared to existing methods.

**Strengths:**

1. The paper is well structured and the introduced idea is interesting.
2. The paper offers a solid theoretical analysis of how data imbalance affects decision boundaries.
3. The LNR method is designed to be easily integrated with various classifiers and algorithm-level methods.

**Weaknesses:**

1. The paper could benefit from more detailed implementation guidelines for the label noise model to enhance reproducibility.
2. The paper does not compare with some latest methods on class imbalance.
3. Some results could be presented more clearly, with additional explanations or context to help readers understand the significance of the findings.

**Questions:**

1. How can practitioners effectively estimate the flip-rate for different datasets? Are there specific guidelines or heuristics that can be followed?
2. How does the LNR method perform with larger datasets or in high-dimensional spaces? Are there any computational limitations or considerations?
3. How does LNR compare with other emerging techniques in imbalanced learning that also aim to modify decision boundaries or utilize noise?

---

> ### Author Response · Authors · 2024-11-24
> **Reminder - Discussion Stage Closing Soon - 25 November**
>
> Dear Reviewer M9Zh,
>
> Thank you for taking the time and effort to review our manuscript. We have carefully addressed all your comments and prepared detailed responses. Could you kindly review them at your earliest convenience?
>
> We hope our responses have satisfactorily addressed your key concerns. If anything remains unclear or requires further clarification, please do not hesitate to let us know, and we will address it promptly.
>
> We look forward to your feedback.
>
> Best regards,
>
> Authors of Submission 9243

---

> ### Author Response · Authors · 2024-11-30
> **Kindly Remainder - Discussion**
>
> Dear Reviewer M9Zh,
>
> Thank you for taking the time and effort to review our manuscript and provide valuable feedback. We have thoroughly addressed all of your comments and prepared detailed responses.
>
> We would greatly appreciate it if you could kindly take a moment to review our responses at your earliest convenience. We sincerely hope that our replies have effectively addressed your main concerns. Should you have any further questions or need additional clarification, please feel free to reach out, and we will respond promptly.
>
> Thank you once again for your support. We eagerly await your feedback.
>
> Warm regards,
>
> Authors of Submission 9243

---

> ### Author Response · Authors · 2024-12-03
>
> Dear Reviewer M9Zh,
>
> Thank you for your thoughtful and constructive feedback on our manuscript.
>
> * We are pleased to hear that our revisions to Section 4.2 have addressed your concerns and that you find our implementation guidelines clearer. Your insights have greatly contributed to making our method more comprehensible.
>
> * We appreciate your recommendations regarding the SMOTE-based data-level methods. Although we were unable to provide comparisons with these methods due to the time constraints of the rebuttal period, we will include these comparisons in our revised manuscript.
>
> * Thank you for suggesting that we emphasize the robustness of LNR in the main text. We will ensure that the experimental section of the revised manuscript highlights these findings prominently.
>
> * We also acknowledge your valuable suggestion to clarify the innovative aspect of LNR in leveraging label noise to address imbalanced learning. We will make sure to articulate this more clearly in the revised manuscript.
>
> Overall, we are delighted that you recognize the novelty and uniqueness of our approach and its potential. We hope that our responses and the improvements we make will meet your expectations. We will carefully incorporate all of your suggestions into the revised manuscript.
>
> Thank you once again for your support and guidance.
>
> Best regards,
> Authors of Submission 9243

---

### Official Review · Reviewer_m5qW · 2024-11-02

**Soundness:** 3
**Presentation:** 3
**Contribution:** 2
**Rating:** 5
**Confidence:** 3

**Summary:**

The authors proposed a method for dealing with class imbalance. The main problem with imbalanced datasets is that most labels are from the minority class and few of them are from the majority class. To deal with this, the authors proposed to flip some of the labels in order to balance the dataset and improve the performance of classification models. Moreover, the authors extended their method to multi-class classification problems. Finally, the numerical experiments showed promising results for the method, in a wide range of datasets, including tabular data classification and image data classification problems.

**Strengths:**

The method is generic enough to be used in any classification problem in machine learning, that involves imbalanced data. The numerical experiments demonstrate the value of the method in both tabular data and image data classification problems.

The method has been successfully applied in a very large number of datasets, including 32 tabular datasets and 2 image datasets, and has shown promising results.

The authors extended the method to multi-class classification problems and proposed an efficient training procedure using online label noise.

**Weaknesses:**

The label flipping approach can have a negative impact on the classifier's ability for predicting the majority class.

The flipping decisions of Algorithm 1 are determined based on another classification model (C_f) that is trained on the original imbalanced dataset and therefore they can carry out any error it might have.

What if the distribution of the majority class is very different from that of the minority class ? Would the label flipping approach make sense in this case?

**Questions:**

Can you comment on the effect on out of sample accuracy from your method ? My intuition is that if the test data have the same imbalance ratio as the training data, then accuracy should be worsened from that of the nominal model, while if the test data are more balanced than the training data, then it should be improved.

---

> ### Author Response · Authors · 2024-11-23
> **Official Comment by Authors**
>
> Thank you for your thoughtful reviews. We have carefully addressed your concerns and questions in our response. We would appreciate it if you could let us know whether our response has adequately resolved your concerns. We look forward to continuing the discussion.

---

> ### Author Response · Authors · 2024-11-24
> **Reminder - Discussion Stage Closing Soon - 25 November**
>
> Dear Reviewer m5qW,
>
> Thank you for taking the time and effort to review our manuscript. We have carefully addressed all your comments and prepared detailed responses. Could you kindly review them at your earliest convenience?
>
> We hope our responses have satisfactorily addressed your key concerns. If anything remains unclear or requires further clarification, please do not hesitate to let us know, and we will address it promptly.
>
> We look forward to your feedback.
>
> Best regards,
>
> Authors of Submission 9243

---

> ### Author Response · Authors · 2024-11-30
> **Reminder - Discussion Stage Closing Soon - 1st December**
>
> Dear Reviewer m5qW,
>
> Thank you for taking the time and effort to review our manuscript. We have carefully addressed all your comments and prepared detailed responses. Could you kindly review them at your earliest convenience?
>
> We hope our responses have satisfactorily addressed your key concerns. If anything remains unclear or requires further clarification, please do not hesitate to let us know, and we will address it promptly.
>
> We look forward to your feedback.
>
> Best regards,
>
> Authors of Submission 9243

---

### Author Response · Authors · 2024-12-01

We thank all reviewers for their insightful and constructive feedback. We appreciate that the reviewers found our proposed method to be：

1. **Generality and Versatility**:
   - The method is generic and applicable to any imbalanced data classification problem. (Reviewer m5qW, M9Zh).

2. **Theoretical Insights**:
   - The paper provides a solid theoretical analysis of how data imbalance affects decision boundaries. The method is designed for easy integration with various classifiers and algorithm-level techniques (Reviewer M9Zh,vGZe).

3. **Innovative Approach**:
   - The introduction of a novel method to handle imbalanced data through asymmetric label noise avoids reliance on traditional resampling methods, preventing information loss and generative errors. This approach is supported by comprehensive mathematical proofs (Reviewer vGZe, kaNw).

4. **Clarity and Effectiveness**:
   - The paper presents a simple and effective method for addressing data imbalance along with extensive experiments to validate the effectiveness of the proposed method (Reviewer m5qW, kaNw).

This summary emphasizes the strengths of the paper, with specific endorsements from each reviewer.

Our work offers new insights into the impact of imbalanced data on decision boundaries, providing a foundational and universal approach to imbalanced learning. Specifically:

To our knowledge, we are the first to propose a method that artificially introduces label noise to correct the biased decision boundary.
Our approach achieves data balance by simply flipping labels. This simplicity makes it versatile across various data formats and classification tasks, allowing seamless integration with other methods and classifiers.
**We believe our work opens new avenues for addressing the imbalance learning problem**. We have incorporated the reviewers' suggestions by adding explanations and experimental results during the rebuttal period and hope these updates will be considered in the final evaluation.

We have diligently addressed the primary concerns raised by the area chair and each reviewer, which can be summarized into two major concerns:
1. The distinction between LNR and other data augmentation methods, such as Mixup.

2. The potential impact of the label noise introduced by LNR:
 * whether it disrupts the feature extraction process and negatively affects classifier performance.
 * whether the artificially introduced label noise undermines class fairness.

As a reply:
1. We succinctly summarize the distinction between LNR and other data augmentation methods.
* The fundamental objective of all data-level approaches is to enhance the feature representation of minority class samples. Before LNR, nearly all data-level methods focused on various strategies to generate new samples aimed at augmenting minority class features, constantly refining these processes to minimize generative errors, as exemplified by the advancements from Mixup to SelMix.
* In contrast, LNR adopts a fundamentally different approach by developing a beneficial noise model that involves flipping the labels of majority class samples. This enables the direct reutilization of majority class features to enrich the sample space of the minority class, without the necessity of generating any synthetic samples.
* By modifying only the labels, LNR effectively avoids generative errors while allowing for integration with other methods to provide additional performance enhancements, highlighting the novelty and versatility of our LNR method.

2. In response to the second concern:

* The t-flip parameter can effectively mitigate the negative impact on classifier performance caused by introducing excessive label noise. Since LNR is a data-driven method, the t-flip parameter can be validated through cross-validation (CV) to ensure that this label noise remains harmless.
* LNR employs a strategy similar to Deferred Re-Weighting/Deferred Re-Sampling, introducing label noise only during the fine-tuning phase, thereby effectively avoiding any adverse effects on the feature extraction process.
* The strength of label flipping in LNR depends on the similarity between samples, ensuring that class fairness is maintained. This is demonstrated by the presented confusion matrix on CIFAR-10-LT, which proves that the enhancements brought by LNR to GCL do not come at the expense of class fairness.

The difference in confusion matrices from GCL to GCL-LNR:
|Diff|0|1|2|3|4|5|6|7|8|9|
|------|:-------:|:-------:|:------:|:-------:|:------:|:------:|:------:|:-------:|:-------:|:--------:|
|0|**-26**|+2|-4|0|+1|-2|0|-1|**+13**|**+16**|
|1|1|**-4**|0|-1|+1|0|0|0|0|**+4**|
|2|5|+1|**-4**|-2|0|-7|+1|+3|-1|+3|
|3|-2|-1|+3|**+18**|-4|-13|-5|+2|0|+5|
|4|0|0|-2|-3|**+3**|-5|0|+5|+1|+1|
|5|2|0|-1|-3|0|**-1**|+1|+2|0|+2|
|6|1|+2|-2|+6|-1|-5|**-4**|+2|-1|+1|
|7|-5|0|+3|-20|-1|-8|+4|**+22**|+1|+5|
|8|**-91**|-6|+2|-2|-1|-1|-2|0|**+80**|+20|
|9|**-42**|**-48**|-3|-5|+1|-2|-3|-2|+3|**+102**|

---

### Author Response · Authors · 2024-12-03
**Reviewer m5qW not engaged into discussion at all**

Dear Area Chair,

The discussion deadline is fast approaching. However, we have not yet received any response from Reviewer m5qW, who initially rated our work with scores of 5, respectively. We would greatly appreciate your guidance in facilitating communication with them.

We are pleased to share that the other three reviewers have engaged in constructive discussions regarding our rebuttal, leading two of them (Reviewer vGZe and Reviewer kaNw) to increase their scores after we addressed their concerns. We believe this positive feedback reflects the improvements made to our work.

Thank you very much for your attention to this matter. We are grateful for your support, and we look forward to your assistance.

Best regards,

Authors of Submission 9243

---

### Meta-Review · Area_Chair_HVM2 · 2024-12-22

**Metareview:**

The paper concerns a problem of imbalance learning. The Authors introduce a new balancing technique that relies on random flipping of labels of training examples from the majority class. The numerical experiments show promising results of the introduced method.

The Reviewers find the paper well-written and appreciate the theoretical results, as well as the introduced method, which is model-agnostic, applicable to multi-class classification, and produces promising results. They also point out weaknesses of the paper such as limited discussion and comparison to related work or the choice of experimental metrics that makes the "experimental results not convincing enough."

As an AC I also posted my critical remarks concerning limited theoretical contribution and insufficient discussion on related work.  In fact, the presented results are rather known, limited to two specific metrics without any generalization to a broader class of performance measures. The paper also lacks a clear theoretical analysis of the introduced method how well it can optimize a given metric of interest. Furthermore, the empirical part does not contain any comparison to methods that directly optimize the considered metrics.

Taking into account the above and the ratings of the Reviewers, which indicate this paper as borderline, I recommend to reject this paper.

**Additional Comments On Reviewer Discussion:**

The Authors managed to clarify some of the Reviewers' doubts, which led to an increase in ratings. Nevertheless, even after the changes, the paper was still considered borderline. None of the Reviewers championed the paper during the final phase of the discussion.

The Authors' response to my critical remarks is only partially satisfactory. They need to revise and extend their results by incorporating existing knowledge on complex performance metrics. Additionally, they should properly compare their results to methods that directly optimize the performance metrics of interest.

---

### Decision · Program_Chairs · 2025-01-22

Reject